# Chemistry and deposition in the Model of Atmospheric composition at Global and Regional scales using Inversion Techniques for Trace gas Emissions (MAGRITTEv1.1). Part A. Chemical mechanism

Jean-François Müller[1], Trissevgeni Stavrakou[1], and Jozef Peeters[2]

[1]Royal Belgian Institute for Space Aeronomy, Avenue Circulaire 3, 1180, Brussels, Belgium
[2]Department of Chemistry, University of Leuven, Celestijnenlaan 200F, B-3001 Leuven, Belgium
**Correspondence:** Jean-François Müller (Jean-Francois.Muller@aeronomie.be)

**Abstract.** A new chemical mechanism for the oxidation of biogenic volatile organic compounds (BVOCs) is presented and implemented in the Model of Atmospheric composition at Global and Regional scales using Inversion Techniques for Trace gas Emissions (MAGRITTE v1.1). With a total of 105 organic species and over 265 gas-phase reactions, 69 photodissociations and 7 heterogeneous reactions, the mechanism treats the chemical degradation of isoprene – its main focus – as well as acetaldehyde, acetone, methylbutenol and the family of monoterpenes. Regarding isoprene, the mechanism incorporates a state-of-the-art representation of its oxidation scheme accounting for all major advances put forward in recent theoretical and laboratory studies. The recycling of OH radicals in isoprene oxidation through the isomerisation of $Z$-$\delta$-hydroxyperoxy radicals is found to enhance OH concentrations by up to 40% over Western Amazonia in the boundary layer, and by 10-15% over Southeastern U.S. and Siberia in July. The model and its chemical mechanism are evaluated against the suite of chemical measurements from the SEAC[4]RS (Studies of Emissions and Atmospheric Composition, Clouds and Climate Coupling by Regional Surveys) airborne campaign, demonstrating a good overall agreement for major isoprene oxidation products, although the aerosol hydrolysis of tertiary and non-tertiary nitrates remain poorly constrained. The comparisons for methylnitrate indicate a very low nitrate yield ($< 3 \cdot 10^{-4}$) in the $CH_3O_2+NO$ reaction. The oxidation of isoprene, acetone and acetaldehyde by OH is shown to be a substantial source of enols and keto-enols, primarily through the photolysis of multifunctional carbonyls generated in their oxidation schemes. Oxidation of those enols by OH radicals constitutes a sizable source of carboxylic acids estimated at 9 Tg (HC(O)OH) yr$^{-1}$ and 11 Tg(CH$_3$C(O)OH) yr$^{-1}$, or ~20% of their global identified source. The ozonolysis of alkenes is found to be a smaller source of HC(O)OH (6 Tg HC(O)OH yr$^{-1}$) than previously estimated, due to several factors including the strong deposition sink of hydroxymethylhydroperoxide (HMHP).

## 1 Introduction

The terrestrial biosphere is, by far, the largest source of non-methane volatile organic compounds (NMVOCs) to the global atmosphere (Guenther et al., 2012). Because those biogenic VOCs (BVOCs) are generally very reactive, their chemical degradation takes mostly place in the boundary layer, in the vicinity of the emission regions, where they have a strong impact on

the budget of oxidants and the formation and growth of secondary organic aerosol (SOA), a major component of fine particulate matter (Seinfeld and Pandis, 2006; Hallquist et al., 2009). Even far away from those regions, longer-lived intermediates generated in their oxidation (e.g. organic nitrates and peroxynitrates) have a large impact on nitrogen oxides (NOx), hydroxyl radical (OH) and ozone levels (Paulot et al., 2012).

Among the BVOCs, isoprene has by far the largest global emissions, of the order of 500 Tg yr$^{-1}$, representing about 50% of all BVOCs; other major biogenic compounds in terms of emissions include the monoterpenes, methanol, acetone, acetaldehyde, and ethanol (Guenther et al., 2012). The complex chemical degradation mechanism and the profound impact of isoprene on air quality and the climate has been the topic of numerous field (Trainer et al., 1987; Claeys et al., 2004; Lelieveld et al., 2008; Hofzumahaus et al., 2009; Toon et al., 2016; Carlton et al., 2018; Mao et al., 2018), laboratory (Tuazon and Atkinson,
1989; Paulot et al., 2009a, b; Crounse et al., 2011; Wolfe et al., 2012; Kwan et al., 2012; Lin et al., 2013; Fuchs et al., 2013; Bates et al., 2014, 2016; Nguyen et al., 2015a, 2016; Schwantes et al., 2015; Teng et al., 2017; Novelli et al., 2018b; Berndt et al., 2019), theoretical (Peeters et al., 2009; Kjaergaard et al., 2012; Crounse et al., 2013; Peeters et al., 2014; Peeters and Nguyen, 2012; Liu et al., 2017; Praske et al., 2018; Møller et al., 2019) and modelling studies (Stavrakou et al., 2010; Paulot et al., 2012; Taraborrelli et al., 2012; Jenkin et al., 2015; Squire et al., 2015; Travis et al., 2016; Lelieveld et al., 2016; Silva et al.,
2018; Stadtler et al., 2018; Mao et al., 2018; Li et al., 2018).

Our understanding of isoprene oxidation has expanded considerably in the last decade. Most importantly perhaps, the traditional views regarding the fate of large, multifunctional peroxy radicals formed in the oxidation of isoprene and other NMVOCs has been radically altered by the realization that H-shift reactions in such radicals can sometimes be fast enough to compete with, or even outrun, their reactions with nitric oxide and peroxy radicals (Peeters et al., 2009; Crounse et al., 2011; Teng et al.,
2017). The impact of the 1,6 H-shifts in allylic peroxy radicals formed in the oxidation of isoprene by OH is enhanced by their thermal instability allowing fast interconversion of the different peroxy isomers/conformers (Peeters et al., 2009), such that the 1,6 H-shifts can compete with the conventional bimolecular reactions for the entire pool of initial peroxys, which greatly affects the product yields (Peeters and Müller, 2010; Peeters et al., 2014; Teng et al., 2017). Other examples of peroxy isomerization reactions shown to be of importance include 1,4 aldehyde H-shifts (Asatryan et al., 2010; Crounse et al., 2012) and the
very fast enol-H-shifts (Peeters and Nguyen, 2012) as well as hydroperoxide H-shifts (Jørgensen et al., 2016). The resulting autoxidation reactions generate multifunctional hydroperoxides shown in some cases (in monoterpene oxidation) to be of such extremely low volatility as to play a crucial role in SOA and cloud condensation nuclei (CCN) formation (Crounse et al., 2013; Jokinen et al., 2014, 2015), while in other cases, they are believed to be an important source of HOx radicals through photodissociation (Peeters and Müller, 2010; Wolfe et al., 2012; Liu et al., 2017, 2018). The recycling of HO$_x$ radicals associated
with peroxy H-shifts and their subsequent reactions, as well as with other previously unsuspected reactions such as epoxide formation from activated hydroxy hydroperoxy radicals (Paulot et al., 2009a) has led to a reassessment of the overall impact of isoprene (and other BVOCs) on OH and HO$_2$ levels, now found to be fairly consistent with HOx measurements in isoprene photooxidation experiments (Fuchs et al., 2013; Novelli et al., 2018b) as well as in field experiments in isoprene-rich, low-NOx environments (Bottorff et al., 2018). The importance of isoprene-derived epoxides stems from their major role as precursors
of SOA demonstrated by laboratory and field measurements (Paulot et al., 2009a; Surratt et al., 2010; Lin et al., 2012, 2013).

Finally, the impact of isoprene on NOx levels has been also reevaluated due to a better assessment of organic nitrate formation in isoprene oxidation by OH (Paulot et al., 2009b; Lee et al., 2014; Teng et al., 2017; Wennberg et al., 2018) and $NO_3$ (Kwan et al., 2012; Schwantes et al., 2015; Wennberg et al., 2018) as well as of the balance between NOx-recycling pathways such as photolysis (Müller et al., 2014) and NOx terminal losses through heterogeneous hydrolysis in aqueous aerosols

(Romer et al., 2016) and dry deposition (Nguyen et al., 2015b).

A proper model assessment of the role of BVOCs in the global troposphere and in issues such as air quality and the interaction between the biosphere, the atmosphere and the climate requires the implementation of up-to-date, state-of-the-art chemical mechanisms in large-scale (global or regional) models. Whereas completely explicit mechanisms are not advisable due to computational cost concerns, oversimplified mechanisms are clearly not appropriate as tools to explore the impact of

10 mechanistic changes, especially in the context of the rapid evolution of our understanding of the mechanisms. We present here a semi-explicit mechanism of intermediate complexity incorporating the major advances reported above. It covers the oxidation of isoprene, monoterpenes, methanol, acetone, acetaldehyde, ethanol and 2-methyl-3-butene-2-ol (short-handed as methylbutenol or MBO). This mechanism is implemented in the Model of Atmospheric composition at Global and Regional scales using Inversion Techniques for Trace gas Emissions (MAGRITTE v1.1) which is based on the previous global model

IMAGES (Muller and Brasseur, 1995; Stavrakou et al., 2009a, b, 2015; Bauwens et al., 2016).

Given the very large uncertainties in monoterpene oxidation, their treatment is still very crude in the mechanism, the focus being put on the formation yield of important products. Regarding isoprene, the mechanism relies on the Leuven Isoprene Mechanism (Peeters et al., 2009, 2014), on the extensive, explicit Caltech oxidation mechanism (ca. 900 reactions and 400 species) recently presented by Wennberg et al. (2018) based on a critical appraisal of the relevant theoretical and laboratory

studies, and on the very recent experimental investigation of Berndt et al. (2019). For other reactions not addressed in those studies, it also relies on the Master Chemical Mechanism (MCM) (Saunders et al., 2003; Jenkin et al., 2015) and on our own evaluation. The mechanism also incorporates important new mechanistic developments related to e.g. the revisited role of hydroperoxycarbonyl photolysis (Liu et al., 2017, 2018) and the fate of enols and keto-enols produced from such processes. Due to these developments, the oxidation of isoprene as well as of other compounds (e.g. acetone and acetaldehyde) by OH entails

a previously unsuspected source of formic and acetic acid, for which atmospheric observations suggest the existence of large missing sources (Paulot et al., 2011; Stavrakou et al., 2012; Millet et al., 2015) especially since the $HC(O)OH$ source due to alkene ozonolysis through the Criegee Intermediate $CH_2OO$ recently turned out smaller than previously thought (Sheps et al., 2017; Allen et al., 2018).

The complete chemical mechanism of BVOC oxidation is presented in Sect. 2. The parameterization of Henry's law con-

30 stants and dry deposition velocities is presented and evaluated in a companion paper (Müller et al., 2018). Simulations with the MAGRITTE model and the updated chemical mechanism are presented in Sect. 4, including an evaluation against airborne measurements over the Eastern United States (Sect. 4.3) and a presentation of the global sources of carboxylic acids (Sect. 4.4) and glyoxal(Sect. 4.5) resulting from the implementation of the chemical mechanism.

## 2 The chemical mechanism of BVOC oxidation in MAGRITTE

The list of chemical species and the complete gas-phase BVOC oxidation mechanism are given in Tables 1–3.

### 2.1 Isoprene + OH

#### 2.1.1 Initial steps of mechanism

To limit the number of species and reactions, the OH-adducts formed from the reaction of isoprene with OH are not explicitly represented, and the isoprene peroxys are lumped into three compounds: ISOPBO2 and ISODO2 resulting from addition of OH to carbons 1 and 4, respectively, and ISOPEO2 resulting from OH addition to the central carbons (see Peeters et al. (2014) regarding carbon numbering). For example, ISOPBO2 includes the 1,2-OH-peroxy as well as the 1,4-OH-peroxy which can undergo a 1,6-H shift leading to a $\delta-$hydroperoxy aldehyde (HPALD1) and other products. The ratio of OH addition

to $C_4$ to addition to $C_1$ is 37:63 (Wennberg et al., 2018). Based on a detailed steady-state analysis, the bulk isomerisation rate of ISOPBO2 and ISOPDO2 was shown to increase linearly with the sink rate ($k_p$) of the traditional peroxy reactions (Peeters et al., 2014). The reason for this behaviour is that at low $k_p$, the ratio of the $Z$-$\delta$-OH-peroxys over the lower-energy $\beta$-OH-peroxys is close to their equilibrium ratio, of order of only $\sim$0.01, whereas at the high $k_p$ limit, where all peroxys have a similar lifetime, their ratio is governed by their initial formation branching ratio, which is an order magnitude higher

(Peeters et al., 2014; Teng et al., 2017). The following expressions of the bulk 1,6 isomerisation rates are obtained by linear regression of the bulk rates between 285 and 305 K, based on the experimental estimates of the peroxy unimolecular reaction rates (Teng et al., 2017; Wennberg et al., 2018):

$$k_{\text{ISOPBO2}}^{1,6} = 3.409 \cdot 10^{12} \cdot \exp\left(-10698/T\right) + k_p \cdot 1.07 \cdot 10^{-3} \cdot \exp\left(64/T\right) \tag{1}$$

$$k_{\text{ISOPDO2}}^{1,6} = 4.253 \cdot 10^8 \cdot \exp\left(-7254/T\right) + k_p \cdot 2.33 \cdot 10^{-7} \cdot \exp\left(3662/T\right) \tag{2}$$

The steady-state $Z$-$\delta$-OH-peroxy / $\beta$-OH-peroxy ratio is essentially always established in the atmosphere and remains constant in time at given temperature and NO/HO$_2$ levels, as implied in our approach to represent the bulk peroxy isomerization rate. Note that the steady-state ratio used here, based on the RO$_2$ kinetic coefficients of Teng et al. (2017), differs only slightly from the ratio based on the kinetic coefficients of LIM1 (Peeters et al., 2014) and MCM 3.3.1 (Jenkin et al., 2015).

For the practical purposes of model implementation, the bulk isomerisation rates being dependent on the concentrations of NO and HO$_2$, these reactions are split artificially into an unimolecular reaction and several pseudo-two-body reactions of ISOPBO2 and ISOPDO2 with NO and HO$_2$.

For the 1,5 H-shift reactions of the $\beta$-OH-peroxy radicals, we use their theoretically estimated rates (Peeters et al., 2014) multiplied by 0.95 for ISOPBO2, and 0.94 for ISOPDO2, to account for the small fraction of $\delta$-OH-peroxy radicals not

undergoing those reactions (see Sect. 2.1.3). This parameterizion of the bulk 1,5 and 1,6-H-shifts leads to product yields in excellent agreement with an exact estimation based on the kinetic parameters of Wennberg et al. (2018), as seen on Fig. 1.

### 2.1.2 Products from the isomerization of the $Z$-$\delta$-OH-peroxys

The 1,6 H-shift of the $Z$-$\delta$-OH-peroxys $HOCH_2-C(CH_3)=CH-CH_2O_2$ (Case I) and $O_2CH_2-C(CH_3)=CH-CH_2OH$ (Case II) forms allylic radicals, e.g. $Z$-$HOC°H-C(CH_3)=CH-CH_2OOH \Leftrightarrow Z$-$HOCH=C(CH_3)-C°H-CH_2OOH$ for Case I. Therefore, two second-generation peroxys can result, peroxy $i$ ($Z$-$HOCH(O_2)-C(CH_3)=CH-CH_2OOH$) and per-
oxy $ii$ ($Z$-$HOCH=C(CH_3)-CH(O_2)-CH_2OOH$), in an approximate ratio of 40:60a two pathways are open to product formation (Peeters et al., 2014). The subsequent chemistry is given here for Case I, unless stated otherwise. Peroxy $i$ readily eliminates $HO_2$ at a rate of $\sim$2000 s$^{-1}$ (Hermans et al., 2005) to produce $Z$-$O=CH-C(CH_3)=CH-CH_2OOH$ (HPALD1) (Peeters et al., 2014, 2009; Crounse et al., 2011; Teng et al., 2017). Peroxy $ii$ may isomerise by a fast 1,6 enol-H-shift, promptly at $\sim$1.5·10$^9$ s$^{-1}$ and thermally at $>$10$^4$ s$^{-1}$, to form $Z$-$O=CH-C°(CH_3)-CH(OOH)-CH_2OOH$ (Peeters and Nguyen,
2012; Peeters et al., 2014) that in part arises chemically activated such that it can promptly undergo concerted OH-loss and ring-closure to an hydroperoxy-carbonyl epoxide, $Z$-$HOOCH_2-\overline{CHOC}(CH_3)-CHO$ (HPCE), as proposed and observed by Teng et al. (2017), and for another part lead to a third-generation peroxy, $Z$-$O=CH-C(CH_3)(O_2)-CH(OOH)-CH_2OOH$ (DIHPCARP1) (Peeters et al., 2014). The DIHPCARP radicals were suggested (Peeters et al., 2014) to either undergo a fast aldehyde-H-shift and eliminate CO and expel OH to form dihydroperoxy carbonyls, or react with NO and HO$_2$, to result
mainly in OH + $CH_3C(O)CHO$ (MGLY) + $HOOCH_2CHO$ (HPAC) (Case I), or OH + OCHCHO + $CH_3C(O)CH_2OOH$ (HPACET) (Case II). While the CO elimination above may be fast enough to outrun O$_2$ addition for Case I (Novelli et al., 2018b), this appears less likely for Case II, for which the barrier should be about 2 kcal mol$^{-1}$ higher (Méreau et al., 2001). Note that HPAC and HPACET were observed by Teng et al. (2017), but in a ratio to HPALDs nearly independent of the NO level. Secondly, it is estimated using statistical rate theory that the 1,6 enol-H-shift above can occur for about half while its
peroxy precursor is still chemically activated such that the resulting radical contains close to 30 kcal mol$^{-1}$ internal energy (Peeters et al., 2014), sufficient for prompt HPCE epoxide formation.

In this work, the quantitative product distribution from the 1,6 H-shift of the $Z$-$\delta$-OH-peroxys is adopted from the recent experimental study of Berndt et al. (2019), supported and complemented by computational results of the LIM1 paper (Peeters et al., 2014). Note that the 1,6 H-shifts of the $Z$-$\delta$-OH-peroxys occur for $\sim$85% by tunneling (Coote et al., 2003) at
energies lower than 2 kcal mol$^{-1}$ below the barrier top, such that the Boltzmann population there is only marginally affected by the O$_2$-loss that occurs only at energies above this range; therefore there is no reason to suspect (Wennberg et al., 2018) that the agreement between experimental results (Teng et al., 2017) and the TST-predicted rate constants of Peeters et al. (2014) is fortuitous. The Berndt et al. investigation offers several advantages: (i) the reaction time was so short (8 s) that no secondary products could be formed; (ii) due to the absence of NO and near-absence of HO$_2$, essentially only the products of the $Z$-$\delta$-
OH-peroxy 1,6 H-shift could be formed, so excluding potential interferences; (iii) the peroxy radicals could also be observed; (iv) the sampled products and peroxy radicals could be quasi-quantitatively converted into ion-complexes, detected by high-resolution mass spectrometry capable of measuring concentrations as low as 10$^4$ cm$^{-3}$. Hydroxyl radicals were prepared by reacting 10$^{12}$ cm$^{-3}$ of O$_3$ with 2·10$^{11}$ cm$^{-3}$ of tetramethylethylene, in presence of 2.5·10$^{12}$ cm$^{-3}$ of isoprene. At 8 s reaction time, the modeled total ISOPOO concentration is 1.2·10$^9$ cm$^{-3}$, of which 6·10$^6$ cm$^{-3}$ $Z$-$\delta$-peroxys (50% Case I isomer

HOCH$_2$C(CH$_3$)=CHCH$_2$O$_2$, and 50% Case II isomer O$_2$CH$_2$C(CH$_3$)=CHCH$_2$OH at 8 s). Integrated over the entire reaction time of 8 s, the modeled ratio of these two peroxys is circa 0.8:1.0. Using the isomer-specific 1,6 H-shift rates of 0.36 s$^{-1}$ and 3.7 s$^{-1}$ for $Z$-$\delta$-OH-peroxys I and II (Teng et al., 2017), the expected total formation rate of isomerization products at 8 s is 1.2·10$^6$ cm$^{-3}$ s$^{-1}$. For these conditions, Berndt et al. measured the following concentrations at 8 s: C$_5$H$_8$O$_3$ (HPALDs):

2.3·10$^7$ cm$^{-3}$; C$_5$H$_8$O$_4$ (hydroperoxy carbonyl epoxides): 4.6·10$^6$ cm$^{-3}$; C$_4$H$_8$O$_5$ (dihydroperoxy carbonyls): 6.2·10$^5$ cm$^{-3}$; C$_5$H$_9$O$_5$ (the second-generation peroxys above): 1.7·10$^6$ cm$^{-3}$ ; and C$_5$H$_9$O$_7$ (the third-generation peroxys): 3.5·10$^5$ cm$^{-3}$. In principle, these values are minimum concentrations. No HPAC nor HPACET was detected. The detected product and peroxy concentrations account together for 60% of the modeled total products at 8 s using the experimental kinetic parameters of Teng et al. (2017), which, together with the uncertainties, leaves room for some other products. The theoretically derived

parameters of Peeters et al. (2014) predict a higher product formation from the $Z$-$\delta$-OH-peroxy isomerization at 8 s, but this is due to a too low LIM1-predicted O$_2$-loss from the peroxys, such that the populations of the $Z$-$\delta$-OH-peroxys at 8 s are still too close to their high initial formation fraction and attain their much lower final steady-state fraction too late.

     The Berndt et al. results thus give the following product yields at 8 s: HPALDs: 76%; HPCE: 15%; dihydroperoxy carbonyls: 2%; while 5.5% of the reacted $Z$-$\delta$-OH-peroxys is present as second-generation peroxys C$_5$H$_9$O$_5$ and 1% as third-generation

peroxys C$_5$H$_9$O$_7$. The HPALD yield determined by Berndt et al. is much higher than that of Teng et al. (2017). However, another, non-HPALD, C$_5$H$_8$O$_3$ compound observed by Teng et al. could be speculated to be a perhemiketale formed from HPALDs on the walls of the 1 m sampling tubing. Another observation of Berndt et al. indirectly supports a high HPALD yield. The concentration of the second-generation peroxys is strikingly high, given that the peroxys of type $i$ are expected to react at a rate of $\sim$2000 s$^{-1}$ and those of type $ii$ even at $> 10^4$ s$^{-1}$, such that at the given $Z$-$\delta$-OH-peroxys concentrations, and using the

experimental 1,6 H-shift rates for $Z$-$\delta$-OH-peroxys I and II, they should be present in a quasi- steady-state concentration of only about 10$^4$ cm$^{-3}$. This indicates that a large fraction of the C$_5$H$_9$O$_5$ peroxys are $Z,E'$-HOCH=C(CH$_3$)$-$CH(O$_2$)$-$CH$_2$OOH isomers of peroxy $ii$ (and similar for Case II) with the OH pointing outwards, away from the peroxy function, such that they cannot undergo the 1,6 enol-H-shift, and can only be removed by (repeated ) O$_2$-loss and re-addition, to finally convert to $Z,E'$-HOCH(O$_2$)C(CH$_3$)=CHCH$_2$OOH peroxys $i$ that quickly expel HO$_2$ to form additional HPALDs. Such a high fraction of

$Z,E'$ peroxys $ii$ is consistent with the computational results (Peeters et al., 2014) on the various transition states for the 1,6 H-shift of the $Z$-$\delta$-OH-peroxys. For Case I, a $Z,Z'$-TS with the OH inward was found to account for about 67% of the rate and a $Z,E'$-TS with OH outward for 33%, while for Case II two $Z,E'$-TSs account for 69% and a $Z,Z'$-TS for 31% of the rate. For the conditions of Berndt et al. at 8 s, with the integrated 1,6 H-shift rate due for $\sim$92% to the Case II- and for $\sim$8% to the Case I-$Z$-$\delta$-OH-peroxys, the weighted average is $\sim$65% reaction through $Z,E'$- and 35% through $Z,Z'$-structures. Taken

together, the above strongly suggests that, contrary to a speculative suggestion in the LIM1 paper, the $Z-E$ isomerism of the transition states is conserved in the allylic-radical products and in the resulting peroxys $i$ and $ii$. A statistical rate estimate for the prompt internal rotation of the OH in the $Z,E'$-hydroxyl-allyl product radicals, with computed barrier 12 kcal mol$^{-1}$ and imaginary frequency close to 100 cm$^{-1}$, and for a nascent vibration energy of 21 kcal mol$^{-1}$, predicts $k \sim 10^8$ s$^{-1}$, or 10 times slower than collisional stabilization followed by O$_2$-addition. Therefore, allowing for 10% internal rotation of the OH

in the nascent $Z,E'$ product isomers to form the more stable, H-bonded $Z,Z'$ forms, about 40% of the allylic radicals and

their $O_2$-adducts would end up with the OH inwards and $\sim$60% with the OH outwards in the Berndt et al. conditions. Further adopting also the spin densities in the allylic product radical of the LIM1 paper, i.e. 0.4 on carbon 1 and 0.6 on carbon 3 for Case I (and similarly 0.4 on carbon 4 and 0.6 on carbon 2 for Case II), as well as the corresponding 40:60 branching ratio for peroxy $i$ and $ii$ formation, the mechanism above would result in 40% direct formation of HPALDs through peroxy $i$, only 24% enol-H-shift products through $Z,Z'$ peroxy $ii$, and 36% formation of the slowly reacting $Z,E'$ peroxy $ii$, which in the Berndt et al. conditions would lead to ca. 31% indirect HPALD production through $O_2$-loss and re-addition of the $Z,E'$ peroxy $ii$ to form peroxy $i$, while around 5% still survives as $Z,E'$ peroxy $ii$ in the short reaction time available. The so predicted overall 71% HPALD yield, based on computational results from the LIM1 paper, is strikingly close to the experimental yield of Berndt et al.. Moreover, at a total product formation rate of $1.2 \cdot 10^7$ cm$^{-3}$ s$^{-1}$, the 31% contribution due to $Z,E'$ peroxy $ii$ reacting to HPALDs at 8 s implies a reaction rate of $3.8 \cdot 10^6$ cm$^{-3}$ s$^{-1}$, or at the measured $Z,E'$-peroxy $ii$ concentration of $1.7 \cdot 10^6$ cm$^{-3}$, an effective rate constant of 2.2 s$^{-1}$. Since on average 2.5 cycles of $O_2$-loss and re-addition are required to form HPALD from $Z,E'$ peroxy $ii$ through peroxy $i$, an $O_2$-loss rate of 6 s$^{-1}$ is derived, which is typical for hydroxy-allyl peroxys such as the very similar initial $Z$- and $E$-$\delta$-OH-peroxys from isoprene (Teng et al., 2017).

The 15% HPCE yield measured by Berndt et al. is compatible with the product radical of the 1,6 enol-H-shift of $Z,Z'$-peroxy $ii$ arising for a large fraction with sufficient chemical activation to overcome the barrier of ca. 15 kcal mol$^{-1}$ for the concerted ring-closure and OH loss. The theory-based 24% enol-H-shift products through peroxy $ii$, above, comprises the HPCE epoxides and products of the third-generation peroxys (DIHPCARP). Adopting the experimental 15% HPCE yield would leave room for some 10 % DIHPCARP-derived products, of which, apparently, the dihydroperoxy carbonyls account for only a small fraction of 2%. The minimum concentration of the DIHPCARPs in the Berndt et al. experiment is $3.5 \cdot 10^5$ cm$^{-3}$, while their loss rate by aldehyde-H shift (followed by either CO elimination or $O_2$-addition) should be about 2 s$^{-1}$ according to Møller et al. (2019), and 6 s$^{-1}$ according to Novelli et al. (2018c), such that their expected reaction rate is $0.7$–$2.1 \cdot 10^6$ cm$^{-3}$ s$^{-1}$, or 6–18% of the overall products formation rate of $1.2 \cdot 10^7$ cm$^{-3}$ s$^{-1}$ above. Subtracting the 2% dihydroperoxy carbonyls leaves 4–16 % going to other products, consistent with the $\sim$8% estimated above, and in line with the expectation, in the introduction of this section, that the acyl product of aldehyde-H-shift in the most abundant DIHPCARP (Case II) does not eliminate CO but rather adds $O_2$ to continue the autoxidation chain by forming fourth-generation peroxys $C_5H_9O_9$, with $HOOCH_2-C(CH_3)(O_2)-CH(OOH)-C(O)OOH$ (DHPAO2) likely the most stable isomer after fast hydroperoxide-H shifts (Jørgensen et al., 2016) because it allows three H-bonds of which two are synergic and therefore stronger (Dibble, 2004). Since (other) fast H-shifts for this isomer are not possible, it can only react with NO or $HO_2$. The main resulting oxy product radical should decompose rapidly (Vereecken and Peeters, 2009) into HPACET + OH + OCHC(O)OOH.

In atmospheric conditions, the various peroxys are all in quasi-steady state, which means $\sim$5% more HPALD production from the $Z,E'$-peroxys $ii$, and $\sim$1% more DIHPCARP products than in the Berndt et al. conditions at 8 s. On the other hand, the atmospheric steady-state product formation ratio from the $Z$-$\delta$-OH-peroxys Case I and Case II is rather 18:82, instead of the 8:92 ratio of the Berndt et al. experiment (Teng et al., 2017), such that about 43% of the second-generation radicals would end up with the OH inwards and $\sim$57% with the OH outwards. Taking into account also the above, direct (40%) plus indirect (34%) HPALD formation would add up to 74%, while the expected HPCE yield is 16% and that of the DIHPCARP products

around 10%, of which 2% the dihydroperoxy carbonyl DHPMEK. Acknowledging the large uncertainties in those yields, we represent the $Z$-$\delta$-OH-peroxy isomerisations as

$$\text{ISOPBO2} \rightarrow 0.75\,(\text{HPALD1} + \text{HO}_2) + 0.15\,(\text{HPCE} + \text{OH}) + 0.1\,(\text{DHPMEK} + \text{CO} + \text{OH})$$

$$\text{ISOPDO2} \rightarrow 0.75\,(\text{HPALD2} + \text{HO}_2) + 0.15\,(\text{HPCE} + \text{OH}) + 0.1\,(\text{DHPAO2})$$

Here, HPCE is a mixture of 18% Case I and 82% Case II compounds. Its oxidation by OH proceeds mainly by aldehyde-H abstraction, forming a carbonyl radical; the same radical can also be formed through OH-abstraction of the hydroperoxide-H in HPCE, followed by a 1,6 aldehyde-H-shift. The carbonyl radical can undergo concerted CO elimination and ring opening, forming $\text{CH}_3\text{C(O)CH(O}_2)\text{CH}_2\text{OOH}$ (for Case I) or $\text{OCHC(O}_2)(\text{CH}_3)\text{CH}_2\text{OOH}$ (for Case II). The latter peroxy undergoes a 1,4 H-shift to $\text{CO} + \text{OH} + \text{CH}_3\text{C(O)CH}_2\text{OOH}$ (HPACET). Such H-shift being not open for the Case I peroxy radical, it

reacts primarily with NO or $\text{HO}_2$, leading for the most part to $\text{CH}_3\text{C(O)CH(O}^\circ)\text{CH}_2\text{OOH}$ that promptly decomposes into either $\text{CH}_3\text{C(O)} + \text{OCHCH}_2\text{OOH}$ (HPAC), or $\text{HCHO} + \text{OH} + \text{MGLY}$. Photolysis of HPCE can be expected to proceed by splitting off the formyl radical, leading to the same peroxy radicals as above.

### 2.1.3   Traditional chemistry of the initial $\delta$-OH peroxy radicals

The reactions of ISOPBO2 and ISOPDO2 with NO and $\text{HO}_2$ generate a mixture of $\beta$- and $\delta$-OH-peroxy reaction products.

The share of the $\delta$-OH-peroxy reaction products is small (5% for ISOPBO2 and 6% for ISOPDO2 at 297 K for a bimolecular peroxy lifetime of 50 s) and assumed here to be constant. The absolute error on product yields due to this assumption does not exceed 0.5% in most atmospherically-relevant conditions ($\text{RO}_2$ lifetime between 10 and 100 s). As MAGRITTE is not intended to model local urban conditions, we omit the minor products of the bimolecular reactions of the $\delta$-hydroxyperoxy radicals, such as their reactions with other peroxy radicals. The hydroperoxides formed from their reactions with $\text{HO}_2$ are

lumped with the $\beta$-OH-counterparts, as are also the further-generation $\delta$-OH-epoxides. Besides nitrate formation, the reactions with NO form $Z$- and $E$-$\delta$-OH-allyloxy radicals that were shown (Nguyen and Peeters, 2015) to interconvert rapidly and to react both in the $Z$-form by a fast $\alpha$-hydroxy-H shift that leaves the products activated by a total of 32 kcal mol$^{-1}$; this allows rotation of the OH in the hydroxy-allyl group over the barrier of $\sim$12 kcal mol$^{-1}$ (Peeters et al., 2014) and therefore dominant formation of the more stable H-bonded $Z, Z'$ form of the $\delta$-di-OH-allylic radicals, $\text{HOC}^\circ\text{HC(CH}_3)=\text{CHCH}_2\text{OH}$

and $\text{HOC}^\circ\text{HCH=C(CH}_3)\text{CH}_2\text{OH}$. $\alpha$-Addition of $\text{O}_2$, for 45% (Teng et al., 2017), results in $\text{C}_5$ hydroxyaldehydes HALD1 and HALD2 (4,1- and 1,4-HC5 in Wennberg et al. (2018), HALD1 and HALD2 in the MCM) + $\text{HO}_2$. $\gamma$-Addition of $\text{O}_2$ (for 55%) result in $Z, Z'$-enol-peroxys which were shown (Peeters and Nguyen, 2012) to undergo very fast 1,6 enol-H-shifts leading to next-generation peroxys that can isomerize by 1,4 aldehyde-H shifts facing a barrier of only 20.2 kcal mol$^{-1}$; indeed, for 1,4 aldehyde-H-shifts in similar hydroperoxy-formyl-peroxys with barriers of 20.6-21.2 kcal mol$^{-1}$, rates of $\sim$1.5 s$^{-1}$ were

calculated and the products were shown to quickly lose CO and OH (Liu et al., 2017). Here, the expected products are $\text{OH} + \text{CO} + \text{CH}_3\text{C(O)CH(OOH)CH}_2\text{OH}$ or $\text{OCHC(CH}_3)(\text{OOH})\text{CH}_2\text{OH}$. At very high NO as in some laboratory conditions, the NO-reaction will dominate and yield either $\text{MGLY} + \text{GLYALD} + \text{OH}$ or $\text{GLY} + \text{HYAC} + \text{OH}$, so explaining these observed first-generation products (Paulot et al., 2009b; Galloway et al., 2011).

### 2.1.4 Hydroperoxycarbonyl photolysis

The isoprene oxidation mechanism generates several hydroperoxycarbonyls. Photolysis is expected to dominate the loss of all $\alpha$-hydroperoxy aldehydes (e.g. HPAC, $O=CHCH_2OOH$) and of several hydroperoxyketones (among which HPACET, $CH_3C(O)CH_2OOH$) due to estimated near-unit quantum yields and to the strong enhancement of the absorption cross sec-
tions caused by the interaction between the hydroperoxy and carbonyl chromophores (Jorand et al., 2000; Liu et al., 2018). The expected likely major pathway in the photolysis of 2-hydroperoxy-propanal was theoretically determined to be a 1,5 H-shift in the S1 state leading to enol formation (along with triplet $O_2$), at an estimated yield of 84%, whereas intersystem crossing (ISC) resulting in C–C scission (i.e. formyl release) and OH expulsion, makes up the rest (Liu et al., 2018). Similar yields are expected (and adopted here) for e.g. HPAC and HPACET. However, the enol yield should be lower for heavier
compounds due to expected faster ISC rates. It is taken to be 50% for e.g. $CH_3C(O)CH(OOH)CH=O$ (HPKETAL) and $O=CHC(OOH)(CH_3)CH=O$ (HPDIAL). Furthermore, when H-bonding between the carbonyl-O and the hydroperoxide-H supposed to undergo the H-shift leading to enol formation is not favoured, e.g. because of possible H-bonds of this hydrogen with another oxygen in the molecule, enol formation is disadvantaged and therefore neglected here for simplicity. In those cases, formyl or acetyl loss, followed by OH expulsion, is taken to be the only photolysis channel. Note that, to limit the
number of compounds and reactions in the mechanism, several hydroperoxycarbonyls are not considered explicitly, and are replaced by their estimated photolysis products.

The theoretical investigation of the reaction of OH with vinyl alcohol (VA) (So et al., 2014) and with propenols (Lei et al., 2018) is the basis for our evaluation of OH-reactions with enols. OH-addition generally follows e.g.

$$RCH=CHOH + OH(+O_2) \rightarrow RCH(O_2)CH(OH)_2 \xrightarrow{1,5\text{ H-shift}} HC(O)OH + OH + RCHO$$
$$\rightarrow RCH(OH)CH(OH)O_2 \rightarrow HO_2 + RCH(OH)CHO$$

In the case of vinyl alcohol (generated in HPAC photolysis), the formic acid yield is ca. 60% according to So et al. (2014). Acetic acid is similarly formed from the OH-reaction of 2-propenol generated in the photolysis of hydroperoxyacetone (Lei et al.,
2018). $HC(O)OH$ should also be formed in the OH-reaction of hydroxyvinylmethylketone (HMVK, $HOCH=CHC(O)CH_3$) and hydroxymethacrolein (HMAC, $O=CHC(CH_3)=CHOH$), although at a lower yield due to the competition with other possible reactions. Note that the acid-catalysed tautomerization of enols is neglected, based on the theoretical study of the case of vinyl alcohol (Peeters et al., 2015).

### 2.1.5 HPALD photolysis

The HPALD photolysis quantum yield is taken equal to 0.8, a compromise between the experimental value of $1\pm0.4$ for a $C_6$ HPALD proxy (Wolfe et al., 2012) and the theoretical value (actually a lower limit) of 0.55 by Liu et al. (2017). The mechanism following HPALD photolysis is based on the theoretical study of Liu et al. (2017):

$$\text{HPALD1} + h\nu \rightarrow \text{OH} + 0.11\,(\text{HO}_2 + \text{O=CHCH=C(CH}_3)\text{CH=O (MBED)})$$
$$+0.11\,(\text{CO} + \text{OH} + \text{O=CHCH(OOH)C(O)CH}_3\,(\text{HPKETAL}))$$
$$+0.56\,(\text{CO} + \text{OH} + \text{O=CHCH=C(CH}_3)(\text{OH})\,(\text{HMVK}))$$
$$+0.22\,(\text{CO} + \text{CH}_3\text{C(O}_2)\text{=CHCH}_2\text{OH}^\dagger\,(\text{V1O2}^\dagger))$$

$$\text{HPALD2} + h\nu \rightarrow \text{OH} + 0.18\,(\text{HO}_2 + \text{O=CHCH=C(CH}_3)\text{CH=O (MBED)})$$
$$+0.18\,(\text{CO} + \text{OH} + \text{O=CHCH(OOH)C(O)CH}_3\,(\text{HPKETAL}))$$
$$+0.46\,(\text{CO} + \text{OH} + \text{O=CHC(CH}_3)\text{=CHOH}\,(\text{HMAC}))$$
$$+0.18\,(\text{CO} + \text{HOCH}_2\text{C(CH}_3)\text{=CHO}_2^\dagger\,(\text{V2O2}^\dagger))$$

Note that the formation of $\text{OCHC(CH}_3)(\text{OOH})\text{CHO}$, considered in Wennberg et al. (2018) besides HPKETAL formation in the second photolysis channel of each HPALD, is neglected here as it was found to be minor (Liu et al., 2017).

Based on a reaction chamber study of butenedial and 4-oxo-2-pentenal photolysis (Thuner et al., 2003), the photolysis of methylbutenedial (MBED) should be very fast (lifetime of minutes) and lead to a furanone-type compound as major product, as well as methylmaleic anhydride (MMAL) and other compounds. Relying on MCM for the further oxidation of the furanone by OH, we replace MBED by its assumed photooxidation products:

$$\text{MBED} \xrightarrow{\text{fast}} 0.55\,(-\text{OH} + 2\text{CO}_2 + \text{HCHO} + \text{CH}_3\text{CO}_3)$$
$$+0.20\,\text{MMAL} + 0.15\,(\text{MGLY} + \text{CO} + \text{HO}_2 + \text{CO}_2) + 0.10\,(\text{GLY} + \text{CH}_3\text{CO}_3 + \text{CO}_2)$$

The major sink of the enols HMAC and HMVK should be their reaction with OH, leading in part to formic acid formation (see Table 2). Based on the experimental study of Yoon et al. (1999), photolysis of the analogous ketone-enol form of acetylacetone $(\text{CH}_3\text{C(O)CH=C(OH)CH}_3)$ yields OH and a vinylic co-product radical up to a wavelength of 312 nm, with an OH appearance rate of $10^8$ s$^{-1}$ or higher around 300 nm, implying a quantum yield at atmospheric pressure of order 0.1 (instead of a near-unit quantum yield as assumed by Liu et al. (2017)). The absorption cross sections of the enols are obtained from the acetylacetone study of Nakanishi et al. (1977). By analogy with the $\text{CH}_2\text{=CH}^\circ + \text{O}_2$ reaction (Mebel and Kislov, 2005), we assume that the vinylic co-product radicals of HMAC and HMVK photolysis react rapidly with $\text{O}_2$ to give HCO + MGLY and $\text{CH}_3\text{CO}$ + GLY, respectively.

The activated vinylperoxy radicals $\text{V1O2}^\dagger$ and $\text{V2O2}^\dagger$ from HPALD photolysis might be stabilized by collisions and undergo reactions with NO, $\text{HO}_2$ and $\text{NO}_2$ (Liu et al., 2017), but a more probable fate is decomposition (Mebel and Kislov, 2005), to $\text{CH}_3\text{CO}$ + GLYALD in the case of V1O2, and HCO + HYAC, in the case of V2O2.

## 2.2 Isoprene + $O_3$

The ozonolysis mechanism follows the experimentally-derived model of Nguyen et al. (2016), except regarding the fate of the Criegee intermediate $\text{CH}_2\text{OO}$, formed with a yield of 58% (and assumed to be entirely stabilized). Whereas Nguyen et al.

attributed a significant role to the reaction of $CH_2OO$ with the water monomer, motivated by the dependence of the observed yields on relative humidity, the reaction of $CH_2OO$ with the water dimer has been shown by several groups to be largely dominant at all relevant conditions (Berndt et al., 2014; Chao et al., 2015; Smith et al., 2015; Lewis et al., 2015; Sheps et al., 2017) and is therefore the only reaction considered here. More work is needed to elucidate the humidity dependence of the yields. Reaction with the dimer follows the recent study of Sheps et al. (2017):

$$CH_2OO + (H_2O)_2 \rightarrow 0.55\,(HOCH_2OOH + H_2O)$$
$$+0.4\,(HCHO + H_2O_2 + H_2O)$$
$$+0.05\,(HC(O)OH + 2\,H_2O)$$

## 2.3 Isoprene + $NO_3$

The mechanism for $NO_3$-initiated oxidation follows largely the laboratory study of Schwantes et al. (2015). Several minor pathways are neglected, however, as the further degradation mechanism of several products remain unclear. The title reaction, followed by $O_2$–addition, forms several peroxy radical isomers lumped into one compound (NISOPO2). Generalizing the mechanism proposed by Schwantes et al., the reaction of NISOPO2 with non-tertiary peroxy radicals proceeds following

$$NISOPO2 + RO_2 \rightarrow 0.2\,(NISOPO + RO + O_2)$$
$$+0.4\,(0.88\,NC4CHO + 0.12\,MACR + 0.12\,HCHO + 0.12\,NO_2 + ROH)$$
$$+0.4\,(0.74\,ISOPCNO3 + 0.14\,ISOPANO3 + 0.12\,ISOPDNO3 + R'CHO)$$

whereas for tertiary peroxy radicals the reaction reads

$$NISOPO2 + RO_2 \rightarrow 0.5\,(NISOPO + RO + O_2)$$
$$+0.5\,(0.88\,NC4CHO + 0.12\,MACR + 0.12\,HCHO + 0.12\,NO_2 + ROH)$$

The proposed 1,6 H-shift of the trans-[1,4] isomer of NISOPO2 radicals (Schwantes et al., 2015) is neglected, as it is slow ($4 \cdot 10^{-4}\,s^{-1}$) compared to the other reactions. The different isomers of the oxy radical NISOPO have different fates: decomposition to MVK or MACR (for the $\beta$-nitroxy oxys), reaction with $O_2$ (for the $\delta$'s), and a fast 1,5 H-shift (Kwan et al., 2012) (ca. $2 \cdot 10^5\,s^{-1}$) for the $\delta$-(1-$ONO_2$,4-O) radical, outrunning the $O_2$-reaction by a factor of about 4. The isomerisation leads, after $O_2$-addition, to a peroxy of which the reaction with NO or $NO_3$ forms an enal nitrate, $O_2NOCH_2C(=CH_2)CH=O$, along with HCHO and $HO_2$ (Wennberg et al., 2018). The main expected fate of this enal nitrate is photolysis, to $NO_2$ + HCHO + $O=CH-C(=CH_2)O_2$. The latter radical can undergo a fast 1,4 H-shift to give CO + OH + $H_2C=C=O$ (ketene). Ketene can react with OH, at a rate of ca. $1.7 \cdot 10^{-11}\,molec^{-1}\,cm^3\,s^{-1}$, producing CO + $^\circ CH_2OH$ (Calvert et al., 2011); it also photolyzes to $^1CH_2$ (or $^3CH_2$) + CO. The fate of methylene is mainly oxidation to CO or $CO_2$ (Baulch et al., 2005). Based on photolysis parameter data provided by Calvert et al. (2011), photolysis is estimated to be slightly less important than the OH-reaction, and is therefore neglected here for simplicity.

Based on the above, the lumped oxy radical undergoes the condensed fast reaction

$$\text{NISOPO} \rightarrow 0.42\,\text{MVK} + 0.04\,\text{MACR} + 1.54\,\text{HCHO} + 0.82\,\text{NO}_2 + 0.18\,\text{NC4CHO} + 0.9\,\text{HO}_2 + 0.72\,\text{CO}$$

The $\beta$- and $\delta$-nitroxy hydroperoxides formed in the $\text{NISOPO2} + \text{HO}_2$ reaction are explicitly considered. Their reactions with OH form nitroxy hydroxy epoxides (IHNE) as well as hydroperoxy and nitroxy carbonyls, also explicitly considered in the mechanism. A major product of the NISOPO2 reaction with NO or $\text{RO}_2$ is the enal nitrate NC4CHO. Laboratory work on an analogous compound (Xiong et al., 2016) has shown that photolysis is by far its dominant sink, owing to high quantum yields and to enhanced absorption cross sections attributed to the interaction of the nitrate and carbonyl chromophore. The NC4CHO photolysis cross sections and quantum yield recommendation follow Xiong et al. (2016). As the mechanism and products are uncertain (Xiong et al., 2016), we tentatively adopt a similar photolysis mechanism as for the analogous HPALDs, but with $\text{O}--\text{NO}_2$ bond scission substituted for $\text{O}--\text{OH}$ scission. (see above, Sect. 2.1).

## 2.4  Monoterpene oxidation

Due to the complexity and poor understanding of monoterpene oxidation, we adopt a simple parameterization based on box model simulations of $\alpha$- and $\beta$-pinene oxidation using the MCMv3.2 (Saunders et al., 2003). The scope of the parameterization is limited to the reproduction of total yields of several key products; those yields reflect not only primary production but also secondary formation. The influence of monoterpenes on radicals (e.g. HOx, $\text{RO}_2$) and on ozone production is therefore likely not well represented by this simple mechanism. It should be stressed that even the monoterpene mechanism in MCM is greatly oversimplified, as it neglects many possibly important pathways (in particular H-shift isomerisations in peroxy radicals), with potentially very large effects on radicals and other products. A thorough evaluation of mechanisms against laboratory data will be needed in order to assess their uncertainties, but is out of scope of the present study.

The parameterization relies on sixty-day simulations performed using the Kinetic PreProcessor (KPP) package (Damian et al., 2002). The photolysis rates are calculated for clear-sky conditions at 30°N on July 15th. Although both high-NOx (1 ppbv $\text{NO}_x$, 40 ppbv $\text{O}_3$ and 250 ppbv CO maintained throughout the simulation) and low-NOx simulations (100 pptv $\text{NO}_x$, 20 ppbv $\text{O}_3$ and 150 ppbv CO) are conducted, only the low-NOx results are used for the parameterization. Temperature and $\text{H}_2\text{O}$ are kept at 298 K and 1% v/v. To determine the product yields, counter compounds are introduced in the equation file (e.g. HCHOa, MGLYOXa, etc.) having the same production terms as the species they represent, but without any chemical loss.

The yield of acetone from both $\alpha$- and $\beta-$pinene is very close to 100% after several days of reaction, independent of the NOx level. The yield of methylglyoxal is low (4% and 5% for $\alpha$- and $\beta$-pinene, not counting the contribution of acetone oxidation by OH). The overall yield of formaldehyde obtained in these simulations is ~4.2 HCHO per monoterpene oxidized, almost independent of $\text{NO}_x$, for both precursors. The HCHO yield comes down to 2.3 after subtracting the contributions of acetone and methylglyoxal oxidation. This yield is further reduced by 45% to account for wet/dry deposition of intermediates and secondary organic aerosol formation. That fraction is higher, but of the same order, as the estimated overall impact of deposition on the average formaldehyde yield from isoprene oxidation (~30%), based on global model (MAGRITTE) calculations. The higher fraction is justified by the larger number of oxidation steps and the generally lower volatility of intermediates involved in formaldehyde formation from monoterpene oxidation. Nevertheless, this adjustment introduces a significant uncertainty in the

model results. A sensitivity calculation shows that adopting a lower yield reduction (20% instead of 45%) in the global model (Sect. 4.1) has negligible impact on the calculated HCHO abundances ($<\sim$1%) in most regions, but leads to higher HCHO vertical columns in monoterpene emission regions, by $\sim$5% over Amazonia and by up to 8% over Siberia. The associated impact on OH reaches +2% in those regions, due to the additional $HO_x$ formation through HCHO photolysis.

The overall carbon balance of monoterpene oxidation in the mechanism is $\sim$50% due to the combined effects of deposition, SOA formation and CO and $CO_2$ formation besides their production through the degradation of the explicit products.

## 2.5   Cross-reactions of peroxy radicals

The channel ratios and rates of the cross reactions of peroxy radicals generally follow Capouet et al. (2004), except for the peroxy radicals from ISOP+OH, for which we follow the recommendations of Wennberg et al. (2018) (based on measurements
from Jenkin et al. (1998)) and ISOP+$NO_3$, based on Wennberg et al. (2018) and Schwantes et al. (2015). The cross reaction rates are calculated as twice the geometric mean of the self-reaction rates, except for acylperoxy radicals for which the rate and channel data reported for $CH_3CO_3$ are used (Atkinson et al., 2006). The self-reaction rates are obtained from compiled data for similar compounds (Capouet et al., 2004; Peeters and Müller, 2010; Atkinson et al., 2006).

## 2.6   Peroxy radical reactions with NO and $HO_2$

We adopt the recommendations of Wennberg et al. (2018) for the rates of non-acyl peroxy radical reactions with NO ($2.7 \cdot 10^{-12} \exp(350/T)$ cm$^3$ molec$^{-1}$ s$^{-1}$) as well as with $HO_2$ ($2.82 \cdot 10^{-13} \exp(1300/T) \cdot [1 - \exp(-0.231\,n)]$ cm$^3$ molec$^{-1}$ s$^{-1}$, with $n$ the number of heavy atoms in the radical, excluding the peroxy moiety).

   We also follow Wennberg et al. (2018) for estimating the nitrate yield in the reactions of organic peroxys with NO. The parameterization is based on the temperature- and pressure-dependent expressions proposed by Carter and Atkinson (1989)
and by Arey et al. (2001), modified to account for the recommendation by Teng et al. (2015) to relate this yield to the number ($n$) of heavy atoms in the peroxy radical, excluding the peroxy moiety. The branching ratios of the nitrate pathway ($Y_{\mathrm{nit}}$) and for the oxy radical pathway ($Y_{\mathrm{oxy}}$) are given by

$$Y_{\mathrm{nit}}(T, M, n, Z) = \frac{A(T, M, n)}{A(T, M, n) + Z} \tag{3}$$

$$Y_{\mathrm{oxy}}(T, M, n, Z) = 1 - Y_{\mathrm{nit}}(T, M, n, Z) \tag{4}$$

with

$$A(T, M, n) = \frac{k_0[M]}{1 + k_0[M]/k_\infty} \cdot 0.41^{\{1 + [\log_{10}(k_0[M]/k_\infty)]^2\}^{-1}} \tag{5}$$

$$k_0 = \alpha \cdot \mathrm{e}^n \tag{6}$$

$$k_\infty = 0.43 \cdot (T/298)^{-8} \tag{7}$$

where $\alpha = 2 \cdot 10^{-22}$ cm$^3$ molec$^{-1}$. $Z$ is a normalization term adjusted in order to match experimental determinations of the branching ratio, when available. In absence of such constraint, it is calculated (for $n > 2$) using

$$Z = A_0(n) \frac{1 - \alpha_0}{\alpha_0}, \tag{8}$$

with $A_0(n) = A(T{=}293$ K, $M{=}2.45 \cdot 10^{19}$ molec. cm$^{-3}$, $n)$ and

$$\alpha_0 = 0.045 \cdot n - 0.11 \tag{9}$$

The nitrate yield is further modified according to molecular structure as recommended in Wennberg et al. (2018). The dependence of the yields on atmospheric pressure is shown in Fig. 2 for January and July at mid-latitudes. For small values of $n$ (especially $n = 1$), $Y_{\mathrm{nit}}$ decreases with altitude. For large values of $n$ (e.g. $n = 11$), the yield increases with altitude due to the strong temperature dependence of the high-pressure limit (Eq. 7).

## 2.7 $\mathbf{CH_3O_2 + OH}$

Methylperoxy radical ($CH_3O_2$) was shown to react rapidly with OH (Bossolasco et al., 2014) although two more recent experimental studies inferred a lower rate constant (Yan et al., 2016; Assaf et al., 2016). The possible pathways include

$$CH_3O_2 + OH \xrightarrow{a} CH_3O + HO_2$$
$$\xrightarrow{b} CH_3OH + O_2$$
$$\xrightarrow{c} CH_2O_2 + H_2O$$
$$\xrightarrow{d} CH_3OOOH$$

The stabilized trioxide ($CH_3OOOH$) formed in channel d has several possible fates, among which reaction with OH and uptake by aqueous aerosols followed by decomposition into $CH_3OH + O_2$ are expected to be the most important (Müller et al., 2016). An upper limit of 5% for the yield of Criegee radicals was also determined by Assaf et al. (2017), in agreement with the theoretical expectation that it should be negligible (Müller et al., 2016). A yield of 0.9±0.1 for the methoxy + HO$_2$ channel was determined experimentally at low pressure (50 Torr) (Assaf et al., 2018), in good agreement with the best theoretical estimate (0.92, range 0.77–0.97) determined in Müller et al. (2016) and used in our mechanism. It is also consistent with the methanol yield measurements reported recently by Caravan et al. (2018) at both low and high pressure (0.06±0.02 at 740 Torr). Those results imply however a methanol yield much lower than the value (0.23) used in our global model to reconcile its predictions with atmospheric methanol observations at remote locations (Müller et al., 2016). Note that at low pressure (as used in the experiments by Assaf et al. (2017) and Assaf et al. (2018)), stabilisation of the trioxide is negligible, given the quadratic dependence of the stabilisation fraction ($f_{\mathrm{stab}}$) on atmospheric pressure (Müller et al., 2016),

$$f_{\mathrm{stab}} = f_0 \cdot p^2 \cdot (T/298)^{-5}, \tag{10}$$

where $p$ is atmospheric pressure (atm) and T is temperature (K). In the lower troposphere, however, stabilisation is significant, with a best theoretical estimate of $f_0 = 0.107$. Significant experimental evidence for this partial stabilisation was found by Caravan et al. (2018) at 740 Torr (but not at low pressure).

The mechanism does not account for the possible reaction of OH with other peroxy radicals. As noted by Müller et al. (2016), its relevance for larger peroxys (such as those formed in the oxidation of biogenic VOCs) is expected to be lower than in the case of $CH_3O_2$. Furthermore, the fate of the stabilised trioxide formed at high yield (Müller et al., 2016; Assaf et al., 2018) in the reaction of large $RO_2$ radicals with OH is so far unexplored.

## 2.8 Model species and chemical mechanism

**Table 1.** Chemical species of the oxidation mechanism of isoprene, monoterpenes and methylbutenol (MBO).

| Notation | Chemical formula |
|---|---|
| $C_1$ *compounds* | |
| HCHO | HCHO |
| CO | CO |
| $CH_3OH$ | $CH_3OH$ |
| HCOOH | HC(O)OH |
| $CH_3OOH$ | $CH_3OOH$ |
| $CH_3OOOH$ | $CH_3OOOH$ |
| $CH_3ONO_2$ | $CH_3ONO_2$ |
| HMHP | $HOCH_2OOH$ |
| $C_2$ *compounds* | |
| $CH_3CHO$ | $CH_3CHO$ |
| GLYALD | $HOCH_2CHO$ |
| GLY | CHOCHO |
| $C_2H_5OH$ | $C_2H_5OH$ |
| $CH_3COOH$ | $CH_3C(O)OH$ |
| PAA | $CH_3C(O)OOH$ |
| GPA | OCHC(O)OOH |
| ETHLN | $OCHCH_2ONO_2$ |
| HPAC | $OCHCH_2OOH$ |
| GCO3H | $HOCH_2C(O)OOH$ |
| GCOOH | $HOCH_2C(O)OH$ |
| PAN | $CH_3C(O)OONO_2$ |
| GPAN | $HOCH_2C(O)OONO_2$ |
| VA | $CH_2{=}CHOH$ |
| $C_3$ *compounds* | |
| $CH_3COCH_3$ | $CH_3C(O)CH_3$ |
| HYAC | $CH_2OHC(O)CH_3$ |
| MGLY | $CH_3C(O)CHO$ |
| $C_2H_5COOH$ | $CH_3CH_2C(O)OH$ |

| Notation | Chemical formula |
| --- | --- |
| NOA | $CH_3C(O)CH_2ONO_2$ |
| HPACET | $CH_3C(O)CH_2OOH$ |
| MVA | $CH_2{=}C(CH_3)OH$ |
| DHA | $CH_3C(O)CH(OH)_2$ |
| | *$C_4$ compounds* |
| MACR | $CH_2{=}C(CH_3)CHO$ |
| MVK | $CH_2{=}CHC(O)CH_3$ |
| MPAN | $CH_2{=}C(CH_3)C(O)OONO_2$ |
| MCO3H | $CH_2{=}C(CH_3)C(O)OOH$ |
| MCOOH | $CH_2{=}C(CH_3)C(O)OH$ |
| MVKOOH | $0.55\,CH_3C(O)CH(OOH)CH_2OH + 0.45\,CH_3C(O)CH(OH)CH_2OOH$ |
| MACRNO3 | $OCHC(CH_3)(ONO_2)CH_2OH$ |
| MVKNO3 | $0.2\,CH_3C(O)CH(OH)CH_2ONO_2 + 0.8\,CH_3C(O)CH(ONO_2)CH_2OH$ |
| MACROH | $HOCH_2C(CH_3)(OH)CHO$ |
| BIACETOH | $CH_3C(O)C(O)CH_2OH$ |
| DHBO | $CH_3C(O)CH(OH)CH_2OH$ |
| HOBA | $CH_3C(O)CH(OH)CHO$ |
| DIHPMEK | $CH_3C(O)CH(OOH)CH_2OOH$ |
| HPKETAL | $CH_3C(O)CH(OOH)CHO$ |
| HPDIAL | $OCHC(CH_3)(OOH)CHO$ |
| HMVK | $CH_3C(O)CH{=}CHOH$ |
| HMAC | $OCHC(CH_3){=}CHOH$ |
| HMML | $HOCH_2\overline{C(CH_3)OC}{=}O$ |
| | *$C_5$ compounds* |
| ISOP | $CH_2{=}C(CH_3)CH{=}CH_2$ |
| MBO | $CH_3C(OH)(CH_3)CH{=}CH_2$ |
| HCOC5 | $CH_2{=}C(CH_3)C(O)CH_2OH$ |
| ISOPBOOH | $0.95\,CH_2{=}CHC(CH_3)(OOH)CH_2OH + 0.05\,OHCH_2C(CH_3){=}CHCH_2OOH$ |
| ISOPDOOH | $0.94\,CH_2{=}C(CH_3)CH(OOH)CH_2OH + 0.06\,OHCH_2CH{=}C(CH_3)CH_2OOH$ |
| ISOPEOOH | $CH_2{=}C(CH_3)CH(OH)CH_2OOH$ |
| INDOOH | $HOCH_2CH(ONO_2)C(CH_3)(OOH)CH_2OH$ |

| Notation | Chemical formula |
| --- | --- |
| ISOPANO3 | $HOCH_2C(CH_3)=CHCH_2ONO_2$ |
| ISOPBNO3 | $CH_2=CHC(CH_3)(ONO_2)CH_2OH$ |
| ISOPCNO3 | $HOCH_2CH=C(CH_3)CH_2ONO_2$ |
| ISOPDNO3 | $CH_2=C(CH_3)CH(ONO_2)CH_2OH$ |
| ISOPENO3 | $CH_3C(=CH_2)CH(OH)CH_2ONO_2$ |
| MBONO3 | $0.67\,CH_3C(OH)(CH_3)CH(ONO_2)CH_2OH + 0.33\,CH_3C(OH)(CH_3)CH(OH)CH_2ONO_2$ |
| INCCO | $HOCH_2C(O)C(CH_3)(OH)CH_2ONO_2$ |
| INCNO3 | $HOCH_2CH(ONO_2)C(CH_3)(OH)CH_2ONO_2$ |
| NISOPOOHB | $0.9\,CH_2=CHC(CH_3)(OOH)CH_2ONO_2 + 0.1\,CH_2=C(CH_3)CH(OOH)CH_2ONO_2$ |
| NISOPOOHD | $0.84\,HOOCH_2CH=C(CH_3)CH_2ONO_2 + 0.26\,O_2NOCH_2CH=C(CH_3)CH_2OOH$ |
| IEPOX | $HOCH_2\overline{CHOC}(CH_3)CH_2OH$ |
| ICHE | $HOCH_2\overline{CHOC}(CH_3)CHO$ and 3 isomers |
| HPCE | $0.18\ HOOCH_2\overline{CHOC}(CH_3)CHO + 0.82\ OCH\overline{CHOC}(CH_3)CH_2OOH$ |
| DHHEPOX | $HOCH_2C(CH_3)(OOH)\overline{CHOCH}(OH)$ |
| NC4CHO | $0.75\,OCHCH=C(CH_3)CH_2ONO_2 + 0.25\,OCHC(CH_3)=CHCH_2ONO_2$ |
| ISOPBOH | $CH_2=CHC(CH_3)(OH)CH_2OH$ |
| ISOPDOH | $CH_2=C(CH_3)CH(OH)CH_2OH$ |
| HALD1 | $OCHC(CH_3)=CHCH_2OH$ |
| HALD2 | $OCHCH=C(CH_3)CH_2OH$ |
| HPALD1 | $OCHC(CH_3)=CHCH_2(OOH)$ |
| HPALD2 | $OCHCH=C(CH_3)CH_2(OOH)$ |
| MMAL | $O=\overline{CCH}=C(CH_3)C(=O)\overline{O}$ |
| IHNE | $0.57\,O_2NOCH_2\overline{C(CH_3)OCH}CH_2OH + 0.25\,O_2NOCH_2C(CH_3)(OH)\overline{CHOCH_2}$ and isomers |
| | *$C_{10}$ compounds* |
| APIN | $C_{10}H_{16}$ (sum of monoterpenes) |
| APINONO2 | $C_{10}H_{16}(OH)(ONO_2)$ |
| | *Peroxy radicals* |
| $CH_3O_2$ | $CH_3O_2$ |
| $CH_3CO_3$ | $CH_3C(O)O_2$ |
| $OCHCH_2O_2$ | $OCHCH_2O_2$ |
| $HOCH_2CH_2O_2$ | $HOCH_2CH_2O_2$ |

| Notation | Chemical formula |
| --- | --- |
| GCO3 | $HOCH_2C(O)O_2$ |
| QO2 | $HOCH_2CH_2O_2$ |
| ACETO2 | $CH_3COCH_2O_2$ |
| MVKO2 | $0.75\,CH_3C(O)CH(O_2)CH_2OH + 0.25\,CH_3C(O)CH(OH)CH_2O_2$ |
| MCO3 | $CH_2{=}C(CH_3)C(O)O_2$ |
| ISOPBO2 | $0.95\,HOCH_2C(CH_3)(O_2)CH{=}CH_2 + 0.05\,OHCH_2C(CH_3){=}CHCH_2O_2$ |
| ISOPDO2 | $0.94\,CH_2{=}C(CH_3)CH(O_2)CH_2OH + 0.06\,OHCH_2CH{=}C(CH_3)CH_2O_2$ |
| ISOPEO2 | $CH_3C({=}CH_2)CH(OH)CH_2O_2$ |
| DIHPCARP1 | $CH_3C(OO)(CHO)CH(OOH)CH_2OOH$ |
| DIHPCARP2 | $OCHCH(OO)C(CH_3)(OOH)CH_2OOH$ |
| DHPAO2 | $HOOCH_2C(CH_3)(O_2)CH(OOH)C(O)OOH$ |
| KPO2 | $0.5\,CH3C(O)CH(O_2)CH_2OOH + 0.5\,CH3C(O)CH(OOH)CH_2O_2$ |
| IEPOXAO2 | $HOCH_2CH(OH)C(CH_3)(O_2)CHO$ |
| IEPOXBO2 | $HOCH_2C(OH)(CH_3)CH(O_2)CHO$ |
| C59O2 | $HOCH_2C(CH_3)(O_2)C(O)CH_2OH$ |
| INAO2 | $0.73\,HOCH_2C(O_2)(CH_3)CH(OH)CH_2ONO_2 + 0.27\,HOCH_2C(OH)(CH_3)CH(O_2)CH_2ONO_2$ |
| INBO2 | $0.85\,HOCH_2CH(O_2)C(CH_3)(ONO_2)CH_2OH + 0.15\,O_2CH_2CH(OH)C(CH_3)(ONO_2)CH_2OH$ |
| INCO2 | $0.67\,HOCH_2CH(OH)C(O_2)(CH_3)CH_2ONO_2 + 0.33\,HOCH_2CH(O_2)C(OH)(CH_3)CH_2ONO_2$ |
| INDO2 | $HOCH_2CH(ONO_2)C(CH_3)(O_2)CH_2OH$ |
| NISOPO2 | $0.45\,O_2CH_2CH{=}C(CH_3)CH_2ONO_2 + 0.42\,CH_2{=}CHC(CH_3)(O_2)CH_2ONO_2 +$ |
|  | $0.085\,O_2NOCH_2CH{=}C(CH_3)CH_2O_2 + 0.045\,CH_2{=}C(CH_3)CH(O_2)CH_2ONO_2$ |
| MBOO2 | $0.67\,CH_3C(OH)(CH_3)CH(O_2)CH_2OH + 0.33\,CH_3C(OH)(CH_3)CH(OH)CH_2O_2$ |
| APINOHO2 | peroxy radical from $APIN + OH$ |
| APINO3O2 | peroxy radical from $APIN + O_3$ |

**Table 2.** Chemical mechanism and rates. Units for $1^{st}$-, $2^{nd}$-, and $3^d$-order reactions are $s^{-1}$, $cm^3molec.^{-1}s^{-1}$ and $cm^6molec.^{-2}s^{-1}$. Read $2.7(-11)$ as $2.7 \cdot 10^{-11}$; $T$=temperature (K); $[M]$ is air density (molec.cm$^{-3}$); $K_{RO2NO} = 2.7(-12)\exp(350/T)$; the PAN-like compounds formation and decomposition rates are calculated with $k = \frac{k_0[M]}{1+k_0[M]/k_\infty}0.3^{\{1+[\log_{10}(k_0[M]/k_\infty)/1.414]^2\}^{-1}}$. References: 1, MCM (Saunders et al., 2003; Jenkin et al., 2015); 2, Nguyen et al. (2016); 3, Wennberg et al. (2018); 4, Liu et al. (2013); 5, Peeters and Müller (2010); 6, Capouet et al. (2004); 7, Atkinson et al. (2006); 8, Peeters et al. (2014); 9, St. Clair et al. (2016); 10, D'Ambro et al. (2017); 11, Lee et al. (2014); 12, Jacobs et al. (2014); 13, Paulot et al. (2009b); 14, Bates et al. (2016); 15, Schwantes et al. (2015); 16, Xiong et al. (2016); 17, Crounse et al. (2012); 18, Gross et al. (2014); 19, Burkholder et al. (2015); 20, Nguyen et al. (2015a); 21, Galloway et al. (2011); 22, Praske et al. (2015); 23, Vu et al. (2013); 24, Baeza-Romero et al. (2007); 25, Magneron et al. (2005); 26, Taraborrelli et al. (2012); 28, So et al. (2014); 29, Assaf et al. (2016); 30, Assaf et al. (2018); 31, Müller et al. (2016); 32, Allen et al. (2018); 34, Chan et al. (2009).

| Reaction | Rate | Ref. | Note |
|---|---|---|---|
| *C$_5$ compounds* | | | |
| $ISOP + OH \rightarrow 0.586\,ISOPBO2 + 0.344\,ISOPDO2 + 0.02\,ISOPEO2$ $+ 0.10\,HO_2 + 0.05\,ACETO2 + 0.05\,HCHO + 0.05\,CO_2$ | $2.7(-11)\exp(360/T)$ | | N1 |
| $ISOP + NO_3 \rightarrow NISOPO2$ | $3.15(-12)\exp(-450/T)$ | 1 | |
| $ISOP + O_3 \rightarrow 0.41\,MACR + 0.17\,MVK + 0.86\,HCHO + 0.03\,MCOOH$ $+ 0.3\,CO_2 + 0.3\,HO_2 + 0.1\,CH_3O_2 + 0.24\,CO + 0.05\,CH_3CO_3$ $+ 0.14\,OH + 0.58\,(0.55\,HMHP + 0.4\,HCHO + 0.4\,H_2O_2$ $+ 0.05\,HCOOH)$ | $1.03(-14)\exp(-1995/T)$ | 2 | N2 |
| $ISOPBO2 + NO \rightarrow NO_2 + 0.95\,MVK + 0.95\,HCHO + 0.973\,HO_2$ $+ 0.023\,HALD1 + 0.027\,MVKOOH + 0.027\,CO + 0.027\,OH$ | $K_{RO2NO} \cdot Y_{oxy}(T,M,6,1.19)$ | 3 | N3 |
| $ISOPBO2 + NO \rightarrow 0.96\,ISOPBNO3 + 0.04\,ISOPANO3$ | $K_{RO2NO} \cdot Y_{nit}(T,M,6,1.19)$ | 3 | N3 |
| $ISOPBO2 + NO_3 \rightarrow NO_2 + 0.95\,MVK + 0.95\,HCHO + 0.973\,HO_2$ $+ 0.023\,HALD1 + 0.027\,MVKOOH + 0.027\,CO + 0.027\,OH$ | $2.3(-12)$ | 1,3 | |
| $ISOPBO2 + HO_2 \rightarrow 0.94\,ISOPBOOH + 0.06\,OH$ $+ 0.06\,MVK + 0.06\,HCHO + 0.06\,HO_2$ | $2.1(-13)\exp(1300/T)$ | 1,3,4 | |
| $ISOPBO2 + ISOPBO2 \rightarrow 2\,MVK + 2\,HCHO + 2\,HO_2$ | $6.6(-14)$ | 3 | |
| $ISOPBO2 + ISOPBO2 \rightarrow 0.5\,HO_2 + 0.5\,HALD1 + 0.5\,CO + 0.5\,OH$ $+ 0.5\,MVKOOH$ | $1.1(-13)$ | 3 | |
| $ISOPBO2 + ISOPDO2 \rightarrow 0.9\,MVK + 1.8\,HCHO + 1.8\,HO_2$ $+ 0.1\,ISOPBOH + 0.9\,MACR + 0.1\,HCOC5$ | $3.08(-12)$ | 3 | |
| $ISOPBO2 + CH_3O_2 \rightarrow 0.5\,MVK + 1.5\,HCHO + 0.7\,HO_2$ $+ 0.5\,ISOPBOH$ | $2.0(-12)$ | 3 | |
| $ISOPBO2 + CH_3CO_3 \rightarrow MVK + HCHO + HO_2 + CH_3O_2 + CO_2$ | $1.8(-12)\exp(500/T)$ | 6,7 | |
| $ISOPBO2 \rightarrow 0.75\,HPALD1 + 0.75\,HO_2 + 0.15\,HPCE$ | $3.409(+12)\exp(-10698/T)$ | | N4 |

| Reaction | Rate | Ref. | Note |
|---|---|---|---|
| $+0.25\,OH + 0.1\,CO + 0.1\,CO + 0.1\,DIHPMEK$ | $+\,2.89(-15)\exp(414/T)\cdot[NO]$ | | |
| | $+\,2.26(-16)\exp(1364/T)\cdot[HO_2]$ | | |
| $ISOPBO2 \rightarrow MVK + HCHO + OH$ | $9.9(+10)\exp(-9746/T)$ | 8 | |
| $ISOPBOOH + OH \rightarrow 0.85\,IEPOX + 0.15\,DHHEPOX + OH$ | $1.7(-11)\exp(390/T)$ | 9,3,10 | N6 |
| $ISOPBOOH + OH \rightarrow 0.75\,ISOPBO2 + 0.2\,HCOOH + 0.3\,HO_2$ | $4.6(-12)\exp(200/T)$ | 9,3 | N7 |
| $\quad +0.05\,HCHO + 0.05\,OH + 0.25\,MVK$ | | | |
| $ISOPDO2 + NO \rightarrow NO_2 + 0.94\,MACR + 0.94\,HCHO + HO_2$ | $K_{RO2NO}\cdot Y_{oxy}(T,M,6,1.3)$ | 1,3 | N3 |
| $\quad +0.027\,HALD2 + 0.033\,HYAC + 0.066\,CO + 0.066\,OH$ | | | |
| $ISOPDO2 + NO \rightarrow 0.944\,ISOPDNO3 + 0.056\,ISOPCNO3$ | $K_{RO2NO}\cdot Y_{nit}(T,M,6,1.3)$ | 1,3 | N3 |
| $ISOPDO2 + NO_3 \rightarrow NO_2 + 0.94\,MACR + 0.94\,HCHO + HO_2$ | $2.3(-12)$ | 1 | |
| $\quad +0.027\,HALD2 + 0.037\,HYAC + 0.066\,CO + 0.066\,OH$ | | | |
| $ISOPDO2 + HO_2 \rightarrow 0.941\,ISOPDOOH + 0.059\,OH$ | $2.1(-13)\exp(1300/T)$ | 1,3 | |
| $\quad +0.059\,MACR + 0.059\,HCHO + 0.059\,HO_2$ | | | |
| $ISOPDO2 + ISOPDO2 \rightarrow 1.6\,MACR + 1.6\,HCHO + 1.6\,HO_2$ | $5.74(-12)$ | 3 | |
| $\quad +0.2\,HCOC5 + 0.2\,ISOPDOH$ | | | |
| $ISOPDO2 + CH_3O_2 \rightarrow 0.5\,MACR + 1.25\,HCHO + HO_2$ | $2.0(-12)$ | 3 | |
| $\quad +0.25\,ISOPDOH + 0.25\,HCOC5 + 0.25\,CH_3OH$ | | | |
| $ISOPDO2 + CH_3CO_3 \rightarrow 0.9\,MACR + 0.9\,HCHO + 0.9\,HO_2$ | $2.0(-12)\exp(500/T)$ | 6,7 | |
| $\quad +0.9\,CH_3O_2 + 0.9\,CO_2 + 0.1\,CH_3COOH + 0.1\,HCOC5$ | | | |
| $ISOPDO2 \rightarrow 0.75\,HPALD2 + 0.75\,HO_2 + 0.15\,HPCE$ | $4.253(+8)\exp(-7254/T)$ | | N4 |
| $\quad +0.15\,OH + 0.1\,DHPAO2$ | $+\,6.29(-19)\exp(4012/T)\cdot[NO]$ | | |
| | $+\,4.9(-20)\exp(4962/T)\cdot[HO_2]$ | | |
| $ISOPDO2 \rightarrow MACR + HCHO + OH$ | $1.77(+11)\exp(-9752/T)$ | 8 | |
| $HPCE + OH \rightarrow 1.82\,CO + 0.82\,OH + 0.82\,HPACET + 0.18\,KPO2$ | $2.5(-11)$ | | N5 |
| $KPO2 + NO \rightarrow NO_2 + 0.5\,CH_3CO_3 + 0.5\,HPAC$ | $2.7(-12)\exp(350/T)$ | | N5 |
| $\quad +0.5\,HCHO + 0.5\,OH + 0.5\,MGLY$ | | | |
| $KPO2 + NO_3 \rightarrow NO_2 + 0.5\,CH_3CO_3 + 0.5\,HPAC$ | $2.3(-12)$ | | N5 |
| $\quad +0.5\,HCHO + 0.5\,OH + 0.5\,MGLY$ | | | |
| $KPO2 + HO_2 \rightarrow OH + 0.5\,CH_3CO_3 + 0.5\,HPAC$ | $2.26(-13)\exp(1300/T)$ | | N5 |
| $\quad +0.5\,HCHO + 0.5\,OH + 0.5\,MGLY$ | | | |
| $DHPAO2 + NO \rightarrow NO_2 + HPACET + OH + PGA$ | $2.7(-12)\exp(350/T)$ | | N5 |
| $DHPAO2 + NO_3 \rightarrow NO_2 + HPACET + OH + PGA$ | $2.3(-12)$ | | |
| $DHPAO2 + HO_2 \rightarrow OH + HPACET + OH + PGA$ | $2.64(-13)\exp(1300/T)$ | | N5 |
| $ISOPDOOH + OH \rightarrow 0.85\,IEPOX + 0.15\,DHHEPOX + OH$ | $3.0(-11)\exp(390/T)$ | 9,3,10 | N6 |

| Reaction | Rate | Ref. | Note |
|---|---|---|---|
| ISOPDOOH + OH → 0.6 ISOPDO2 + 0.32 HCOOH + 0.48 HO$_2$ $\quad$ + 0.08 HCHO + 0.08 OH + 0.4 MACR | $4.1(-12)\exp(200/T)$ | 9,3 | N8 |
| ISOPEO2 + NO → MACR + HO$_2$ + HCHO + NO$_2$ | $K_{\text{RO2NO}} \cdot Y_{\text{oxy}}(T, M, 6, 1.27)$ | 1,3 | N3 |
| ISOPEO2 + NO → ISOPENO3 | $K_{\text{RO2NO}} \cdot Y_{\text{nit}}(T, M, 6, 1.27)$ | 1,3 | N3 |
| ISOPEO2 + HO$_2$ → ISOPEOOH | $2.1(-13)\exp(1300/T)$ | 1,3 | |
| ISOPEO2 + ISOPBO2 → 0.7 MVK + 1.4 HCHO + 1.4 HO$_2$ $\quad$ + 0.3 ISOPBOH + 0.7 MACR + 0.3 HCOC5 | $1.2(-12)$ | 5 | |
| ISOPEO2 + ISOPDO2 → MACR + HCHO + HO$_2$ + 0.5 HCOC5 $\quad$ + 0.5 ISOPDOH | $1.1(-11)$ | 5 | |
| ISOPEO2 + ISOPEO2 → MACR + HCHO + HO$_2$ $\quad$ + 0.5 HCOC5 + 0.5 ISOPDOH | $5.0(-12)$ | 5 | |
| ISOPEOOH + OH → 0.83 HYAC + 0.83 GLY + 0.17 MACR + HO$_2$ | $1.0(-10)$ | 1 | N9 |
| ISOPENO3 + OH → HYAC + ETHLN + HO$_2$ | $6.0(-11)$ | 1,11 | N9 |
| ISOPBNO3 + OH → 0.85 INBO2 + 0.15 IEPOX + 0.15 NO$_2$ | $8.4(-12)\exp(390/T)$ | 1,3 | |
| INBO2 → 2 HO$_2$ + CO + MVKOOH + NO$_2$ | $7.5E12 * exp(-10000/T)$ | 3 | N11 |
| INBO2 + NO → HNO$_3$ | $K_{\text{RO2NO}} \cdot Y_{\text{nit}}(T, M, 11, 6.3)$ | 1,3 | N12 |
| INBO2 + NO → 1.85 NO$_2$ + 0.85 GLYALD + 0.85 HYAC $\quad$ + 0.15 MACRNO3 + 0.15 HO$_2$ + 0.15 HCHO | $K_{\text{RO2NO}} \cdot Y_{\text{oxy}}(T, M, 11, 6.3)$ | 1,13,3 | |
| INBO2 + NO$_3$ → 1.85 NO$_2$ + 0.85 GLYALD + 0.85 HYAC $\quad$ + 0.15 MACRNO3 + 0.85 HO$_2$ + 0.15 HCHO | $2.3(-12)$ | 1 | |
| INBO2 + HO$_2$ → HNO$_3$ | $2.5(-13)\exp(1300/T)$ | 1,3 | N13 |
| ISOPDNO3 + OH → 0.85 INDO2 + 0.15 IEPOX + 0.15 NO$_2$ | $3.9(-11)$ | 1,3 | |
| INDO2 → 3 HO$_2$ + 2 CO + OH + HYAC + NO$_2$ | $7.5E12 * exp(-10000/T)$ | 3 | N14 |
| INDO2 + NO → HNO$_3$ | $K_{\text{RO2NO}} \cdot Y_{\text{nit}}(T, M, 11, 7.9)$ | 1,3 | N12 |
| INDO2 + NO → HCHO + HO$_2$ + MVKNO3 + NO$_2$ | $K_{\text{RO2NO}} \cdot Y_{\text{oxy}}(T, M, 11, 7.9)$ | 1,3,11,12 | |
| INDO2 + NO3 → HCHO + HO$_2$ + MVKNO3 + NO$_2$ | $2.3(-12)$ | 1 | |
| INDO2 + HO$_2$ → 0.39 INDOOH + 0.65 HCHO + 0.65 HO$_2$ $\quad$ + 0.65 MVKNO3 | $2.5(-13)\exp(1300/T)$ | 1,3 | |
| INDOOH + OH → 0.39 INDO2 + 1.22 HO$_2$ + 0.61 CO $\quad$ + 0.61 MVKNO3 + 0.61 OH | $9.2(-12)$ | 1 | N15 |
| IEPOX + OH → 0.19 ICHE + 0.58 IEPOXAO2 + 0.23 IEPOXBO2 | $4.4(-11)\exp(-400/T)$ | 3 | N16 |
| ICHE + OH → 0.28 OH + 1.28 CO + 0.28 HYAC + 0.72 MVKO2 | $1.5(-11)$ | | N17 |
| ICHE + OH → CO + HO$_2$ + 0.28 HPDIAL + 0.72 HPKETAL | $2.2(-11)\exp(-400/T)$ | | N18 |
| IEPOXAO2 → DHBO + OH + CO | $1.0(7)\exp(-5000/T)$ | 3 | N19 |

| Reaction | Rate | Ref. | Note |
|---|---|---|---|
| IEPOXAO2 $\rightarrow$ CO + 2.5 HO$_2$ + 1.5 OH + 0.5 HOBA<br>  + 0.5 HPDIAL | $1.875(13)\exp(-10000/T)$ | 3 | N20 |
| IEPOXAO2 + NO $\rightarrow$ NO$_2$ + HO$_2$ + 0.8 MGLY + 0.8 GLYALD<br>  +0.2 DHBO + 0.2 CO | $K_{\text{RO2NO}}$ | 1,3 | |
| IEPOXAO2 + HO$_2$ $\rightarrow$ OH + HO$_2$ + 0.8 MGLY + 0.8 GLYALD<br>  +0.2 DHBO + 0.2 CO | $1.6(-13)\exp(1300/T)$ | 3 | N21 |
| IEPOXAO2 + HO$_2$ $\rightarrow$ CO + HO$_2$ + OH + DHBO | $0.8(-13)\exp(1300/T)$ | 3 | N22 |
| IEPOXBO2 $\rightarrow$ MACROH + OH + CO | $1.0(7)\exp(-5000/T)$ | 3 | N19 |
| IEPOXBO2 $\rightarrow$ 1.5 CO + 3 HO$_2$ + 0.5 MGLY + 0.5 HPKETAL | $1.875(13)\exp(-10000/T)$ | 3 | N23 |
| IEPOXBO2 + NO $\rightarrow$ NO$_2$ + HO$_2$ + 0.8 GLY + 0.8 HYAC<br>  +0.2 MACROH + 0.2 CO | $K_{\text{RO2NO}}$ | 1,3 | |
| IEPOXBO2 + HO$_2$ rightarrowOH + HO$_2$ + 0.8 GLY + 0.8 HYAC<br>  +0.2 MACROH + 0.2 CO | $1.6(-13)\exp(1300/T)$ | 3 | N21 |
| IEPOXBO2 + HO$_2$ $\rightarrow$ CO + HO$_2$ + OH + MACROH | $0.8(-13)\exp(1300/T)$ | 3 | N24 |
| HCOC5 + OH $\rightarrow$ C59O2 | $3.81(-11)$ | 1 | |
| C59O2 + NO $\rightarrow$ HYAC + GCO3 + NO$_2$ | $K_{\text{RO2NO}}$ | 1 | |
| C59O2 + NO$_3$ $\rightarrow$ HYAC + GCO3 + NO$_2$ | $2.3(-12)$ | 1 | |
| C59O2 + HO$_2$ $\rightarrow$ HYAC + GCO3 + OH | $2.4(-13)\exp(1300/T)$ | 1,3 | N25 |
| C59O2 + CH$_3$O$_2$ $\rightarrow$ HYAC + GCO3 + HCHO + HO$_2$ | $9.2(-14)$ | 1 | |
| C59O2 + CH$_3$CO$_3$ $\rightarrow$ HYAC + GCO3 + CO$_2$ + CH$_3$O$_2$ | $1.8(-12)\exp(500/T)$ | 6,7 | |
| ISOPBOH + OH $\rightarrow$ DHBO + CO | $3.85(-11)$ | 10 | N26 |
| ISOPDOH + OH $\rightarrow$ 0.9 DHBO + 0.9 CO + 0.1 HCOC5 + 0.1 HO$_2$ | $7.38(-11)$ | 10 | N26 |
| HPALD1 + OH $\rightarrow$ 0.45 OH + 1.35 CO$_2$ + 0.55 HCHO + 0.65 CH$_3$CO$_3$<br>  +0.2 MMAL + 0.15 MGLY + 0.15 CO + 0.1 GLY | $1.0(-11)$ | 5,3 | N27 |
| HPALD1 + OH $\rightarrow$ MVK + OH + 0.5 CO + 0.5 CO$_2$ | $0.5(-11)$ | 5,3 | N27 |
| HPALD1 + OH $\rightarrow$ MVK + OH + CO$_2$ | $1.5(-11)$ | 5,3 | N27 |
| HPALD1 + OH $\rightarrow$ MVKOOH + OH + CO | $1.4(-11)$ | 5,3 | N27 |
| HPALD1 + OH $\rightarrow$ ICHE | $0.8(-11)$ | 5,3 | N27 |
| HPALD1 + O$_3$ $\rightarrow$ 0.35 MGLY + 0.27 GLY + 1.19 OH + 0.65 CO<br>  +0.65 CH$_3$CO$_3$ + 0.08 H$_2$O$_2$ + 0.73 HPAC | $2.4(-17)$ | 1 | |
| HPALD2 + OH $\rightarrow$ 0.45 OH + 1.35 CO$_2$ + 0.55 HCHO + 0.65 CH$_3$CO$_3$<br>  +0.2 MMAL + 0.15 MGLY + 0.15 CO + 0.1 GLY | $1.0(-11)$ | 5,3 | N28 |
| HPALD2 + OH $\rightarrow$ MACR + OH + 0.5 CO + 0.5 CO$_2$ | $0.5(-11)$ | 5,3 | N28 |
| HPALD2 + OH $\rightarrow$ MACR + OH + CO$_2$ | $1.5(-11)$ | 5,3 | N28 |

| Reaction | Rate | Ref. | Note |
|---|---|---|---|
| $HPALD2 + OH \rightarrow OH + 2\,CO + 2\,HO_2 + HPACET$ | $0.8(-11)$ | 5,3 | N28 |
| $HPALD2 + OH \rightarrow ICHE$ | $1.4(-11)$ | 5,3 | N28 |
| $HPALD2 + O_3 \rightarrow 0.27\,HPACET + 1.7\,OH + 0.28\,HO_2$ $\quad + 0.5\,CO + 0.73\,MGLY + 0.74\,GLY + 0.02\,CO_2$ | $2.4(-17)$ | 1 | |
| $MMAL + OH \rightarrow MGLY + HO_2 + 2\,CO_2$ | $1.5(-12)$ | 1 | N29 |
| $DIHPMEK + OH \rightarrow 2\,OH + CH_3CO_3 + CO + HCHO$ | $1.63(-11)$ | 1 | N30 |
| $DIHPMEK + OH \rightarrow OH + HPKETAL$ | $1.28(-11)$ | 1 | |
| $HPKETAL + OH \rightarrow 0.6\,OH + CO + 0.6\,MGLY$ $\quad + 0.4\,CH_3CO_3 + 0.4\,HO_2$ | $3.0(-11)$ | | N31 |
| $HPDIAL + OH \rightarrow OH + CO + MGLY$ | $3.0(-11)$ | | N32 |
| $NISOPO2 + NO \rightarrow 1.82\,NO_2 + 0.42\,MVK + 0.04\,MACR$ $\quad + 1.54\,HCHO + 0.18\,NC4CHO + 0.9\,HO_2 + 0.72\,CO$ | $K_{RO2NO}$ | 1,15,3 | N33 |
| $NISOPO2 + NO_3 \rightarrow 1.82\,NO_2 + 0.42\,MVK + 0.04\,MACR$ $\quad + 1.54\,HCHO + 0.18\,NC4CHO + 0.9\,HO_2 + 0.72\,CO$ | $2.3(-12)$ | 1,15,3 | |
| $NISOPO2 + HO_2 \rightarrow 0.535\,NISOPOOHD + 0.22\,NISOPOOHB$ $\quad + 0.245\,OH + 0.245\,NO_2 + 0.225\,MVK + 0.02\,MACR + 0.245\,HCHO$ | $2.5(-13)\exp(1300/T)$ | 1,15,3 | |
| $NISOPO2 + NISOPO2 \rightarrow 0.17\,MVK + 0.11\,MACR + 0.7\,HCHO$ $\quad + 0.42\,NO_2 + 0.78\,NC4CHO + 0.36\,HO_2 + 0.28\,CO$ $\quad + 0.59\,ISOPCNO3 + 0.11\,ISOPANO3 + 0.1\,ISOPDNO3$ | $2.0(-12)$ | 15,3 | N34 |
| $NISOPO2 + CH_3O_2 \rightarrow 0.08\,MVK + 0.06\,MACR + 0.95\,HCHO$ $\quad + 0.21\,NO_2 + 0.39\,NC4CHO + 0.38\,HO_2 + 0.14\,CO + 0.4\,CH_3OH$ $\quad + 0.29\,ISOPCNO3 + 0.06\,ISOPANO3 + 0.05\,ISOPDNO3$ | $7.5(-13)$ | 15,3 | N34 |
| $NISOPO2 + CH_3CO_3 \rightarrow 0.38\,MVK + 0.05\,MACR + 1.39\,HCHO$ $\quad + 0.75\,NO_2 + 0.25\,NC4CHO + 0.81\,HO_2 + 0.64\,CO + 0.9\,CH_3O_2$ $\quad + 0.9\,CO_2 + 0.1\,CH_3COOH$ | $2.0(-12)\exp(500/T)$ | 15,3 | N34 |
| $NISOPO2 + ISOPBO2 \rightarrow 0.71\,MVK + 0.08\,MACR + 1.33\,HCHO$ $\quad + 0.47\,NO_2 + 0.53\,NC4CHO + 0.95\,HO_2 + 0.36\,CO + 0.5\,ISOPBOH$ | $7.5(-13)$ | 15,3 | N34 |
| $NISOPO2 + ISOPDO2 \rightarrow 0.08\,MVK + 0.26\,MACR + 0.55\,HCHO$ $\quad + 0.21\,NO_2 + 0.39\,NC4CHO + 0.38\,HO_2 + 0.14\,CO + 0.4\,ISOPDOH$ $\quad + 0.29\,ISOPCNO3 + 0.06\,ISOPANO3 + 0.05\,ISOPDNO3 + 0.4\,HCOC5$ | $6.8(-12)$ | 15,3 | N34 |
| $NISOPOOHD + OH \rightarrow NISOPO2$ | $3.4(-12)\exp(200/T)$ | 3 | N35 |
| $NISOPOOHD + OH \rightarrow OH + NC4CHO$ | $7.5(-12)\exp(20/T)$ | 3 | N35 |
| $NISOPOOHD + OH \rightarrow 0.19\,CO + 0.95\,HO_2 + 0.43\,OH + 0.69\,NOA$ $\quad + 0.19\,HCHO + 0.5\,HPAC + 0.07\,HPACET + 0.07\,ETHLN$ | $2.37(-11)\exp(390/T)$ | 3 | N36 |

| Reaction | Rate | Ref. | Note |
|---|---|---|---|
| $+0.24\,IHNE$ | | | |
| $NISOPOOHD + O_3 \rightarrow 0.2\,OH + 0.87\,NOA$ | $1.3(-17)$ | 15 | N37 |
| $\quad +0.13\,HPACET + 0.84\,HPAC + 0.16\,ETHLN$ | | | |
| $NISOPOOHB + OH \rightarrow NISOPO2$ | $3.4(-12)\exp(200/T)$ | 3 | N38 |
| $NISOPOOHB + OH \rightarrow 0.23\,GLYALD + 0.47\,NOA + 0.76\,OH + 0.09\,CO$ | $8.72(-12)\exp(390/T)$ | 3 | N39 |
| $\quad +0.33\,HO_2 + 0.09\,HCHO + 0.15\,HPAC + 0.04\,HYAC$ | | | |
| $\quad +0.04\,ETHLN + 0.51\,IHNE$ | | | |
| $IHNE + OH \rightarrow 0.23\,HMVK + 0.03\,HMAC + 0.82\,HCHO + 0.8\,NO_2$ | $3.22(-11)\exp(-400/T)$ | 3 | N40 |
| $\quad +0.8\,CO + 0.17\,NOA + 0.45\,MGLY + 0.72\,HO_2 + 0.38\,OH$ | | | |
| $\quad +0.03\,MVKNO3 + 0.09\,HYAC + 0.09\,CO_2$ | | | |
| $NC4CHO + OH \rightarrow 0.45\,CO_2 + 1.08\,CO + 0.85\,HO_2 + 0.58\,NOA + 0.5\,OH$ | $4.1(-11)$ | 15,3 | N41 |
| $\quad +0.12\,HCHO + 0.12\,MGLY + 0.17\,NO_2 + 0.11\,MVKNO3$ | | | |
| $\quad +0.05\,ICHE + 0.14\,CH_3CO_3 + 0.14\,ETHLN$ | | | |
| $NC4CHO + NO_3 \rightarrow HNO_3 + CO_2 + 0.75\,NOA + 0.75\,CO + 0.75\,HO_2$ | $6.0(-12)\exp(-1860/T)$ | 1,3 | N41 |
| $\quad +0.25\,CH_3CO_3 + 0.25\,ETHLN$ | | | |
| $NC4CHO + O_3 \rightarrow 0.555\,NOA + 0.89\,CO + 0.89\,OH + 0.445\,MGLY$ | $4.4(-18)$ | 1 | |
| $\quad +0.445\,HO_2 + 0.075\,H_2O_2 + 0.445\,NO_2 + 0.52\,GLY$ | | | |
| $\quad +0.035\,OCHCOOH$ | | | |
| $ISOPCNO3 + O_3 \rightarrow 0.555\,NOA + 0.52\,GLYALD + 0.07\,C_2H_5COOH$ | $2.8(-17)$ | 1,11 | |
| $\quad +0.075\,H_2O_2 + 0.89\,OH + 0.445\,NO_2 + 0.445\,MGLY$ | | | |
| $\quad +0.445\,HO_2 + 0.445\,CO + 0.445\,HCHO$ | | | |
| $ISOPCNO3 + OH \rightarrow 1.2\,OH + 1.2\,CO + HO_2 + 0.6\,NOA + 0.4\,NC4CHO$ | $7.5(-12)\exp(20/T)$ | 3 | N42 |
| $ISOPCNO3 + OH \rightarrow 0.92\,INCO2 + 0.08\,IEPOX + 0.08\,NO_2$ | $2.04(-11)\exp(390/T)$ | 3 | N43 |
| $INCO2 \rightarrow 4\,HO_2 + 2\,CO + OH + NOA$ | $1.256(13)\exp(-10000/T)$ | 3 | N44 |
| $INCO2 + NO \rightarrow INCNO3$ | $K_{\text{RO2NO}} \cdot Y_{\text{nit}}(T,M,11,4.7)$ | 3 | |
| $INCO2 + NO \rightarrow NO_2 + HO_2 + NOA + GLYALD$ | $K_{\text{RO2NO}} \cdot Y_{\text{oxy}}(T,M,11,4.7)$ | 3 | N43 |
| $INCO2 + NO_3 \rightarrow NO_2 + HO_2 + NOA + GLYALD$ | $2.3(-12)$ | 1 | N43 |
| $INCO2 + HO_2 \rightarrow 0.32\,INCCO + 0.11\,INCO2 + 0.57\,NOA + 0.57\,GLYALD$ | $2.5(-13)\exp(1300/T)$ | 3 | N45 |
| $\quad +0.57\,HO_2 + 0.46\,OH$ | | | |
| $INCCO + OH \rightarrow HCHO + 3\,HO_2 + CH_3CO_3 + 2\,CO + NO_2$ | $3.3(-12)$ | 1 | N46 |
| $INCNO3 + OH \rightarrow 0.445\,INCCO + 0.414\,GLY + 0.414\,HO_2$ | $1.98(-12)$ | 1 | N47 |
| $\quad +0.555\,NOA + 0.141\,GLYALD + NO_2$ | | | |
| $ISOPANO3 + O_3 \rightarrow 0.555\,HYAC + 0.555\,ETHLN + 0.89\,OH$ | $2.8(-17)$ | 1,11 | |
| $\quad +0.445\,NO_2 + 0.445\,GLY + 0.445\,HO_2 + 0.055\,H_2O_2$ | | | |

| Reaction | Rate | Ref. | Note |
|---|---|---|---|
| ISOPANO3 + OH → 1.2 OH + 0.6, CO + 0.6 CH$_3$CO$_3$ + 0.6 ETHLN <br> +0.4 HO$_2$ + 0.4 NC4CHO | $7.5(-12)\exp(20/T)$ | 3 | N42 |
| ISOPANO3 + OH → 0.96 INAO2 + 0.04 IEPOX + 0.04 NO$_2$ | $2.95(-11)\exp(390/T)$ | 3 | N43 |
| INAO2 → 3 HO$_2$ + CO + CH$_3$CO$_3$ + OH + ETHLN | $5.092(12)\exp(-10000/T)$ | 3 | N48 |
| INAO2 + NO → HNO$_3$ | $K_{\text{RO2NO}} \cdot Y_{\text{nit}}(T, M, 11, 2.3)$ | 1 | N12 |
| INAO2 + NO → 0.86 HYAC + 0.86 ETHLN + 0.14 MVKNO3 <br> +0.14 HCHO + HO$_2$ + NO$_2$ | $K_{\text{RO2NO}} \cdot Y_{\text{oxy}}(T, M, 11, 2.3)$ | 3 | N43 |
| INAO2 + NO$_3$ → 0.86 HYAC + 0.86 ETHLN + 0.14 MVKNO3 <br> +0.14 HCHO + HO$_2$ + NO$_2$ | $2.3(-12)$ | 1 | N43 |
| INAO2 + HO$_2$ → 0.32 CO + 0.64 HO$_2$ + 0.33 OH + 0.18 INAO2 <br> +0.44 HYAC + 0.44 ETHLN + 0.06 HCHO + 0.38 MVKNO3 | $2.6(-13)\exp(1300/T)$ | 3 | N49 |
| HALD1 + OH → CO + 2 OH + CO$_2$ + 0.5 CH$_3$CO$_3$ + 0.5 HMVK | $1.5(-11)$ | | N50 |
| HALD1 + OH → 0.65 IEPOXAO2 + 0.35 GLYALD + 0.35 MGLY + 0.35 HO$_2$ | $2.2(-11)$ | | N51 |
| HALD1 + NO$_3$ → 2 CO + CO$_2$ + 3 OH + HO$_2$ + CH$_3$CO$_3$ + HNO$_3$ | $5.6(-12)\exp(-1860/T)$ | N50 | |
| HALD1 + O$_3$ → 0.55 GLYALD + 0.55 MGLY + 0.9 OH <br> +0.45 CO + 0.45 CH$_3$CO$_3$ + 0.45 HO$_2$ + 0.45 GLY | $2.4(-17)$ | 1 | |
| HALD2 + OH → 0.5 CO + 1.5 OH + 0.5 CH$_3$CO$_3$ + 0.5 CO$_2$ <br> +0.5 PGA + 0.5 HMAC | $1.5(-11)$ | | N50 |
| HALD2 + OH → 0.35 IEPOXBO2 + 0.65 HYAC + 0.65 GLY + 0.65 HO$_2$ | $2.2(-11)$ | | N51 |
| HALD2 + NO$_3$ → CO + 2 OH + CH$_3$CO$_3$ + PGA + HNO$_3$ | $5.6(-12)\exp(-1860/T)$ | | N50 |
| HALD2 + O$_3$ → 0.55 HYAC + 0.55 GLY + 0.9 OH + 0.9 HO$_2$ <br> +0.9 CO + 0.05 H$_2$O$_2$ + 0.45 MGLY | $2.4(-17)$ | 1 | |

*C$_4$ compounds*

| Reaction | Rate | Ref. | Note |
|---|---|---|---|
| MACR + OH → CO + 0.036 HPACET + 0.036 HO$_2$ + 0.964 HYAC <br> +0.964 OH | $4.4(-12)\exp(380/T)$ | 3 | N52 |
| MACR + OH → MCO3 | $2.7(-12)\exp(470/T)$ | 3 | |
| MACR + O$_3$ → 0.9 MGLY + 0.12 HCHO + 0.1 CO + 0.1 OH <br> +0.1 CH$_3$CO$_3$ + 0.88 (0.55 HMHP + 0.4 HCHO + 0.4 H$_2$O$_2$ <br> +0.05 HCOOH) | $1.4(-15)\exp(-2100/T)$ | 1 | N2 |
| MACR + NO$_3$ → MCO3 + HNO$_3$ | $3.4(-15)$ | 1 | |
| MCO3 + NO → CO$_2$ + 0.65 CH$_3$O$_2$ + 0.65 CO + 0.35 CH$_3$CO$_3$ <br> +HCHO + NO$_2$ | $8.70(-12)\exp(290/T)$ | 1 | |
| MCO3 + NO$_3$ → CO$_2$ + 0.65 CH$_3$O$_2$ + 0.65 CO + 0.35 CH$_3$CO$_3$ <br> +HCHO + NO$_2$ | $4.0(-12)$ | 1 | |

| Reaction | Rate | Ref. | Note |
|---|---|---|---|
| $MCO3 + HO_2 \rightarrow MCO3H$ | $2.43(-13)\exp(980/T)$ | 1,18 | |
| $MCO3 + HO_2 \rightarrow MCOOH + O_3$ | $1.25(-13)\exp(980/T)$ | 1,18 | |
| $MCO3 + HO_2 \rightarrow CO_2 + 0.65\,CH_3O_2 + 0.65\,CO + 0.35\,CH_3CO_3$ $+ HCHO + OH$ | $4.15(-13)\exp(980/T)$ | 1,18 | |
| $MCO3 + CH_3O_2 \rightarrow 0.585\,CH_3O_2 + 0.585\,CO + 0.315\,CH_3CO_3$ $+ 1.9\,HCHO + 0.9\,HO_2 + 0.9\,CO_2 + 0.1\,MCOOH$ | $2.0(-12)\exp(500/T)$ | 1,6,7 | |
| $MCO3 + CH_3CO_3 \rightarrow 1.65\,CH_3O_2 + 0.65\,CO + 0.35\,CH_3CO_3$ $+ HCHO + 2\,CO_2$ | $5.4(-12)\exp(500/T)$ | 1,6,7 | |
| $MCO3 + ISOPBO2 \rightarrow 0.65\,CH_3O_2 + 0.65\,CO + 0.35\,CH_3CO_3$ $+ 2\,HCHO + MVK + HO_2 + CO_2$ | $1.8(-12)\exp(500/T)$ | 1,6,7 | |
| $MCO3 + ISOPDO2 \rightarrow 0.585\,CH_3O_2 + 0.585\,CO + 0.315\,CH_3CO_3$ $+ 1.8\,HCHO + 0.9\,MACR + 0.9\,HO_2 + 0.9\,CO_2$ $+ 0.1\,MCOOH + 0.1\,HCOC5$ | $2.0(-12)\exp(500/T)$ | 1,6,7 | |
| $MCO3 + NO_2 \rightarrow MPAN$ | $k_0 = 3.28(-28)(300/T)^{6.87}$ $k_\infty = 1.125(-11)(300/T)^{1.105}$ | 1,19 | |
| $MPAN \rightarrow MCO3 + NO_2$ | $1.6(16)\exp(-13500/T)$ | 1 | |
| $MPAN + OH \rightarrow HYAC + CO + NO_3$ | $7.5(-12)$ | 20 | |
| $MPAN + OH \rightarrow HMML + NO_3$ | $2.25(-11)$ | 20 | |
| $MPAN + O_3 \rightarrow HCHO + CH_3CO_3 + NO_3 + CO_2$ | $8.2(-18)$ | 1 | |
| $MCO3H + OH \rightarrow MCO3$ | $3.6(-12)$ | 1 | |
| $MCO3H + OH \rightarrow 0.83\,HYAC + 0.83\,CO + 0.17\,HMML + OH$ | $1.3(-11)$ | 1 | |
| $MCOOH + OH \rightarrow CO_2 + 0.65\,CH_3O_2 + 0.65\,CO$ $+ 0.35\,CH_3CO_3 + HCHO$ | $1.51(-11)$ | 1 | |
| $HMML + OH \rightarrow 1.13\,CO + 1.05\,OH + 0.39\,HO_2 + 0.48\,CH_3CHO$ $+ 0.87\,CO_2 + 0.44\,CH_3CO_3 + 0.08\,CH_3COOH$ | $4.33(-12)$ | | N53 |
| $MVK + OH \rightarrow MVKO2$ | $2.6(-12)\exp(610/T)$ | 1 | |
| $MVK + O_3 \rightarrow 0.313\,CH_3CO_3 + 0.545\,MGLY + 0.129\,HO_2$ $+ 0.19\,CO + 0.22\,OH + 0.8\,HCHO + 0.136\,CH_3CHO$ $+ 0.165\,CO_2 + 0.245\,H_2O_2 + 0.275\,HMHP$ $+ 0.025\,HCOOH + 0.006\,CH_3COOH)$ | $8.5(-16)\exp(-1520/T)$ | 1 | N54 |
| $MVKO2 + NO \rightarrow 0.28\,MGLY + 0.28\,HCHO + 0.28\,HO_2$ $+ 0.72\,GLYALD + 0.72\,CH_3CO_3 + NO_2$ | $K_{RO2NO} \cdot Y_{oxy}(T,M,6,4.6)$ | 1,21,22 | N55 |
| $MVKO2 + NO \rightarrow MVKNO3$ | $K_{RO2NO} \cdot Y_{nit}(T,M,6,4.6)$ | 22 | |
| $MVKO2 + NO_3 \rightarrow 0.28\,MGLY + 0.28\,HCHO + 0.28\,HO_2$ | $2.3(-12)$ | 1 | N55 |

| Reaction | Rate | Ref. | Note |
|---|---|---|---|
| $+0.72\,GLYALD + 0.72\,CH_3CO_3 + NO_2$ | | | |
| $MVKO2 + HO_2 \rightarrow 0.35\,GLYALD + 0.35\,CH_3CO_3 + 0.52\,OH$ | $2.1(-13)\exp(1300/T)$ | 22,3 | N55 |
| $\quad +0.174\,HO_2 + 0.48\,MVKOOH + 0.13\,BIACETOH$ | | | |
| $\quad +0.04\,MGLY + 0.04\,HCHO$ | | | |
| $MVKO2 + CH_3O_2 \rightarrow 0.14\,MGLY + 0.36\,GLYALD$ | $1.16(-12)$ | 1 | N55 |
| $\quad +0.36\,CH_3CO_3 + 0.89\,HCHO + 0.64\,HO_2 + 0.25\,DHBO$ | | | |
| $\quad +0.18\,BIACETOH + 0.07\,HOBA + 0.25\,CH_3OH$ | | | |
| $MVKO2 + CH_3CO_3 \rightarrow 0.25\,MGLY + 0.65\,GLYALD$ | $2.0(-12)\exp(500/T)$ | 1,6,7 | |
| $\quad +0.65\,CH_3CO_3 + 0.25\,HCHO + 0.25\,HO_2 + 0.9\,CH_3O_2$ | | | |
| $\quad +0.9\,CO_2 + 0.1\,CH_3COOH + 0.1\,DHBO$ | | | |
| $MVKOOH + OH \rightarrow 0.55\,BIACETOH + 0.55\,OH + 0.45\,HOBA$ | $4.5(-11)$ | 1 | N56 |
| $MACRNO3 + OH \rightarrow 0.5\,HYAC + 0.5\,MGLY + 0.5\,HO_2 + 0.5\,CO$ | $3.0(-12)$ | 1 | N57 |
| $\quad +0.5\,CO_2 + NO_2$ | | | |
| $MVKNO3 + OH \rightarrow 0.5\,BIACETOH + 0.4\,GLY + 0.4\,CH_3CO_3$ | $1.76(-12)$ | 1 | N58 |
| $\quad +0.1\,MGLY + 0.1\,CO_2 + 0.5\,HO_2 + NO_2$ | | | |
| $MVKNO3 + OH \rightarrow HOBA + NO_2$ | $0.44(-12)$ | 1 | N58 |
| $HOBA + OH \rightarrow 0.84\,MGLY + HO_2 + 0.16\,CH_3CO_3 + 0.32\,CO$ | $2.45(-11)$ | 1,14 | N59 |
| $HOBA + NO_3 \rightarrow HNO_3 + MGLY + HO_2$ | $5.6(-12)\exp(-1860/T)$ | 1 | |
| $DHBO + OH \rightarrow 0.61\,BIACETOH + 0.39\,HOBA$ | $8.7(-12)\exp(70/T)$ | 14 | |
| $MACROH + OH \rightarrow HO_2 + 0.84\,HYAC + 0.84\,OH + 0.84\,CO$ | $2.4(-11)\exp(70/T)$ | 3 | N60 |
| $\quad -0.16\,OH + 0.16\,MGLY + 0.16\,HO_2 + 0.16\,CO_2$ | | | |
| $BIACETOH + OH \rightarrow CH_3CO_3 + 2\,CO + HO_2$ | $2.69(-12)$ | 14 | |
| $HMVK + OH \rightarrow HCOOH + OH + MGLY$ | $6.0(-11)$ | | N61 |
| $HMVK + OH \rightarrow HO2 + HOBA$ | $2.4(-11)$ | | N61 |
| $HMAC + OH \rightarrow 0.5\,HCOOH + 0.5\,OH + 0.5\,MGLY$ | $3.0(-11)$ | | N62 |
| $\quad +0.5\,CO + 0.5\,OH + 0.5\,DHA$ | | | |
| $HMAC + OH \rightarrow 0.89\,CO + 1.34\,OH + 0.78\,CH_3CO_3$ | $2.7(-11)$ | | N63 |
| $\quad +0.89\,CO_2 + 0.44\,HO_2 + 0.22\,MGLY$ | | | |
| $HMAC + NO_3 \rightarrow CO + 2\,OH + CH_3CO_3 + CO_2 + HNO_3$ | $3.4(-15)$ | | N63 |
| $C_3$ compounds | | | |
| $CH_3COCH_3 + OH \rightarrow ACETO2$ | $1.33(-13) + 3.82(-11)\exp(-2000/T)$ | 1 | |
| $HPACET + OH \rightarrow MGLY + OH$ | $8.39(-12)$ | 1 | |
| $HPACET + OH \rightarrow ACETO2$ | $1.9(-12)\exp(190/T)$ | 1 | |
| $ACETO2 + NO \rightarrow NO_2 + HCHO + CH_3CO_3$ | $K_{\mathrm{RO2NO}} \cdot Y_{\mathrm{oxy}}(T, M, 4, 5.2)$ | 1 | |

| Reaction | Rate | Ref. | Note |
|---|---|---|---|
| $ACETO2 + NO \rightarrow NOA$ | $K_{RO2NO} \cdot Y_{nit}(T, M, 4, 5.2)$ | 1 | N64 |
| $ACETO2 + NO_3 \rightarrow NO_2 + HCHO + CH_3CO_3$ | $2.3(-12)$ | 1 | |
| $ACETO2 + HO_2 \rightarrow 0.85\,HPACET$ <br> $\quad + 0.15\,HCHO + 0.15\,CH_3CO_3$ | $8.6(-13)\exp(700/T)$ | 1,19 | |
| $ACETO2 + CH_3O_2 \rightarrow 0.3\,CH_3CO_3 + 0.8\,HCHO + 0.3\,HO_2$ <br> $\quad + 0.2\,HYAC + 0.5\,MGLY + 0.5\,CH_3OH$ | $3.8(-12)$ | 7 | |
| $ACETO2 + CH_3CO_3 \rightarrow CH_3COOH + MGLY$ | $2.5(-12)$ | 7 | |
| $ACETO2 + CH_3CO_3 \rightarrow CH_3O_2 + CO_2 + CH_3CO_3 + HCHO$ | $2.5(-12)$ | 7 | |
| $ACETO2 + ACETO2 \rightarrow HYAC + MGLY$ | $3.0(-12)$ | 7 | |
| $ACETO2 + ACETO2 \rightarrow 2\,CH_3CO_3 + 2\,HCHO$ | $5.0(-12)$ | 7 | |
| $HYAC + OH \rightarrow MGLY + HO_2$ | $1.46(-13)\exp(1100/T) \cdot (T/300)^{2.6}$ | 1,23 | |
| $MGLY + OH \rightarrow 0.6\,CH_3CO_3 + 0.4\,CH_3O_2 + 1.4\,CO + H_2O$ | $1.9(-12)\exp(575/T)$ | 1,24 | |
| $MGLY + NO_3 \rightarrow HNO_3 + CO + CH_3CO_3$ | $3.36(-12)\exp(-1860/T)$ | 1 | |
| $NOA + OH \rightarrow MGLY + NO_2$ | $6.7(-13)$ | 1 | |
| $MVA + OH \rightarrow 0.5\,CH_3COOH + 0.5\,HCHO + 0.5\,OH$ <br> $\quad + 0.5\,HYAC + 0.5\,HO_2$ | $9.0(-11)$ | | N65 |
| $DHA + OH \rightarrow 1.39\,HO_2 + 0.48\,CH_3CHO + 0.87\,CO_2$ <br> $\quad + 0.44\,CH_3CO_3 + 0.08\,CH_3COOH + 0.13\,CO + 0.05\,OH$ | $8.0(-12)\exp(70/T)$ | 3,19 | N66 |

$C_2$ compounds

| Reaction | Rate | Ref. | Note |
|---|---|---|---|
| $GLYALD + OH \rightarrow 0.78\,GCO3 + 0.22\,GLY + 0.22\,HO_2$ | $1.0(-11)$ | 1,25 | |
| $GLYALD + NO_3 \rightarrow GCO3 + HNO_3$ | $1.4(-12)\exp(-1860/T)$ | 1 | |
| $GCO3 + NO \rightarrow NO_2 + HO_2 + HCHO + CO_2$ | $6.7(-12)\exp(340/T)$ | 1 | |
| $GCO3 + NO_3 \rightarrow NO_2 + HO_2 + HCHO + CO_2$ | $4.0(-12)$ | 1 | |
| $GCO3 + HO_2 \rightarrow 0.21\,GCO3H + 0.04\,GCOOH + 0.04\,O_3$ <br> $\quad + 0.75\,HO_2 + 0.75\,HCHO + 0.75\,OH + 0.75\,CO_2$ | $7.84(-13)\exp(980/T)$ | 1,17,26 | |
| $GCO3 + CH_3O_2 \rightarrow 1.9\,HCHO + 1.8\,HO_2 + 0.1\,GCOOH + 0.9\,CO_2$ | $1.8(-12)\exp(500/T)$ | 1,6,7 | |
| $GCO3 + CH_3CO_3 \rightarrow CH_3O_2 + HO_2 + HCHO + 2\,CO_2$ | $5.4(-12)\exp(500/T)$ | 1,6,7 | |
| $GCO3 + NO_2 \rightarrow GPAN$ | $k_0 = 3.28(-28)(300/T)^{6.87}$ <br> $k_\infty = 1.125(-11)(300/T)^{1.105}$ | 1,19 | |
| $GPAN \rightarrow GCO3 + NO_2$ | $k_0 = 1.1(-5)\exp(-10100/T)$ <br> $k_\infty = 1.9(17)\exp(-14100/T)$ | 1,19 | |
| $GPAN + OH \rightarrow HCHO + CO + NO_2$ | $1.12(-12)$ | 1 | |
| $GCO3H + OH \rightarrow GCO3$ | $6.19(-12)$ | 1 | |
| $GLY + OH \rightarrow 0.72\,HO_2 + 0.28\,OH + 1.55\,CO + 0.45\,CO_2$ | $3.1(-12)\exp(340/T)$ | 1 | N67 |

| Reaction | Rate | Ref. | Note |
|---|---|---|---|
| $GLY + NO_3 \rightarrow HNO_3 + 0.72\,HO_2 + 0.28\,OH + 1.55\,CO + 0.45\,CO_2$ | $1.4(-12)\exp(-1860/T)$ | 1 | N67 |
| $HPAC + OH \rightarrow GLY + OH$ | $1.0(-11)$ | 1 | N68 |
| $HPAC + OH \rightarrow 0.25\,CO + HCHO + OH + 0.75\,CO_2$ | $1.8(-11)$ | 1 | N68 |
| $HPAC + OH \rightarrow OCHCH_2O_2$ | $1.90(-12)\exp(190/T)$ | 1 | |
| $C_2H_5OH + OH \rightarrow 0.95\,CH_3CHO + 0.95\,HO_2 + 0.05\,HOCH_2CH_2O_2$ | $3.0(-12)\exp(20/T)$ | 1 | |
| $CH_3CHO + OH \rightarrow 0.95\,CH_3CO_3 + 0.05\,OCHCH_2O_2$ | $4.7(-12)\exp(345/T)$ | 1 | |
| $CH_3CHO + NO_3 \rightarrow CH_3CO_3 + HNO_3$ | $1.4(-12)\exp(-1860/T)$ | 1 | |
| $OCHCH_2O_2 + NO \rightarrow NO_2 + HCHO + CO + HO_2$ | $K_{RO2NO}$ | 1 | |
| $OCHCH_2O_2 + NO_3 \rightarrow NO_2 + HCHO + CO + HO_2$ | $2.3(-12)$ | 1 | |
| $OCHCH_2O_2 + HO_2 \rightarrow HPAC$ | $1.4(-13)\exp(1300/T)$ | 1,3 | |
| $OCHCH_2O_2 + CH_3O_2 \rightarrow 1.25\,HCHO + 0.5\,CO + HO_2$ $\qquad + 0.25\,GLY + 0.25\,CH_3OH + 0.25\,GLYALD$ | $2.0(-12)$ | 1,5 | |
| $CH_3CO_3 + NO \rightarrow NO_2 + CH_3O_2 + CO_2$ | $7.5(-12)\exp(290/T)$ | 1 | |
| $CH_3CO_3 + NO_3 \rightarrow NO_2 + CH_3O_2 + CO_2$ | $4.0(-12)$ | 1 | |
| $CH_3CO_3 + HO_2 \rightarrow 0.31\,PAA + 0.16\,CH_3COOH + 0.16\,O_3$ $\qquad + 0.53\,CH_3O_2 + 0.53\,OH + 0.53\,CO_2$ | $7.84(-13)\exp(980/T)$ | 1,18 | |
| $CH_3CO_3 + CH_3O_2 \rightarrow HCHO + 0.9\,HO_2 + 0.9\,CH_3O_2$ $\qquad + 0.9\,CO_2 + 0.1\,CH_3COOH$ | $2.0(-12)\exp(500/T)$ | 6,7 | |
| $CH_3CO_3 + CH_3CO_3 \rightarrow 2\,CH_3O_2 + 2\,CO_2$ | $2.9(-12)\exp(500/T)$ | 6,7 | |
| $CH_3CO_3 + NO_2 \rightarrow PAN$ | $k_0 = 3.28(-28)(300/T)^{6.87}$ $k_\infty = 1.125(-11)(300/T)^{1.105}$ | 1,19 | |
| $PAN \rightarrow CH_3CO_3 + NO_2$ | $k_0 = 1.1(-5)\exp(-10100/T)$ $k_\infty = 1.9(17)\exp(-14100/T)$ | 1,19 | |
| $PAA + OH \rightarrow CH_3CO_3$ | $3.7(-12)$ | 1 | |
| $CH_3COOH + OH \rightarrow CH_3O_2 + CO_2$ | $3.15(-14)\exp(920/T)$ | 1,19 | |
| $ETHLN + OH \rightarrow HCHO + NO_2 + CO_2$ | $2.0(-12)$ | 1 | N69 |
| $ETHLN + NO_3 \rightarrow HCHO + NO_2 + CO_2$ | $1.4(-12)\exp(1860/T)$ | 1 | |
| $VA + OH \rightarrow 0.64\,HCOOH + 0.64\,HCHO + 0.64\,OH$ $\qquad + 0.36\,GLYALD + 0.36\,HO_2$ | $6.8(-11)$ | 28 | N70 |
| $PGA + OH \rightarrow CO + CO_2 + OH$ | $1.6(-11)$ | 1 | |
| $C_1$ compounds | | | |
| $CH_3O_2 + NO \rightarrow NO_2 + HCHO + HO_2$ | $2.8(-12)\exp(300/T)$ | 19 | |
| $CH_3O_2 + NO \rightarrow CH_3ONO_2$ | $2.8(-12)\exp(300/T) \cdot Y_{nit}(T,M,1,50.)$ | 19 | N71 |
| $CH_3O_2 + NO_3 \rightarrow NO_2 + HCHO + HO_2$ | $1.2(-12)$ | 1 | |

| Reaction | Rate | Ref. | Note |
|---|---|---|---|
| $CH_3O_2 + HO_2 \rightarrow 0.9\,CH_3OOH + 0.1\,HCHO$ | $4.1(-13)\exp(750/T)$ | 19 | |
| $CH_3O_2 + CH_3O_2 \rightarrow 2\,HCHO + 2\,HO_2$ | $9.5(-14)\exp(390/T)$ | 19 | |
| | $/(1 + 0.0382\exp(1130/T))$ | | |
| $CH_3O_2 + CH_3O_2 \rightarrow HCHO + CH_3OH$ | $9.5(-14)\exp(390/T)$ | 19 | |
| | $/(1 + 26.2\exp(-1130/T))$ | | |
| $CH_3O_2 + O_3 \rightarrow HCHO + HO_2$ | $2.9(-16)\exp(-1000/T)$ | 19 | |
| $CH_3O_2 + OH \rightarrow 0.92HCHO + 1.84HO_2 + 0.08\,CH_3OH$ | $1.6(-10)\cdot(1 - f_{\text{stab}})$ | 28-31 | N72 |
| $CH_3O_2 + OH \rightarrow CH_3OOOH$ | $1.6(-10)\cdot f_{\text{stab}}$ | 31 | N72 |
| $CH_3OOOH + OH \rightarrow HCHO + HO_2$ | $2.2(-11)$ | 31 | |
| $CH_3OOOH \rightarrow 0.2\,CH_3OH + 0.8\,HCHO + 1.6\,HO_2$ | $1.1(14)(T/300)^{3.5}\exp(-12130/T)$ | 31 | |
| $CH_3OOOH + (H_2O)_2 \rightarrow CH_3OH$ | $3.0(-15)\exp(-2500/T)$ | 31 | N73 |
| $CH_3OOH + OH \rightarrow 0.3\,HCHO + 0.3\,OH + 0.7\,CH_3O_2$ | $3.8(-12)\exp(200/T)$ | 19 | |
| $CH_3ONO_2 + OH \rightarrow HCHO + NO_2$ | $8.0(-13)\exp(-1000/T)$ | 19 | |
| $HMHP + OH \rightarrow 0.45\,HCOOH + 0.45\,OH$ | $1.3(-12)\exp(500/T)$ | 3,32 | N74 |
| $\quad + 0.55\,HCHO + 0.55\,HO_2$ | | | |
| $CH_3OH + OH \rightarrow HCHO + HO_2$ | $2.9(-12)\exp(-345/T)$ | 19 | |
| $HCHO + OH \rightarrow CO + HO_2$ | $55(-12)\exp(125/T)$ | 19 | |
| $HCHO + NO_3 \rightarrow CO + HO_2 + HNO_3$ | $5.8(-16)$ | 19 | |
| $HCOOH + OH \rightarrow CO_2 + HO_2$ | $4.5(-13)$ | 1 | |

*oxidation of monoterpenes*

| Reaction | Rate | Ref. | Note |
|---|---|---|---|
| $APIN + OH \rightarrow APINOHO2 + 0.1\,HCOOH + 1.3\,HCHO$ | $1.2(-11)\exp(440/T)$ | 1 | N75 |
| $\quad + CH_3COCH_3 + 0.2\,GLY + 0.05\,MGLY$ | | | |
| $APIN + O_3 \rightarrow APINO3O2 + 0.15\,OH + 0.1\,HCOOH$ | $8.05(-16)\exp(-640/T)$ | 1 | N75 |
| $\quad + 1.3\,HCHO + 0.06\,HMHP + CH_3COCH_3$ | | | |
| $\quad + 0.2\,GLY + 0.05\,MGLY$ | | | |
| $APIN + NO_3 \rightarrow 0.74\,NO_2 + 0.26\,APINONO2$ | $1.2(-12)\exp(490/T)$ | 1 | N75 |
| $\quad + 1.3\,HCHO + CH_3COCH_3 + 0.2\,GLY + 0.05\,MGLY$ | | | |
| $APINOHO2 + NO \rightarrow 0.74\,NO_2 + 0.26\,APINONO2$ | $K_{\text{RO2NO}}$ | 1 | N76 |
| $APINOHO2 + NO_3 \rightarrow NO_2$ | $2.3(-12)$ | 1 | |
| $APINOHO2 + HO_2 \rightarrow$ products | $2.6(-13)\exp(1300/T)$ | 1 | |
| $APINO3O2 + NO \rightarrow 0.74\,NO_2 + 0.26\,APINONO2$ | $K_{\text{RO2NO}}$ | 1 | N76 |
| $APINO3O2 + NO_3 \rightarrow NO_2$ | $2.3(-12)$ | 1 | |
| $APINO3O2 + HO_2 \rightarrow$ products | $2.6(-13)\exp(1300/T)$ | 1 | |
| $APINONO2 + OH \rightarrow NO_2$ | $4.5(-12)$ | 1 | |

| Reaction | Rate | Ref. | Note |
|---|---|---|---|
| | *MBO oxidation* | | |
| $MBO + OH \rightarrow MBOO2$ | $8.1(-12)\exp(610/T)$ | 1 | |
| $MBO + O_3 \rightarrow 0.308\,HCHO + 0.992\,CH_3COCH_3 + 1.31\,HO_2$ | $1.0(-17)$ | 1 | N77 |
| $\quad + 0.01\,CH_3CHO + 0.89\,CO_2 + 0.168\,HMHP + 0.64\,CO$ | | | |
| $MBOO2 + NO \rightarrow MBONO3$ | $K_{RO2NO} \cdot Y_{nit}(T,M,7,2.4)$ | 1,34 | N78 |
| $MBOO2 + NO \rightarrow 0.67\,GLYALD + CH_3COCH_3 + HO_2$ | $K_{RO2NO} \cdot Y_{oxy}(T,M,7,2.4)$ | 1 | N78 |
| $\quad + 0.33\,HCHO + 0.33\,CO_2 + NO_2$ | | | |
| $MBOO2 + NO_3 \rightarrow 0.67\,GLYALD + CH_3COCH_3 + HO_2$ | $2.3(-12)$ | 1 | N78 |
| $\quad + 0.33\,HCHO + 0.33\,CO_2 + NO_2$ | | | |
| $MBOO2 + HO_2 \rightarrow 0.67\,CO + CH_3COCH_3 + 2\,HO_2 + 1.33\,CO_2$ | $2.3(-13)\exp(1300/T)$ | 1,3 | N79 |
| $MBONO3 + OH \rightarrow NO_2 + 0.67\,CO + 0.33\,CO_2$ | $2.0(-12)$ | 1 | N80 |
| $\quad + CH_3COCH_3 + 2\,HO_2$ | | | |

## 2.9 Notes to Table 2

N1. Rate equal to 90% of evaluation (Burkholder et al., 2015) to account for isoprene–OH segregation (Pugh et al., 2011). See Sect. 2.1.1 for main products. The minor addition channels (7%) include a hydroxyperoxy radical (ISOPEO2) as well as unsaturated carbonyls along with

$HO_2$. The unsaturated carbonyls are replaced by their major further oxidation products at high NO according to MCM (ACETO2 + HCHO + $HO_2$ + $CO_2$).

N2. See Sect. 2.2. The stabilized Criegee intermediate ($CH_2OO$) is currently not a model compound; its production is replaced by the products of its main atmospheric sink, the reaction with water dimer, namely $0.55\,HMHP + 0.4\,HCHO + 0.4\,H_2O_2 + 0.05\,HC(O)OH$ (Sheps et al., 2017).

N3. $Y^{nit}(T,M,n,Z)$ denotes the nitrate yield, as defined in Sect. 2.6. $Z$ is adjusted to match laboratory-based estimates at room conditions ($\sim$298 K and 1 atm): 14% and 13% for the 1,2- and 4,3-isoprene hydroxyperoxys, and 12% for the $\delta$-hydroxyperoxys (Wennberg et al., 2018). $Y^{oxy}(T,M,n,Z)$ (equal to $1 - Y^{nit}(T,M,n,Z)$) is the oxy radical channel branching ratio. The reaction products account for the relative proportions of $\beta$- and $\delta$-hydroxyperoxys (Sect. 2.1.3) as well as for the different organic nitrate yields in their reactions with NO.

N4. Bulk 1,6-H-shift reaction. See Sect. 2.1.1 for the rate, and Sect. 2.1.2 for the products.

N5. See Sect. 2.1.2 for details.

N6. Addition channels (Wennberg et al., 2018). The product yields account for the small contribution of the $\delta$-hydroxyperoxy pathways. The minor $\delta$-IEPOX compounds are lumped with $\beta$-IEPOX. The non-IEPOX products observed by St. Clair et al. (2016) in presence of NO (HYAC, GLYALD, HPAC, $CH_3CHO$) as well as the dihydroxy dihydroperoxides ($ISOP(OOH)_2$) proposed to be a potentially significant component of isoprene SOA in low-NOx conditions (Liu et al., 2016) are assumed to have a negligible yield in most atmospheric conditions

due to the proposed isomerisation of the peroxy radical formed in the reaction (D'Ambro et al., 2017). The further chemistry of the dihydroxy hydroperoxy epoxide resulting from this isomerisation, DHHEPOX, is not considered. Its saturation vapour pressure is estimated to be of the order of $3 \cdot 10^{-9}$ atm at 298 K using a group contribution method (Compernolle et al., 2011), i.e. three orders of magnitude lower than the

estimated vapour pressure of $\beta$-IEPOX ($3\cdot10^{-6}$ atm). The Henry's law constant (HLC) of DHHEPOX estimated as described in Müller et al. (2018) is equal to $\sim 3\cdot10^9$ M atm$^{-1}$ at 298 K, almost three orders above the estimated value for IEPOX. DHHEPOX is therefore very probably more soluble and prone to loss by deposition or SOA formation than IEPOX, which has been shown to deposit very rapidly on vegetation (Nguyen et al., 2015b) and to be a prominent SOA precursor (Surratt et al., 2010). Furthermore, the products of the oxidation

of DHHEPOX by OH (at a rate estimated at $\sim 2.1\cdot10^{-11}$ molec.$^{-1}$ cm$^3$ $s^{-1}$) are also expected to consist, for the most part, of highly oxygenated products prone to deposition and heterogeneous uptake.

N7. Abstraction of hydroperoxide-H (75%) and of hydroxy-$\alpha$-H (25%) (Wennberg et al., 2018). The latter leads to a radical proposed to undergo epoxide formation (Wennberg et al., 2018); we neglect this very minor and uncertain pathway as the product was suggested to be due to an impurity (St. Clair et al., 2016). Addition of $O_2$ to the radical forms $HO_2$ + $O{=}CHC(CH_3)(OOH)CH{=}CH_2$. The main fate of

10 the unsaturated hydroperoxy aldehyde is photolysis to an enol, $HOCH{=}C(CH_3)CH{=}CH_2$ (80%) or to $HCO + OH + MVK$ (20%) (see Sect. 2.1.4). The enol reacts primarily by OH addition to the first carbon, followed by a 1,5 H-shift to $OH + HC(O)OH + MVK$.

N8. Abstraction of hydroperoxide-H (60%) and of hydroxy-$\alpha$-H (40%), followed by similar reactions as for ISOPBOOH (see previous note). Hydroperoxy-$\alpha$-H abstraction is neglected.

N9. Assume fast reaction of MCM product with OH, followed by fast reaction with NO, neglecting side products.

N10. INBO2 is a mix of two peroxys (see Table 1). Assume 85% external and 15% internal OH-addition to ISOPBNO3.

N11. The rates of the 1,5 and 1,6 $\alpha$-hydroxy-H-shifts from the $C_1$ $HOCH_2$ group in the radicals $HOCH_2C(CH_3)(ONO_2)CH(O_2)CH_2OH$ and $HOCH_2C(CH_3)(ONO_2)CH(OH)CH_2O_2$, respectively, suggested by Wennberg et al. (2018) are assumed equal to 0.02 s$^{-1}$ at 298 K (instead of 0.05 s$^{-1}$ in Wennberg et al. (2018)), at the lower end of the range estimated by Møller et al. (2019) for $\alpha$-hydroxy H-shifts, given the unfavorable H-bonding between the peroxy group and the hydroxy-H of the other, $C_4$ or $C_3$ alcohol group. The nitroxyhydroxy

hydroperoxycarbonyls formed from the H-shift are assumed to photolyze rapidly, releasing HCO, $NO_2$ and a hydroxyhydroperoxy carbonyl (here, $CH_3C(O)CH(OOH)CH_2OH$ and $CH_3C(O)CH(OH)CH_2OOH$, respectively, or MVKOOH).

N12. Assume fast hydrolysis of the dinitrate in the aqueous aerosol phase, as it bears a tertiary nitrate group. The hydrolysis product (besides $HNO_3$) is very soluble and can be assumed to remain in the particulate phase.

N13. The hydroperoxide bears a tertiary nitrate group and asumed to undergo hydrolysis in the aerosol phase. The hydrolysis product (besides

$HNO_3$) is assumed to remain in the aerosol phase.

N14. As for INBO2 (see Note N11), the 1.5 $\alpha$-hydroxy-H-shift in the peroxy $HOCH_2C(O_2)(CH_3)CH(ONO_2)CH_2OH$ is assumed to be 2.5 times slower compared to Wennberg et al. (2018). The nitroxyhydroxy hydroperoxycarbonyls formed from the H-shift are assumed to photolyze rapidly, releasing HCO, $NO_2$ and a hydroxyhydroperoxy carbonyl ($HOCH_2C(OOH)(CH_3)CHO$). The latter compound photolyzes also very rapidly, to $HCO + OH + HYAC$.

N15. The hydroperoxy aldehyde ($O{=}CHC(CH_3)(OOH)CH(ONO_2)CH_2OH$ or INDHPCHO in MCM) formed in the reaction is assumed to photolyze rapidly to $HCO + OH + CH_3C(O)CH(ONO_2)CH_2OH$.

N16. The *trans* and *cis* isomers are lumped, adopting the *trans* to *cis* ratio (2:1) of Bates et al. (2016). The epoxide-retaining products are lumped into ICHE.

N17. Formyl-H abstraction from the carbonyl hydroxyepoxides (e.g. $HOCH_2\overline{CHOC}(CH_3)CHO$ and isomers) primarily formed from

35 IEPOX + OH. The isomer distribution follows Wennberg et al. (2018). H-abstraction is followed by concerted CO elimination and ring

opening, O$_2$-addition leading to CH$_3$C(O)CH(O$_2$)CH$_2$OH (for the major isomer) and OCHC(O$_2$)(CH$_3$)CH$_2$OH (minor) which undergoes a 1,4 aldehyde H-shift, to CO + OH + HYAC.

N18. Hydroxyl-$\alpha$-H abstraction from the carbonyl hydroxyepoxides (see previous note), at a rate taken equal to half the OH-reaction rate constant of $\beta$-IEPOX. It is followed by ring opening to give (for the main isomer) OCHC(CH$_3$)(O$^\circ$)CH=CHOH, followed by 1,5 enol-H shift and O$_2$-addition to form OCHC(CH$_3$)(OH)CH(O$_2$)CHO. This is followed by a fast 1,5 aldehydic-H shift and (for a large part) by CO elimination to give, after O$_2$-addition, CH$_3$C(O)CH(OOH)CHO + HO$_2$.

N19. The 1,4 H-shift in HOCH$_2$C(OH)(CH$_3$)CH(O2)CHO and its isomer is taken to be fast (0.5 s$^{-1}$ at 298 K), following Wennberg et al. (2018).

N20. The 1,5 H-shift in HOCH$_2$CH(OH)C(CH$_3$)(O2)CHO forms HO$_2$ + O=CHC(OOH)(CH$_3$)CH(OH)CHO assumed to photolyze rapidly either to CHO + OH + CH$_3$C(O)CH(OH)CHO (HOBA), or to CHO + HO$_2$ + OCHC(OOH)(CH3)CHO (HPDIAL).

N21. Oxy radical channel (65%) (Wennberg et al., 2018).

N22. The hydroperoxide channel (35%) forms O=CHC(OOH)(CH$_3$)CH(OH)CH$_2$OH, assumed to photolyze very rapidly to HCO + OH + CH$_3$C(O)CH(OH)CH$_2$OH.

N23. The 1,5 H-shift in HOCH$_2$C(OH)(CH$_3$)CH(O2)CHO forms HO$_2$ + O=CHC(OH)(CH$_3$)CH(OOH)CHO assuming to photolyze rapidly either to CHO + OH + OCHC(CH$_3$)(OH)CHO, or to CHO + HO$_2$ + CH$_3$C(O)CH(OOH)CHO (HPKETAL). The hydroxy-dialdehyde is assumed to react exclusively with OH, forming CO + MGLY + HO$_2$.

N23. The hydroperoxide channel (35%) forms O=CHCH(OOH)C(OH)(CH$_3$)CH$_2$OH, assumed to photolyze very rapidly to HCO + OH + O=CHC(OH)(CH$_3$)CH$_2$OH.

N25. Neglect hydroperoxide channel, i.e. assume formation of oxy radical + OH. Note that if the hydroperoxide is formed, it is expected to photolyze rapidly (Liu et al., 2018), for a large part to the same products as the oxy radical pathway.

N26. Based on D'Ambro et al. (2017), the main OH-addition channel forms a hydroxyperoxy of which the main fate in low-NO regions should be reaction with HO$_2$, followed by reaction of the hydroperoxide with OH, forming HOCH$_2$CH(OH)C(CH$_3$)(OOH)CHO as main product (C75OOH in MCM). Note that isomerisation of the hydroperoxy forms also C75OOH (along with HO$_2$). C57OOH is a $\alpha$-hydroperoxyaldehyde, assumed to photolyze rapidly (Liu et al., 2018) to HCO+OH+CH$_3$C(O)CH(OH)CH$_2$OH, therefore regenerating OH and HO$_2$.

N27. The branching ratios are from Peeters and Müller (2010). The further mechanism mostly follows Wennberg et al. (2018); however, collisional deactivation of the radical (OCHC(CH$_3$)C$^\circ$CH$_2$(OOH)) formed in the minor OH-addition channel is neglected, since epoxide formation should be largely dominant, as for the radical formed by OH-addition to ISOPOOH, for which epoxide formation constitutes ca. 90% of the sink. The unsaturated dialdehyde O=CHC(CH$_3$)=CHCH(O) (MBED) undergoes very fast photolysis and is replaced by its oxidation products, as described in Sect. 2.1.5.

N28. Branching ratios from Peeters and Müller (2010), further mechanism from Wennberg et al. (2018), except for the collisional stabilisation of the radical formed in the major addition channel, which is neglected (see previous note). As above, the unsaturated dialdehyde O=CHC(CH$_3$)=CHCH(O) should photolyze rapidly to compounds replaced by their further reaction products. The hydroxyhydroperoxy aldehyde HOOCH$_2$C(CH$_3$)(OH)CH=O should photolyze rapidly to (and is therefore replaced by) HCO + HO$_2$ + CH$_3$C(O)CH$_2$OOH (HPACET).

N29. The peroxy radical ($CH_3C(O)CH(OH)C(O)O_2$) formed in the reaction is replaced by its further oxidation products in presence of NO.

N30. H-abstraction from CH group leads to $CH_3C(O)C(O)CH_2OOH$ which can be assumed to photolyze very rapidly to $OH + CH_3C(O)O_2$ + HCHO + CO. H-abstraction of the $CH_2$ group yields $CH_3C(O)CH(OOH)CHO$ (HPKETAL).

N31. The acyl radical formed from $CH_3C(O)CH(OOH)CHO$ through aldehydic H-abstraction can add $O_2$ to form an acylperoxy radical which (upon reaction with NO) leads to $CO_2 + OH + MGLY$. Note that the acyl radical can also decompose to $CO + OH + MGLY$. Abstraction of the hydroperoxide H is followed by a 1,4 H-shift of the peroxy radical $CH_3C(O)CH(O_2)CHO$ to the same acyl radical as above. H-abstraction from the carbon bearing the OOH group (40% of reactivity) leads to $CH_3C(O)C(O)CHO$ assumed to photolyze rapidly to $CH_3CO + CO + HCO$.

N32. The acyl radical formed from $OCHC(CH_3)(OOH)CHO$ can add $O_2$ to form an acylperoxy radical which (upon reaction with NO) leads to $CO_2 + OH + MGLY$. Note that the acyl radical can also decompose to $CO + OH + MGLY$.

N33. NISOPO2 is a mix of several radicals (Schwantes et al., 2015; Wennberg et al., 2018). The dinitrate formed in the reaction is ignored, as its further chemistry is unclear.

N34. See Sect. 2.3. A higher self-reaction rate was used by Schwantes et al. (2015) in their kinetic modelling, but there is suggestion that it might be overestimated (Schwantes et al., 2015).

N35. H-abstraction from $HOOCH_2CH=C(CH_3)CH_2ONO_2$ and isomer.

N36. OH-addition to $HOOCH_2CH=C(CH_3)CH_2ONO_2$ (for 84%) and isomer (16%). The mechanism follows Wennberg et al. (2018), except that 1) the 1,5-H shift in the peroxy $O_2NOCH_2C(O_2)(CH_3)CH(OH)CH_2OH$ (and isomer) formed in the reaction is neglected, as it should be slow due to stabilization by H-bonding between the peroxy and hydroxy groups, 2) epoxide formation (ca. 9% yield) is neglected, 3) the minor pathways in the bimolecular reactions of the hydroxyperoxy radicals (e.g. dinitrate formation in $RO_2+NO$ and dihydroperoxide formation in $RO_2+HO_2$, also the minor oxy decomposition channel proposed by Wennberg et al.) are neglected since their yields are small and uncertain, 4) the peroxys are replaced by the products of their reactions with NO or $HO_2$, and 5) the nitroxy hydroperoxy aldehyde $OCH-C(CH_3)(OOH)CH_2ONO_2$ is assumed to photolyze rapidly (Liu et al., 2018) to $CHO + OH + CH_3C(O)CH_2ONO_2$.

N37. The minor products C3CNO2 and C3CPO2 are replaced by assumed further oxidation product (NOA). The nitrooxy hydroperoxy epoxide (IHPE) formed in the reaction (Schwantes et al., 2015) is neglected and the other yields are increased for carbon balance.

N38. H-abstraction from $CH_2=CHC(CH_3)(OOH)CH_2ONO_2$ and isomer.

N39. OH-addition to $CH_2=CHC(CH_3)(OOH)CH_2ONO_2$ and isomer. The mechanism follows Wennberg et al. (2018), with simplications similar to the case of the $\delta$-hydroperoxynitrates (see Note N36). The peroxy radical $O_2NOCH_2C(CH_3)(OOH)CH(OH)CH_2O_2$ (INPHO2$\beta$ in Schwantes et al. (2015)) is assumed to react fast with NO or $NO_3$, leading to $O_2NOCH_2C(CH_3)(OOH)CHO$ (C4CPNA in Schwantes et al.) assumed to photolyze rapidly (Liu et al., 2018) to $CHO + OH + NOA$.

N40. IHNE is a mix of two $\beta$- and two $\delta$-nitroxy hydroxyepoxides. The mechanism follows Wennberg et al. (2018). The peroxy radicals $O_2NOCH_2C(OH)(CH_3)C(O)CH_2O_2$ and $HOCH_2C(O_2)(CH_3)CH_2ONO_2$ formed from the $\beta$-IHNE are replaced by the products of their reaction with NO, neglecting dinitrate formation and minor oxy decomposition products. The radical $O=C°CH_2ONO_2$ formed in these reactions adds $O_2$, forming an acylperoxy radical replaced by its further reaction product in presence of NO, i.e. $CO_2$ + HCHO + $NO_2$. The peroxy $O_2NOC(OH)(CH_3)CH(O_2)CHO$ undergoes a fast 1,4 H-shift outrunning bimolecular reactions, forming

CO + OH + $O_2NOCH_2C(OH)(CH_3)CHO$, which is assumed to photolyze rapidly to $NO_2$ + HCHO + MGLY + $HO_2$ (Müller et al., 2014). The carbonyl nitroxyepoxides (ICNE in Wennberg et al.) are assumed to react with OH, following the Caltech reduced mechanism: ICNE + OH → 2 CO + 0.35 NOA + 0.65 MGLY + 0.65 $HO_2$ + 0.65 $NO_2$. The peroxys $O_2NOCH_2C(OH)(CH_3)CH(O_2)CHO$ and $OCHC(O_2)(CH_3)CH(OH)CH_2ONO_2$ formed from the $\delta$-IHNE undergo fast H-shift reactions outrunning the bimolecular reactions, forming CO + OH + either $O_2NOC(OH)(CH_3)CH(O_2)CHO$ (in the first case) or $CH_3C(O)CH(OH)CH_2ONO_2$ (second case) (Wennberg et al., 2018).

N41. The OH-reaction rate was measured by Xiong et al. (2016) for $OCHC(CH_3)=CHCH_2ONO_2$. The yields account for the NC4CHO isomer distribution estimated by Schwantes et al. (2015). The OH-reaction essentially follows Wennberg et al. (2018). Aldehyde H-abstraction from $OCHCH=C(CH_3)CH_2ONO_2$ by either OH or $NO_3$ leads to an acylperoxy radical here replaced by its NO-reaction product according to MCM ($CO_2$ + CO + $HO_2$ + NOA). Note that alternative reaction pathways proposed by Wennberg et al. also lead eventually to CO + NOA. OH-addition generates peroxy radicals undergoing fast isomerisation (Schwantes et al., 2015) leading to the nitroxy hydroxy aldehyde $O_2NOCH_2C(OH)(CH_3)CHO$ assumed to photolyze rapidly to $NO_2$ + HCHO + $HO_2$ + MGLY; the nitrooxy hydroperoxyaldehyde $O_2NOCH_2C(CH_3)(OOH)CHO$ assumed to photolyze rapidly to HCO + OH + NOA; and the nitrooxy hydroperoxyketone $CH_3C(O)CH(OOH)CH_2ONO_2$ assumed to photolyze to $CH_3CO$ + OH + $OCHCH_2ONO_2$ (ETHLN).

N42. Abstraction of $\alpha$-hydroxy H in ISOPCNO3 ($HOCH_2CH=C(CH_3)CH_2ONO_2$) and ISOPANO3 ($HOCH_2C(CH_3)=CHCH_2ONO_2$) Wennberg et al. (2018), leading in part to photolabile hydroperoxynitroxy carbonyls (e.g. $O_2NOCH_2C(OOH)(CH_3)CHO$) assumed to photolyze rapidly (to either HCO + OH + NOA for ISOPCNO3, or $CH_3CO_3$ + OH + ETHLN for ISOPANO3).

N43. OH-addition to ISOPCNO3 ($HOCH_2CH=C(CH_3)CH_2ONO_2$ and ISOPANO3 ($HOCH_2C(CH_3)=CHCH_2ONO_2$). The mechanism follows Wennberg et al. (2018), except that two different dihydroxy nitroxyperoxy radicals are lumped into one radical (INCO2 or INAO2). In each case, only one of the two peroxy isomers undergoes an 1,5-H-shift. For simplicity, and since the H-shift dominates largely the fate of the peroxy undegoing it, the bimolecular reactions are the reactions of the isomer which does not undergo the H-shift.

N44. INCO2 includes two isomers, only one of which ($O_2NOCH_2C(O_2)(CH_3)CH(OH)CH_2OH$) undergoes an 1.5 H-shift. It leads to $HO_2$ + $O_2NOCH_2C(OOH)(CH_3)CH(OH)CHO$, assumed to be rapidly followed by fast photolysis (Liu et al., 2018) to CHO + $HO_2$ + $O_2NOCH_2C(OOH)(CH_3)CHO$, itself followed by photolysis to CHO + OH + $CH_3C(O)CH_2ONO_2$ (NOA).

N45. Mechanism adapted from Wennberg et al. (2018). The hydroperoxide $HOCH_2CH(OOH)C(OH)(CH_3)CH_2ONO_2$ formed with a 43 % yield is assumed to react with OH, primarily by $\alpha$-hydroperoxide-H abstraction, forming OH + $HOCH_2C(O)C(OH)(CH_3)CH_2ONO_2$ (INCCO), and by abstraction of the terminal hydroperoxide hydrogen to regenerate INCO2.

N46. The dicarbonyl nitrate $O_2NOCH_2C(CH_3)(OH)C(O)CHO$ formed in the reaction is assumed to photolyze rapidly to HCO + $O_2NOCH_2C(CH_3)(OH)-C°=O$, which decomposes (for a large part) into CO + $HO_2$ + $O_2NOCH_2C(O)CH_3$ (NOA).

N47. The mechanism follows the MCM. Among the three considered channels, formation of $O_2NOCH(CHO)C(CH_3)(OH)CH_2ONO_2$ + $HO_2$ is assumed to be followed by photolysis of the carbonyl dinitrate to $NO_2$ + GLY + NOA + $HO_2$ (Müller et al., 2014).

N48. INAO2 includes two peroxy isomers. The minor peroxy $HOCH_2C(OH)(CH_3)CH(O_2)CH_2ONO_2$ can undergo an 1,5 $\alpha$-hydroxy-H-shift leading to $HO_2$ + $OCHC(OH)(CH_3)CH(OOH)CH_2ONO_2$ (Wennberg et al., 2018), which is assumed to photolyze rapidly (Liu et al., 2018) to CHO + $HO_2$ + $CH_3C(O)CH(OOH)CH_2ONO_2$, itself followed by photolysis to $CH_3CO$ + OH + $OCHCH_2ONO_2$ (ETHLN).

N49. Adapted from Wennberg et al. (2018). The hydroperoxide product (50% yield, $HOCH_2C(CH_3)(OOH)CH(OH)CH_2ONO_2$) is assumed to react with OH, following the mechanism of the MCM and leading in part to $O=CHC(CH_3)(OOH)CH(OH)CH_2ONO_2$ which is assumed to photolyze rapidly to give $CHO + OH + CH_3C(O)CH(OH)CH_2ONO_2$.

N50. The aldehyde-H-abstraction channel yields $HOCH_2CH=C(CH_3)C(O)O_2$ or $HOCH_2C(CH_3)=CHC(O)O_2$ that should isomerize by 1,6 H-shifts of an $\alpha$-hydroxy-H to form the doubly resonance-stabilized radicals $Z$-$HOC°H-CH=C(CH_3)-C(O)OOH$ (Case I) or $Z$-$HOC°H-C(CH_3)=CH-C(O)OOH$ (Case II). As for the similar 1,6 H-shifts in the initial $Z$-$\delta$-OH-peroxys (see Sect. 2.1.2), the product radicals are expected to arise in both the $Z, Z'$ and $Z, E'$ forms, here assumed in a 50:50 ratio. The expected $O_2$-addition-energy to these doubly resonance-stabilized radicals is as low as 15 kcal mol$^{-1}$, such that $O_2$-addition $\alpha$ to the OH-group on $C_1$ (or $C_4$) is likely to result in $O_2$-loss instead of concerted elimination of $HO_2$, whereas $O_2$-addition at the $\gamma$ position leads for 50% to $Z, Z'$-peroxys that undergo fast 1,6 enol-H-shifts facing barriers of only 10 kcal mol$^{-1}$, similar to the H-shifts leading to DIHPCARPs (Peeters et al., 2014). The product radical of these H-shifts adds $O_2$ to form DIHPCARP analogues that may readily isomerize by aldehyde-H-shift, promoted by H-bonding. The resulting radicals are assumed to eliminate CO and OH to yield $OCHC(CH_3)(OOH)C(O)OOH$ or $CH_3C(O)CH(OOH)C(O)OOH$, which are expected to photolyze rapidly (Liu et al., 2018) into $CO + HO_2 + OH + CH_3C(O)C(O)OOH$ or $CH_3CO_3 + OH + OCHC(O)OOH$, respectively. Pyruvic peracid photolyzes radidly into $CH_3CO + CO_2 + OH$, while its reaction with OH is very slow (Saunders et al., 2003). Peroxy glyoxylic acid (PGA) is considered explicitly. The 50% $Z, E'$-peroxys that also arise by $\gamma$ $O_2$-addition can react quasi-exclusively with NO and $HO_2$, here assumed in a 50:50 ratio, to form mainly oxy radicals (e.g. $Z, E'$-$HOCH=CHC(CH_3)(O°)C(O)OOH$) that quickly decompose into $CO_2 + OH +$ either $CH_3C(O)CH=CH_2OH$ (HMVK) or $OCHC(CH_3)=CH_2OH$ (HMAC).

N51. OH-addition channel, with rates from Neeb (2000); Peeters et al. (2004). For OH-addition $\beta$ to the formyl, we follow Wennberg et al. (2018), with product radicals IEPOXAO2 and IEPOXBO2 identical to those resulting from $\beta$-IEPOX + OH. The peroxys from OH-addition $\alpha$ to the formyl are unlikely to undergo 1,5 aldehyde-H-shifts due to unfavorable expected H-bonding pattern, but should rather react with NO or $HO_2$, to yield mainly GLYALD + MGLY + $HO_2$ for HALD1 or HYAC + GLY + $HO_2$ for HALD2 (Peeters et al., 2004).

N52. Account for the fast isomerisations of the hydroxyperoxys resulting from OH addition to MACR (Crounse et al., 2012; Wennberg et al., 2018).

N53. Rate from MCM. The reactions occurs by $\alpha$-hydroxy-H abstraction, after which the 3-ring opens to form the 10-15 kcal mol$^{-1}$ more stable $HOCH=C(CH_3)-C(O)O°$, the latter stabilized by acyloxy resonance. Direct elimination of $CO_2$ as proposed in the MCM appears not likely, since the $C_1=C_2--C_3$ bond is $\sim$10 kcal mol$^{-1}$ stronger than in $CH_3--C(O)O°$ due to the neighbouring double bond. The most likely fate is a 1,5 enol-H shift to $O=CHC°(CH_3)C(=O)OH$ (with double "vinoxy" resonance-stabilization), exothermic for some 25–30 kcal mol$^{-1}$, and almost barrierless. After adding $O_2$, one can expect a 1,4 aldehyde-H-shift followed by CO elimination (barrier $\sim$7 kcal mol$^{-1}$) and OH loss to yield pyruvic acid. The latter is replaced by its photolysis products (Burkholder et al., 2015), i.e. 0.39 $HO_2$ + 0.48 $CH_3CHO$ + 0.87 $CO_2$ + 0.44 $CH_3C(O)O_2$ + 0.08 $CH_3C(O)OH$ + 0.13 CO + 0.05 OH.

N54. See Note N2 regarding the stabilized Criegee intermediate ($CH_2OO$). Pyruvic acid is replaced by its photolysis products (see previous Note).

N55. MVKO2 is a mix of $CH_3C(O)CH(O_2)CH_2OH$ (72%) and $CH_3C(O)CH(OH)CH_2O_2$ (28%). The ratio is adjusted so that the glycolaldehyde yield in MVKO2 + NO is 69% (Galloway et al., 2011), taking the nitrate yield (4%) (Praske et al., 2015) into account.

N56. MVKOOH is a mix of $CH_3C(O)CH(OOH)CH_2OH$ (55%) and $CH_3C(O)CH(OH)CH_2OOH$ (45%). The fractions account for the different hydroperoxide yields in the reaction of their respective peroxy radical precursors with $HO_2$.

N57. Reaction rate taken equal to the average of the MCM and the structure activity relationship (SAR) of Neeb (2000). Assume 50% formyl-H absraction and 50% alcoholic-H absraction. The former leads ultimately to hydroxyacetone + $NO_2$ (in presence of NO). The latter leads to a nitrooxydialdehyde assumed to photolyze immediately into methylglyoxal, $NO_2$ and HCO.

N58. The reaction MVKNO3 + OH is split into two reactions since MVKNO3 represents two isomers, $CH_3C(O)CH(ONO_2)CH_2OH$ (for 80%) and $CH_3C(O)CH(OH)CH_2(ONO_2)$ (for 20%). For the first, assume 50% alcoholic-H abstraction to $CH_3C(O)CH(ONO_2)CHO$ assumed to photolyze (for ca. 80%) into $NO_2$ + GLY + $CH_3CO$, the rest reacting with OH to form eventually MGLY+$HO_2$+$CO_2$ (in the presence of NO). For the second compound, ignore alcoholic-H absraction.

N59. Assume fast reaction of the acylperoxy radical (84% of reactive flux) with NO. Assume fast photolysis of $CH_3C(O)C(O)CHO$ (16% of flux) into $CH_3CO$ + CO + HCO.

N60. Assume immediate reaction of product $OCHC(CH_3)(OH)CHO$ with OH, forming MGLY + $HO_2$ + $CO_2$ upon reaction with NO.

N61. The dominant OH-addition, to $(HO)_2CHCH(O_2)C(O)CH_3$, is followed by a 1,5 H-shift from an alcohol-H to the peroxy group and decomposition (So et al., 2014). The minor addition channel forms $HOC°HCH(OH)C(O)CH_3$ which reacts with $O_2$ to $HO_2$ + $CH_3C(O)CH(OH)CHO$.

N62. The dominant OH-addition ($3\cdot10^{-11}$ $molec^{-1}$ $cm^3$ $s^{-1}$), to $O=CHC(CH_3)(O_2)CH(OH)_2$, is followed by an H-shift from either an alcohol-H (50%) or from the aldehyde-H (50%) to the peroxy group, leading to either HC(O)OH + OH + MGLY or CO +OH + $CH_3C(O)CH(OH)_2$ (DHA).

N63. Combines the minor addition channel ($1.2\cdot10^{-11}$ $molec^{-1}$ $cm^3$ $s^{-1}$) and the aldehyde-H abstraction channel ($1.5\cdot10^{-11}$ $molec^{-1}$ $cm^3$ $s^{-1}$). The minor addition channel leads to $HO_2$ + $O=CHC(CH_3)(OH)CH=O$, which reacts primarily with OH, leading to an acyl radical which can eliminate CO and give MGLY + $HO_2$ or form an acylperoxy radical which can undergo a shift of the aldehyde-H to the peroxy group. The resulting radical can either lose CO, and upon reaction with $O_2$, form $HO_2$ + CO + $CH_3C(O)C(O)OOH$ (PPYR), or react with $O_2$ and then with NO or $HO_2$, forming $CO_2$ + $HO_2$ + PPYR. The H-abstraction channel leads to an acylperoxy radical, $O=C(O_2)C(CH_3)=CHOH$, which undergoes a enol 1,6 H-shift followed by $O_2$-addition, to $O=C(OOH)C(O_2)(CH_3)CH=O$. The latter radical undergoes a 1,4 H-shift of the aldehyde-H, leading to CO + OH + PPYR. PPYR is assumed to photolyze rapidly to $CH_3CO$ + $CO_2$ + OH (Saunders et al., 2003).

N64. The nitrate yield is 1.3% at room conditions (298 K, 1 atm).

N65. Assume equal rates for the two addition channels. See Sect. 2.1.4.

N66. The reaction leads to pyruvic acid (along with $HO_2$), assumed to photolyze very rapidly according to Burkholder et al. (2015).

N67. Yields calculated at room conditions. The acylperoxy radical resulting from $O_2$ addition to the HCOCO radical (ca. 17% of the reactive flux) is replaced by the final reaction products in presence of NO and $O_2$ (i.e. CO + $HO_2$ + $CO_2$).

N68. Contrary to MCM, consider aldehyde-H abstraction, leading in part to CO + OH + HCHO (for 25%) and in part to $HOOCH_2CO_3$ (75%) which (upon reaction with NO) leads to $CO_2$ + OH + HCHO.

N69. Reaction rate taken equal to the average of the MCM and the structure activity relationship (SAR) of Neeb (2000). Products assume fast reaction of peroxy radical with NO.

N70. The minor channel (8%, formation of $CH(OH)_2CH_2O_2$) proposed by So et al. (2014) is neglected.

N71. The methyl nitrate yield adopted here is $2\cdot10^{-4}$ at 298 K and 1 atm, or ca. $5\cdot10^{-5}$ in the lower stratosphere, at the lower end of the range $((5\text{-}10)\cdot10^{-5})$ estimated by Flocke et al. (1998) based on stratospheric $CH_3ONO_2$ observations.

N72. See Sect. 2.7 for details.

N73. The water dimer concentration (molec.cm$^{-3}$) is calculated using

$$[\text{dimer}] = p \cdot K_p \cdot [H_2O]^2 / [M] \tag{11}$$

where $p$ is atmospheric pressure (atm), $[H_2O]$ and M are the water vapour and dry air number density (molec.cm$^{-3}$), and $K_p$ (atm$^{-1}$) is approximated following Scribano et al. (2006) :

$$K_p = 4.7856 \cdot 10^{-4} \exp(1851.09/T - 5.10485 \cdot 10^{-3} T) \tag{12}$$

N74. Rate reported by Wennberg et al. (2018). H-abstraction from hydroperoxide group, followed by decomposition of the hydroxymethylperoxy radical, is slightly dominant (Allen et al., 2018). H-abstraction from the carbon is followed by OH expulsion.

N75. The rate constant is for $\alpha$-pinene although the compound APIN is a surrogate for all monoterpenes. For the products, see Section 2.4.

N76. The 26% yield is the assumed overall organic nitrate formation from monoterpenes (Rindelaub et al., 2015).

N77. Several carbonyl intermediates formed in the reaction are assumed to react rapidly with OH. $CH_3C(OH)(CH_3)C(O)O_2$ is assumed to react with NO, forming $CO_2 + CH_3C(O)CH_3 + HO_2$.

N78. The organic nitrate yield is $\sim$10% at room conditions (295 K and 1 atm) (Chan et al., 2009). Whereas the major isomer peroxy radical leads to $CH_3C(O)CH_3 + GLYALD + HO_2$ upon reaction with NO, the other isomer leads to $HCHO + HO_2 + CH_3C(OH)(CH_3)CHO$ which is here replaced by its OH-reaction product in presence of NO, namely $CO_2 + CH_3C(O)CH_3 + HO_2$. Note that the MCMv3.3.1 mechanism for MBO was recently validated by comparisons with chamber measurements, in particular regarding the production of radicals, acetone and formaldehyde (Novelli et al., 2018a), and that the peroxy radical isomerisation reactions proposed by Knap et al. (2015) can be neglected due to their low rates and resulting impacts.

N79. The hydroperoxides formed in the reaction are replaced by the OH-reaction products in presence of NO.

N80. Average reactivity of the two isomer dihydroxynitrates. The products are replaced by their OH-reaction products in presence of NO.

## 2.10 Photodissociations

The photolysis reactions are listed in Table 3. In many cases, the photolysis parameters are directly obtained from experimental studies, or can be assumed identical to the parameters for other, similar compounds (e.g. the absorption cross sections of many organic hydroperoxides are assumed identical to those of $CH_3OOH$). For nitrooxycarbonyls and for hydroperoxycarbonyls, however, analysis of the (scarce) available laboratory data indicates that the interaction between the two chromophores has a strong influence on the reaction mechanism and on the photodissociation parameters (Müller et al., 2014; Liu et al., 2018). The absorption cross sections for these classes (Fig. 3) are calculated based on available cross section data for structurally similar monofunctional compounds and on wavelength-dependent enhancement factors derived for nitrooxycarbonyls (Müller et al., 2014) and for hydroperoxycarbonyls (Liu et al., 2018) based on available laboratory data.

**Table 3.** Photodissocation reactions. The last column gives the photorate ($J$) calculated using the TUV model (Madronich, 1993) for a zenith angle of $30°$ and 300 DU ozone. References: 1, Burkholder et al. (2015); 2, Röth and Ehhalt (2015); 3, Shaw et al. (2018); 4, Pinho et al. (2005); 5, Jenkin et al. (2015); 6, Atkinson et al. (2006); 7, Liu et al. (2018); 8, Müller et al. (2014); 9, Barnes et al. (1993); 10, Xiong et al. (2016); 11, Liu et al. (2017); 12, Nakanishi et al. (1977); 13, Back and Yamamoto (1985).

| Reaction | Cross section | Quantum yield | Products | $J$ (s$^{-1}$) |
|---|---|---|---|---|
| $HCHO \rightarrow CO + 2HO_2$ | 1 | 2 | | 3.4(-5) |
| $HCHO \rightarrow H_2 + CO$ | 1 | 2 | | 5.2(-5) |
| $CH_3CHO \rightarrow CH_3O_2 + CO + HO_2$ | 1 | 1 | | 5.0(-6) |
| $CH_3CHO \rightarrow VA$ | 1 | 3 | | 1.7(-6) |
| $GLYALD \xrightarrow{83\%} HCHO + CO + 2HO_2$ | 1 | 1 | | 1.2(-5) |
| $\xrightarrow{10\%} CH_3OH + CO$ | | | | |
| $\xrightarrow{7\%} OH + OCHCH_2O_2$ | | | | |
| $GLY \rightarrow 2CO + 2HO_2$ | 1 | 1 | | 7.6(-5) |
| $GLY \rightarrow 2CO + H_2$ | 1 | 1 | | 1.6(-5) |
| $GLY \rightarrow HCHO + CO$ | 1 | 1 | | 3.1(-5) |
| $CH_3COCH_3 \rightarrow CH_3CO_3 + CH_3O_2$ | 1 | 1 | | 5.5(-7) |
| $MGLY \rightarrow CH_3CO_3 + CO + HO_2$ | 1 | 1 | | 1.4(-4) |
| $MACR \xrightarrow{50\%} MCO3 + HO_2$ | 1 | 4[a] | 5 | 2.1(-6) |
| $\xrightarrow{50\%} 0.35\,CH_3CO_3 + HCHO + 1.65\,CO + 0.65\,CH_3O_2 + HO_2$ | | | | |
| $MVK \xrightarrow{50\%} C_3H_6 + CO$ | 1 | 1 | 5 | 4.5(-6) |
| $\xrightarrow{50\%} CH_3CO_3 + HCHO + CO + HO_2$ | | | | |
| $CH_3OOH \rightarrow HCHO + HO_2 + OH$ | 1 | 1[b] | | 5.6(-6) |
| $HMHP \rightarrow HCOOH + OH + HO_2$ | 1 | b | | 4.8(-6) |
| $ISOPBOOH \rightarrow MVK + HCHO + HO_2 + OH$ | 1[c] | b | 5 | 5.6(-6) |
| $ISOPDOOH \rightarrow MACR + HCHO + HO_2 + OH$ | 1[c] | b | 5 | 5.6(-6) |
| $ISOPEOOH \rightarrow MACR + HCHO + HO_2 + OH$ | 1[c] | b | 5 | 5.6(-6) |
| $MACROH \rightarrow HYAC + CO + 2HO_2$ | 6[d] | 6[d] | 5 | 6.2(-5) |
| $MVKOOH \xrightarrow{45\%} CH_3CO_3 + HO_2 + HPAC$ | 7 | 7[e] | 5[f] | 1.3(-4) |
| $\xrightarrow{55\%} CH_3CO_3 + GLYALD + OH$ | | | | |
| $CH_3ONO_2 \rightarrow HCHO + HO_2 + NO_2$ | 1 | 1[b] | | 9.0(-7) |
| $PAN \xrightarrow{70\%} CH_3CO_3 + NO_2$ | 1 | 1[b] | | 7.3(-7) |
| $\xrightarrow{30\%} CH_3O_2 + CO_2 + NO_3$ | | | | |
| $PAA \rightarrow CH_3O_2 + OH + CO_2$ | 1 | b | 5 | 7.9(-7) |
| $HYAC \xrightarrow{50\%} CH_3CO_3 + HCHO + HO_2$ | 1 | 1 | 1 | 1.9(-6) |

| Reaction | Cross section | Quantum yield | Products | $J$ (s$^{-1}$) |
|---|---|---|---|---|
| $\xrightarrow{20\%}$ GCO3 + CH$_3$O$_2$ | | | | |
| $\xrightarrow{15\%}$ CH$_3$O$_2$ + CO + HCHO + HO$_2$ | | | | |
| $\xrightarrow{15\%}$ OH + ACETO2 | | | | |
| INDOOH $\rightarrow$ NO$_2$ + GLYALD + HYAC + OH | $6^g$ | $b$ | $h$ | 2.9(-6) |
| INDOOH $\rightarrow$ OH + 0.15 (HYAC + GLYALD + NO$_2$) | $1^c$ | $b$ | $i$ | 5.6(-6) |
| $\quad\quad$ +0.85 (HCHO + HO$_2$ + MVKNO3) | | | | |
| MACRNO3 $\rightarrow$ HYAC + CO + HO$_2$ + NO$_2$ | 8 | $8^b$ | 8 | 3.6(-4) |
| MVKNO3 $\rightarrow$ 0.8 (CH$_3$CO$_3$ + GLYALD + NO$_2$) | 8 | $8^b$ | 5 | 5.7(-5) |
| $\quad\quad$ +0.2 (MGLY + HCHO + NO$_2$) | | | | |
| INCCO $\rightarrow$ NO$_2$ + HYAC + GCO3 | $6^j$ | $8^b$ | 5 | 1.4(-5) |
| INCNO3 $\rightarrow$ NO$_2$ + HCHO + HO$_2$ + MVKNO3 | $6^k$ | $b$ | $h$ | 1.9(-6) |
| INCNO3 $\rightarrow$ NO$_2$ + GLYALD + NOA + HO$_2$ | $6^g$ | $b$ | $h$ | 2.9(-6) |
| NOA $\rightarrow$ CH$_3$CO$_3$ + HCHO + NO$_2$ | 9 | 8 | 5 | 3.2(-5) |
| ETHLN $\rightarrow$ HCHO + CO + HO$_2$ + NO$_2$ | 8 | 8 | 8 | 1.7(-4) |
| NC4CHO $\xrightarrow{16\%}$ NO$_2$ + 1.15 HO$_2$ + 1.35 CO$_2$ + 0.55 HCHO | 10 | $10^l$ | $5^m$ | 3.9(-4) |
| $\quad\quad$ +0.65 CH$_3$CO$_3$ + 0.2 MMAL + 0.15 MGLY | | | | |
| $\quad\quad$ +0.15 CO + 0.1 GLY $-$ 0.55 OH | | | | |
| NC4CHO $\xrightarrow{16\%}$ NO$_2$ + OH + CO + 0.5 HPKETAL + 0.5 HPDIAL | | | | |
| NC4CHO $\xrightarrow{48\%}$ NO$_2$ + CO + OH + 0.3 HMVK + 0.7 HMAC | | | | |
| NC4CHO $\xrightarrow{20\%}$ NO$_2$ + 1.7 CO + 0.3 MVKO2 + 0.7 HYAC | | | | |
| DHBO $\rightarrow$ CH$_3$CO$_3$ + GLYALD | 5 | 5 | 5 | 2.7(-6) |
| HOBA $\rightarrow$ MGLY + CO + 2 HO$_2$ | $5^n$ | $5^n$ | 5 | 7.9(-6) |
| HOBA $\rightarrow$ CH$_3$CO$_3$ + GLY + HO$_2$ | $6^n$ | $6^n$ | | 1.9(-6) |
| HCOC5 $\rightarrow$ CH$_3$CO$_3$ + HCHO + GCO3 | 5 | 5 | 5 | 2.3(-6) |
| ICHE $\xrightarrow{28\%}$ 2 CO + HO$_2$ + OH + HYAC | $6^d$ | $6^d$ | $o$ | 6.2(-5) |
| $\xrightarrow{72\%}$ CO + HO$_2$ + MVKO2 | | | $o$ | |
| HPCE $\rightarrow$ HO$_2$ + 1.82 CO + 0.82 OH + 0.82 HPACET + 0.18 KPO2 | $6^d$ | $6^d$ | $p$ | 6.2(-5) |
| MCO3H $\rightarrow$ OH + CO$_2$ + 0.65 (CH$_3$O$_2$ + CO + HCHO) | $1^q$ | $b$ | 5 | 7.9(-7) |
| $\quad\quad$ +0.35 (CH$_3$CO$_3$ + HCHO) | | | | |
| GCO3H $\rightarrow$ OH + HO$_2$ + HCHO + CO$_2$ | $1^q$ | $b$ | 5 | 7.9(-7) |
| HPAC $\xrightarrow{84\%}$ VA | 7 | $7^e$ | $7^r$ | 3.6(-4) |
| $\xrightarrow{16\%}$ HO$_2$ + CO + HCHO + OH | | | | |
| HPACET $\xrightarrow{84\%}$ MVA | 7 | $7^e$ | $7^r$ | 1.3(-4) |
| $\xrightarrow{16\%}$ CH$_3$CO$_3$ + HCHO + OH | | | | |

| Reaction | Cross section | Quantum yield | Products | $J$ (s$^{-1}$) |
|---|---|---|---|---|
| HPKETAL $\xrightarrow{50\%}$ HMVK | 7 | $7^e$ | $r$ | 5.4(-4) |
| $\xrightarrow{25\%}$ CH$_3$CO$_3$ + OH + GLY | | | | |
| $\xrightarrow{25\%}$ CO + HO$_2$ + OH + MGLY | | | | |
| HPDIAL $\xrightarrow{50\%}$ HMAC | 7 | $7^e$ | $r$ | 5.2(-4) |
| $\xrightarrow{50\%}$ CO + HO$_2$ + OH + MGLY | | | | |
| DIHPMEK $\rightarrow$ OH + CH$_3$CO$_3$ + HPAC | 7 | $7^e$ | $5^r$ | 1.3(-4) |
| BIACETOH $\xrightarrow{50\%}$ CH$_3$CO$_3$ + GCO3 | $6^s$ | $6^s$ | $t$ | 7.1(-5) |
| $\xrightarrow{50\%}$ CH$_3$CO$_3$ + CO + HO$_2$ + HCHO | | | | |
| HPALD1 $\xrightarrow{11\%}$ 0.45 OH + 1.15 HO$_2$ + 1.35 CO$_2$ + 0.55 HCHO | $1^u$ | $u$ | $11^u$ | 4.2(-4) |
| +0.65 CH$_3$CO$_3$ + 0.2 MMAL + 0.15 MGLY + 0.15 CO + 0.1 GLY | | | | |
| $\xrightarrow{11\%}$ 2 OH + CO + HPKETAL | | | | |
| $\xrightarrow{56\%}$ CO + 2 OH + HMVK | | | | |
| $\xrightarrow{22\%}$ CO + CH$_3$CO$_3$ + GLYALD | | | | |
| HPALD2 $\xrightarrow{18\%}$ 0.45 OH + 1.15 HO$_2$ + 1.35 CO$_2$ + 0.55 HCHO | $1^u$ | $u$ | $11^u$ | 4.2(-4) |
| +0.65 CH$_3$CO$_3$ + 0.2 MMAL + 0.15 MGLY + 0.15 CO + 0.1 GLY | | | | |
| $\xrightarrow{18\%}$ 2 OH + CO + HPKETAL | | | | |
| $\xrightarrow{46\%}$ CO + 2 OH + HMAC | | | | |
| $\xrightarrow{18\%}$ 2 CO + HO$_2$ + HYAC | | | | |
| HMAC $\rightarrow$ OH + CO + HO$_2$ + MGLY | 12 | $v$ | $w$ | 1.0(-5) |
| HMVK $\rightarrow$ OH + CH$_3$CO$_3$ + GLY | 12 | $v$ | $w$ | 1.0(-5) |
| PGA $\rightarrow$ CO + HO$_2$ + CO$_2$ + OH | $x$ | $x$ | 5 | 1.1(-4) |
| APINONO2 $\rightarrow$ NO$_2$ | $6^g$ | $b$ | | 2.9(-6) |

**Notes:**

$a$) Total quantum yield of 0.004.

$b$) Unit quantum yield.

$c$) As for CH$_3$OOH.

$d$) As for i$-$C$_3$H$_7$CHO.

$e$) Total quantum yield of 0.8.

$f$) See Sect. 2.1.4 regarding hydroperoxycarbonyl photolysis, and note N56 above.

$g$) As for CH$_3$CH(ONO$_2$)CH$_3$.

$h$) Oxy radical decomposition follows Vereecken and Peeters (2009).

$i$) Oxy decomposition as in INDO2 + NO (Table 2).

$j$) Sum of absorption cross sections of CH$_3$C(O)C$_2$H$_5$ and n$-$C$_4$H$_9$ONO$_2$.

$k$) As n$-$C$_4$H$_9$ONO$_2$.

$l$) Quantum yield of 1 below 336 nm, zero above (Xiong et al., 2016).

$m$) NC4CHO photolysis follows HPALD2 photolysis for 75% and HPALD1 for 25% (isomer distribution of Schwantes et al. (2015)).

$n$) For the aldehyde channel, use $J(C_2H_5CHO)$; for the ketone channel, use $J(HYAC)$.

$o$) C-C scission leading to HCO and the same product radicals as in the formyl-H-abstraction pathway in ICHE+OH (Note N17).

$p$) C-C scission leading to HCO and the same product radicals as in the formyl-H-abstraction pathway in HPCE+OH (Sect. 2.1.2).

$q$) As for $CH_3C(O)OOH$.

$r$) See Sect. 2.1.4 regarding hydroperoxycarbonyl photolysis.

$s$) Photorate taken as 25% of $J(CH_3C(O)C(O)CH_3)$ based on the experimental photorate determination of Praske et al. (2015).

$t$) The reaction gives dominantly $CH_3C°O + HOCH_2C°O$. The latter radical is formed with an internal energy ranging between 5 and 20 kcal mol$^{-1}$. Below $\sim$11.5 kcal mol$^{-1}$, it mostly adds $O_2$; above that threshold, it mostly dissociates to $CO + CH_2OH$ (barrier $\sim$11 kcal mol$^{-1}$) (Méreau et al., 2001)).

$u$) Absorption cross sections of MACR, quantum yield of 0.8. See Sect. 2.1.5 for the products.

$v$) Quantum yield of 0.1 below the threshold of 312 nm (see Sect. 2.1.5).

$w$) See Sect. 2.1.5.

$x$) For peroxyglyoxylic acid, use the same photolysis parameters as for glyoxylic acid (Back and Yamamoto, 1985). The quantum yield is equal to 0.71.

## 2.11   Uptake by aerosols

The heterogeneous reactions on aerosols are listed in Table 4 with their associated reactive uptake coefficients. The rate ($\lambda$) for the heterogeneous uptake of a chemical compound on aqueous aerosols is calculated using

$$\lambda = \frac{A}{r_n/D_g + 4/(\omega \cdot \gamma)}, \tag{13}$$

where $A$ is the aerosol surface density (cm$^2$ cm$^{-3}$), $r_n$ is the number mean particle radius (cm), $D_g$ is the gas-phase diffusivity parameterized
as described in Müller et al. (2008), $\omega$ is the mean molecular speed (cm s$^{-1}$), and $\gamma$ the reactive uptake coefficient (Table 4). The aerosol surface density is calculated following (Stavrakou et al., 2009b). Aqueous aerosols include inorganic (sulfate/ammonium/nitrate/water) and carbonaceous (OC and BC) calculated by the model as described in Stavrakou et al. (2013) and sea-salt aerosol from the MACC (Monitoring Atmospheric Composition and Climate) Reanalysis (apps.ecmwf.int/datasets/data/macc-reanalysis/levtype=sfc/).

The heterogeneous uptake of alkyl nitrates by aqueous aerosols followed by their hydrolysis has been suggested as a substantial organic
nitrate sink and a large source of nitric acid in forested environments (Romer et al., 2016). Since tertiary nitrates were shown in the laboratory to undergo hydrolysis much faster than primary and secondary nitrates, we neglect the hydrolysis of non-tertiary nitrates while assuming fast hydrolysis of tertiary nitrates from isoprene. The reactive uptake coefficient ($\gamma$) calculated by Marais et al. (2016) based on measured hydrolysis rates of a primary and a secondary hydroxynitrate from isoprene in neutral solution (Jacobs et al., 2014) is much too low ($1.3 \cdot 10^{-7} - 5.2 \cdot 10^{-5}$) to account for the loss observed during the Southern Oxidant and Aerosol Study (SOAS) campaign (Romer et al.,
2016), due to the relatively low estimated Henry's law constant of isoprene hydroxynitrates. A much higher $\gamma$ (0.03) is assumed here for the major (tertiary) 1,2-hydroxynitrate from isoprene (ISOPBNO3), such that heterogeneous loss is its dominant fate in the troposphere, whereas the uptake of non-tertiary isoprene hydroxynitrates is neglected. Although crude, this assumption leads to a good model agreement against aircraft observations of isoprene hydroxynitrates over the Southeastern U.S. (see Sect. 4.2). Furthermore, the calculated average $\gamma$ for the sum of isoprene hydroxynitrates weighted by their respective abundances is $\sim$0.02, consistent with the upper limit (0.02) inferred
for the isoprene hydroxynitrate family by Wolfe et al. (2015) based on SOAS measurements. An uncertain, but likely significant, fraction of

**Table 4.** Heterogeneous reactions on aqueous aerosols. $\gamma$ denotes the reactive uptake coefficient. References: 1, Liggio et al. (2005); 2, Marais et al. (2016); 3, Fisher et al. (2016); 4, Müller et al. (2016). Notes:$a$) The dependence on aerosol pH (Marais et al., 2016; Stadtler et al., 2018) is ignored.

| Reaction | $\gamma$ | Ref. |
|---|---|---|
| $GLY \rightarrow GLY(aerosol)$ | $2.9(-3)$ | 1 |
| $IEPOX \rightarrow IEPOX(aerosol)$ | $4.2(-3)$ | $2^a$ |
| $HMML \rightarrow HMML(aerosol)$ | $1.3(-4)$ | $2^a$ |
| $ISOPBNO3 \rightarrow ISOPBOH + HNO_3$ | $0.03$ | $b$ |
| $MACRNO3 \rightarrow MACROH + HNO_3$ | $0.03$ | $b$ |
| $APINONO2 \rightarrow HNO_3 + product$ | $0.005$ | 3 |
| $CH_3OOOH \rightarrow CH_3OH + O_2$ | $0.1$ | 4 |

the monoterpene nitrates (represented in the mechanism by a unique lumped compound APINONO2) is assumed to be tertiary and undergoes hydrolysis (Browne et al., 2013, 2014) with $\gamma = 0.005$ (Fisher et al., 2016). Other, minor tertiary nitrates generated in the mechanism (INB1OOH, INB2OOH, INB1NO3 in MCM) are also assumed to undergo rapid uptake followed by hydrolysis in the aerosol, generating $HNO_3$ and a usually very soluble and condensable co-product assumed to remain in the particulate phase. The saturation vapour pressures

of those hydrolysis products (hydroperoxy triols and nitroxy triol) are calculated to be in the range $(4–40) \cdot 10^{-10}$ atm using the group contribution method of Compernolle et al. (2011), i.e. three orders of magnitude below the estimated vapour pressure of isoprene dihydroxy epoxide (IEPOX). The assumed rapid aerosol sink of the dinitrate INB1NO3 ($O_2NOCH(CH_2OH)C(CH_3)(ONO_2)CH_2OH$) generated in the oxidation of isoprene hydroxynitrates by OH has a potentially significant impact on total $RONO_2$ levels, due to its long expected chemical gas-phase lifetime, with an OH-rate constant of $\sim 2 \cdot 10^{-12}$ molec.$^{-1}$ cm$^3$ s$^{-1}$ (Saunders et al., 2003). However, a global model

sensitivity simulation ignoring the aerosol sink of INB1NO3 and assuming similar gas-phase sink reactions as for the dinitrate INCNO3 ($HOCH_2CH(ONO_2)C(CH_3)(OH)CH_2ONO_2$) shows that dinitrate hydrolysis depletes total $RONO_2$ levels by only $\sim 3\%$ globally, in spite of its strong impact on total dinitrate abundances (factor of 10).

The hydrolysis of non-tertiary nitrates is slow compared to tertiary nitrates, and is therefore neglected here. Gas-aerosol partitioning might occur, leading to possible loss by aerosol dry or wet deposition; this loss could be significant if repartitioning of particulate nitrates to the gas

phase would be inhibited (Fisher et al., 2016). These effects are however very uncertain, and are not considered here for simplicity.

## 3 Box model comparison with other isoprene mechanisms

### 3.1 Description of simulations

The isoprene mechanism is evaluated against the MCMv3.3.1, obtained from http://mcm.leeds.ac.uk/MCM/ (Jenkin et al., 2015), and the Caltech reduced mechanism (version 4.3) obtained from http://dx.doi.org/10.22002/D1.247 (Wennberg et al., 2018). The Caltech mechanism

is also available in its explicit ("full") version, which however does not include the further degradation of many terminal species down to $CO_2$ and is therefore not appropriate for comparison. We perform 30-hour simulations starting at 9 AM with 2 ppbv isoprene. Temperature is set to 298 K, and the $H_2O$ mixing ratio is 1%. Two scenarios are considered: a high-$NO_x$ scenario with 1 ppbv $NO_x$ (also 40 ppbv $O_3$ and 250 ppbv CO) and a low-$NO_x$ scenario with 100 pptv $NO_x$ (with 20 ppbv $O_3$ and 150 ppbv CO). The photolysis rates are calculated for

clear-sky conditions in mid-July at 30°N, with 300 DU ozone and an albedo of 0.05 using the Tropospheric Ultraviolet and Visible (TUV) photolysis model of Madronich (1993). For computational efficiency, the photorates are parameterized as a function of solar zenith angle using MCM-type expressions (Saunders et al., 2003),

$$J = l \cdot (\cos \chi)^m \cdot \exp(-n/\cos \chi) \tag{14}$$

where the parameters $l$, $m$ and $n$ are obtained from TUV calculations at three zenith angles (0°, 30° and 60°). For convenience, the numbering of the photodissociations is the same as in the MCM, except for those (e.g. hydroperoxycarbonyls) for which the MCM falls back on simpler, monofunctional model compounds. Since Wennberg et al. (2018) does not provide specific recommendations for the calculation of photorates, we use our own expressions in their mechanism. The Caltech mechanism files do include noontime photorate estimates, but their derivation is unclear, and their use in the intercomparison would lead to large discrepancies with both MCM and MAGRITTE, obscuring the interpretation of differences. To further facilitate this interpretation, the same inorganic chemistry and the same rates of the major reactions of $CH_3O_2$ and $CH_3CO_3$ (with NO, $HO_2$ and $NO_2$) as well as of PAN-like compounds are adopted in the three mechanisms. Heterogeneous uptake on aerosols are also included, calculated assuming an aerosol surface density of $5 \cdot 10^7$ cm$^2$ cm$^{-3}$ with uptake coefficients as in Table 4. All rate coefficient expressions are available at the MAGRITTE mechanism repository (http://doi.org/10.18758/71021042).

## 3.2 Comparison results for $HO_x$

The temporal evolution of key compounds concentrations calculated with the three mechanisms using the Kinetic PreProcessor (KPP) package (Damian et al., 2002) are displayed on Fig. 4 (for high-NOx) and 5 (low-NOx). The initial isoprene is more rapidly consumed at high-NOx ($< 2$ hours) than at low-NOx ($\sim 5$ hours) due to higher OH levels ($\sim 10^7$ vs. $\sim 2 \cdot 10^6$ molec. cm$^{-3}$). There is generally a much better level of agreement between the mechanisms at high-NOx compared to low-NOx. The Caltech mechanism leads to the highest OH levels. At low-NOx, the Caltech-based average [OH] during the first 4 hours of the numerical experiment is by factors of 1.25 and 1.32 higher than with the MCM and MAGRITTE mechanisms, respectively. The Caltech-based model predicts also higher $HO_2$ (by a factor of $\sim 1.1$), $CH_3O_2$ ($\sim 1.3$) and especially $CH_3CO_3$ ($\sim 1.4$). The differences between the three mechanisms do not exceed a few percent at high-NOx. There are several causes for the large differences at low NOx.

The first reason is that the Caltech mechanism includes a higher direct OH yield (1.5) in the bulk 1,6-isomerisation of isoprene peroxy radicals. This production is the result of the high assumed yield of DIHPCARP (0.6) in this reaction and of the high direct (1) and secondary (1.5) yield of OH radical resulting from the degradation of DIHPCARPs. Furthermore, the $\beta$-HPALDs also formed in the 1,6-isomerisation of isoprene peroxys are mainly lost by photolysis, leading to additional $HO_x$ production. As a sensitivity test, the model was run with the MAGRITTE mechanism modified by replacing the bulk 1,6 H-shift reaction of isoprene peroxys by its representation in the Caltech mechanism. This change alone increases OH concentrations by about 15% compared to the standard MAGRITTE simulation, and reduces also the discrepancies for $HO_2$, $CH_3O_2$ and $CH_3CO_3$.

A second reason for lower $HO_x$ levels lies in the yield of $HO_x$ and other radicals in the photolysis of several major hydroperoxycarbonyls (e.g. HPAC, HPACET and HPKETAL). This yield is much lower in our mechanism, as it accounts for the major enol-forming channel (Liu et al., 2018), which does not produce any radical. Those reactions generate one OH and either one $HO_2$ or one $CH_3CO_3$ radical in the Caltech mechanism, which assumes either scission of the C−−C bond followed by OH expulsion, or equivalently, direct OH release followed by splitting off of either formyl or acetyl radical. A second sensitivity calculation with the MAGRITTE mechanism modified by assuming that the photolysis of those hydroperoxycarbonyls proceeds as in the Caltech mechanism further increases OH by almost 10%, in the first hours. Even larger increases are calculated ($\sim 20\%$) for $CH_3O_2$ and $CH_3CO_3$.

A lesser, but significant, factor also contributing to the differences includes the higher bulk 1,6-isomerisation yield in the reduced Caltech mechanism, in large part due to the neglect of the minor OH-addition pathways to the central carbons of isoprene, which represent 7% of the total ISOP + OH reaction flux in our mechanism.

The results of a sensitivity calculation using the MAGRITTE mechanism modified by adopting the Caltech reduced mechanism representation of 1) isoprene peroxy 1,6 H-shift yield and products, and 2) hydroperoxycarbonyl photolysis reactions are shown on Fig. 5 ("Hybrid mechanism", dashed red lines). The residual differences between Caltech and the modified MAGRITTE mechanisms are very small (a few percent) for $HO_x$, $CH_3O_2$ and $CH_3CO_3$.

## 3.3 Comparison results for isoprene products

The three mechanisms agree well for the main isoprene oxidation products (e.g. MVK, MACR, HCHO) when accounting for differences in OH levels and in the HPALD yield in the bulk 1,6-isomerisation of isoprene peroxys (0.25, 0.5 and 0.75 in the Caltech, MCM and MAGRITTE mechanisms). The lower yield of primary hydroxynitrates (ISOPN) in ISOPO2+NO reactions in the MCM (10%, vs. ~13% following Wennberg et al. (2018)) explains the lower MCM ISOPN and total organic nitrates (RONO2) concentrations during the first hours. Note that higher ISOPN and RONO2 levels (by a factor of ~1.2) are calculated when the aerosol sink of tertiary nitrates is not considered.

In spite of the similar ISOPN concentrations in the three simulations, the calculated RONO2 levels decrease more rapidly after the initial peak in the Caltech simulation than in the MAGRITTE and especially the MCM simulation (Fig. 5). This is partly explained by differences in OH, as seen from the lower discrepancy in RONO2 found between the Caltech and hybrid mechanism simulations which realize very similar OH levels. An additional cause of difference in RONO2 levels is the 1,5 H-shift in dihydroxy nitroxyperoxy radicals (INBO2 and INDO2) formed from the OH-oxidation of isoprene hydroxynitrates. This H-shift forms hydroperoxynitroxy carbonyls assumed to photolyze very rapidly, releasing $NO_2$ and therefore removing RONO2. It is the dominant sink of those peroxys in the Caltech simulation, while it is neglected in the MCM, and assumed to proceed at a slower rate (0.02 s$^{-1}$) in our mechanism, due to the influence of H-bonding (see Notes N11 and N14). This also explains the higher abundance of the carbonylhydroxynitrates (MVKNO3 and MACRNO3) in the MCM and MAGRITTE simulations (Fig. 5), as those are partly formed from the bimolecular reactions of the peroxys INBO2 and INDO2.

Dinitrates make up only a very small contribution to total RONO2 levels in the simulations (<0.5% at low-NOx, <3% at high-NOx). The dinitrates formed from ISOP+OH are indeed mostly tertiary and therefore assumed to hydrolyse rapidly to $HNO_3$ and an alcohol. When the aerosol sink of those nitrates is neglected, their contribution to total RONO2 becomes substantial (13 pptv out of 52 pptv at low-NOx) in the MCM simulation, but remains low in the Caltech simulation (<2 pptv). This large difference stems mostly from lower dinitrate yield in the reactions of dihydroxy nitroxyperoxy radicals with NO in the Caltech mechanism, due to the strong reduction of the yield due to the nitrate group. Moreover, the MCM neglects the photolysis of the dinitrates, which represents about one third of their total (non aerosol-related) sink according to our estimation. Both the aerosol reactions and the dinitrate yield are acknowledged as very uncertain, however, and the overall impact of dinitrates could be larger than assumed in our mechanism.

The total peroxynitrate (RO2NO2), methylglyoxal and glyoxal concentrations calculated in the three simulations are in reasonable agreement. The differences in RO2NO2 level are partly related to differences in yield of the $HOCH_2C(O)O_2$ radical (GCO3) in the photolysis of $CH_3C(O)C(O)CH_2OH$, equal to 1 in the MCM, 0.5 in our mechanism, and 0 in the Caltech mechanism (see Note $t$ in Sect. 2.10).

The production of methanol, however, is much larger with MAGRITTE than with the MCM (factor of 3) and with the Caltech mechanism (factor of 8). A large part of this difference is due to the $CH_3O_2$ + OH reaction (Sect. 2.7), which accounts for about half the $CH_3OH$ production at low NOx, and even more at high NOx. In addition, the rate of the $CH_3O_2$ + RO2 reactions has a unique value for all RO2 compounds (3.5·10$^{-13}$ molec.$^{-1}$ cm$^3$ s$^{-1}$ at 298 K ) in the MCM, much lower than in the Caltech and MAGRITTE mechanism for isoprene

hydroxyperoxys ($2 \cdot 10^{-12}$ molec.$^{-1}$ cm$^3$). Finally, although the full Caltech mechanism includes $CH_3OH$ formation in the reaction of e.g. ISOPDO2 (4,3-ISOPOO) with $CH_3O_2$, this production is neglected in the reduced Caltech mechanism, explaining the very low Caltech-calculated methanol levels on Fig. 4-5.

Very large differences are also found for formic acid. In the first hour of the experiment, MAGRITTE predicts lower formation rates due to lower direct HCOOH formation from the ozonolysis of isoprene: in particular, the primary HCOOH yield is only about 3% in MAGRITTE, about 6 times less than in both the MCM and Caltech mechanism (at 1% $H_2O$ mixing ratio). HMHP ($HOCH_2OOH$) being not formed in the MCM, the overall HCOOH production from alkene ozonolysis (both direct and indirect through HMHP oxidation) is slightly higher in MAGRITTE than in MCM, whereas it is about twice higher in the Caltech mechanism. At later times, the formation of formic acid due to the reactions of enols (VA, HMAC and HMVK) with OH becomes a larger source than the ozonolysis of isoprene and its degradation products according to MAGRITTE, especially at low-NOx. The Caltech mechanism includes an additional HCOOH production pathway through the oxidation of secondary isoprene nitrates (e.g. $CH_3C(O)CH(ONO_2)CH_2OH$) by OH, which becomes significant at high-NOx. This mechanism proposed by Paulot et al. (2009b) involves abstraction of an $\alpha$-hydroxy-H, followed by $O_2$-addition and by a rearrangement leading to $NO_3$ + HCOOH + MGLY, instead of the expected fast dissociation of the $\alpha$-hydroxyperoxy radical into $HO_2$ and a dicarbonyl. This mechanism is ignored in our mechanism, as it is highly complex and likely faces a much higher barrier than the fast $HO_2$ expulsion (at $\sim$1000 s$^{-1}$, Hermans et al. (2005)).

Finally, the production of acetic acid is relatively similar in the three mechanisms. The slightly lower acetic acid production in the Caltech run is primarily due to a lower $CH_3C(O)OH$ yield in the $CH_3C(O)O_3 + HO_2$ reaction (0.13 vs. 0.16 in MCM and MAGRITTE) and to the neglect of $CH_3C(O)OH$ formation through reactions of isoprene peroxys with $CH_3CO_3$. It is partly compensated by higher $CH_3CO_3$ levels in the Caltech simulation, especially at low-NOx. The MAGRITTE mechanism includes an additional acetic acid source through the OH-oxidation of $CH_2$=$C(CH_3)OH$ (MVA) generated from the photolysis of hydroperoxyacetone HPACET. This source accounts for $\sim$28% and 38% of the total $CH_3C(O)OH$ source at high- and low-NOx, respectively.

## 4 Regional and global modelling

### 4.1 Model description and simulations

The MAGRITTE v1.1 model calculates the distribution of 182 chemical compounds, among which 141 species undergo transport processes (advection, deep convection and turbulent diffusion) in the model. MAGRITTE can be run either globally at $2°$ (latitude) $\times$ $2.5°$ (longitude) resolution, or regionally at $0.5° \times 0.5°$ resolution. The lateral boundary conditions of the regional model are provided by the global model. In the vertical, the model uses a hybrid ($\sigma$-pressure) coordinate, with 40 levels between the Earth's surface and the lower stratosphere (44 hPa level). The meteorological fields are provided by ECMWF ERA-Interim analyses (Dee et al., 2011). Most model parameterizations, including the transport scheme and the chemical mechanism for anthropogenic and biomass burning VOCs, inherit from the IMAGES model (Muller and Brasseur, 1995; Stavrakou et al., 2009a, b, 2015; Bauwens et al., 2016). The deposition scheme is described in a companion paper (Müller et al., 2018).

The model uses anthropogenic emissions of CO, NOx, OC, BC, and $SO_2$ from the HTAPv2 dataset for year 2010 (Janssens-Maenhout et al., 2015). Following Travis et al. (2016), the anthropogenic NOx emissions over the U.S. are first scaled down to match the U.S. total (3.5 TgN/yr) for the year 2013 reported by the National Emission Inventory (NEI), and the U.S. NOx emissions due to industry and transport are further reduced by 60% to match observed aircraft $NO_x$ concentrations and nitric acid deposition data, consistent with the recommendation of Anderson et al. (2014). Anthropogenic NMVOC emissions are provided by the EDGARv4.3.2 inventory (Huang et al., 2017) for

the year 2012. The global annual anthropogenic NMVOC source is 154 TgNMVOC (118 TgC). Biomass burning emissions (78 TgNMVOC or 45 TgC in 2013) are obtained from the Global Fire Emission Database version 4 (GFED4s) (van der Werf et al., 2017) and are vertically distributed according to Sofiev et al. (2013).

Isoprene, monoterpene and MBO fluxes (366, 91.5 and 0.93 TgC, respectively, in 2013) are calculated by the MEGAN-MOHYCAN model (Müller et al., 2008; Guenther et al., 2012; Bauwens et al., 2018) and are available online (http://emissions.aeronomie.be). Biogenic emissions of acetaldehyde and ethanol (amounting to 92 and 88 Tg(C) $yr^{-1}$ globally) are parameterized as in Millet et al. (2010). The methanol biogenic emissions are provided by an inverse modelling study constrained by spaceborne methanol abundances and are estimated at 37.5 Tg(C) $yr^{-1}$ (Stavrakou et al., 2011). Biogenic emissions of $C_2H_4$ (scaled to a global total of 4 Tg(C) $yr^{-1}$), $CH_2O$ (1.6 Tg(C) $yr^{-1}$) and $CH_3C(O)CH_3$ (18 Tg(C) $yr^{-1}$) are also provided by MEGAN (Guenther et al., 2012) (available on http://eccad.aeris-data.fr).

The model also includes oceanic emissions of methanol (18.4 Tg(C) $yr^{-1}$), acetone (39.3 Tg(C) $yr^{-1}$) and acetaldehyde (30.4 Tg(C) $yr^{-1}$) (Müller et al., 2018), similar to previous model estimations (Stavrakou et al., 2011; Fischer et al., 2012; Millet et al., 2010). Finally, oceanic emissions of alkyl nitrates are also included, based on comparisons with aircraft campaign measurements as originally proposed by Neu et al. (2008), but taking into account the updated alkylnitrate calibration of the campaign data (Simpson et al., 2011). The adopted rates over Tropical oceans ($10°S - 10°N$) are $6 \cdot 10^8$, $2.5 \cdot 10^8$, $10^8$ and $10^8$ molec. $cm^{-2}$ $s^{-1}$ for $C_1$, $C_2$, $C_3$ and $C_{>3}$ alkyl nitrates, respectively; $3 \cdot 10^7$, $3 \cdot 10^7$, $1.5 \cdot 10^7$ and $10^7$ molec. $cm^{-2}$ $s^{-1}$ over the Southern Ocean ($>10°S$); a uniform rate of $10^7$ molec. $cm^{-2}$ $s^{-1}$ is adopted elsewhere over ice-free oceans. The calculated global emissions are respectively 0.35, 0.3, 0.2 and 0.25 Tg(C) (or 0.4, 0.18, 0.08, 0.07 Tg(N)) for $C_1$, $C_2$, $C_3$ and higher alkylnitrates.

MAGRITTE is run for a period of 18 months starting on July 1, 2012, both at the global scale ($2° \times 2.5°$ resolution) and regional scale for the U.S. ($0.5° \times 0.5°$, 10-54°N, 65-130°W). Only the results for the year 2013 are discussed hereafter.

## 4.2   Model general results

Oxidation of isoprene by OH radicals is by far the largest sink of isoprene, representing ∼85% of the global sink according to the model calculations, in agreement with previous model studies (Paulot et al., 2012), whereas ozonolysis and the $NO_3$-reaction contribute for ∼9% and 5%, respectively. The isomerisation reactions control the fate of about one fifth of the total flux of hydroperoxy radicals formed from the reaction of isoprene with OH (16.5% and 3% for the 1,6 and 1,5 H-shifts, respectively). However, the contribution of 1,6 H-shift is much higher, by about one order of magnitude, for the peroxys resulting from OH-addition to carbon C4 than for those resulting from addition at C1 (Peeters et al., 2014; Wennberg et al., 2018). Furthermore, this contribution is dependent on temperature and on the concentrations of NO and $HO_2$ radicals, as illustrated on Fig. 6: of the order of 50% over remote forests such as Amazonia, it drops to ∼35% over the Southeastern U.S. and below 20% over cooler, more NOx-polluted areas (for C4-addition).

The isomerisation reactions of isoprene peroxys regenerate $HO_x$ ($HO_2 + OH$) radicals, in part directly (see Sect. 2.1.2) and in part from subsequent reactions of the isomerisation products, HPALDs in particular. However, as discussed in Sect. 3.1, the revised isomerisation product distribution of the MAGRITTEv1.1 mechanism, consistent with recent experimental findings (Berndt et al., 2019), lowers the regeneration of OH compared with distributions assuming a large yield of OH radicals and dihydroperoxycarbonyls (Peeters et al., 2014; Wennberg et al., 2018) assumed to release additional $HO_x$ through fast photolysis. Furthermore, our recently proposed enol-forming pathway in the fast photolysis of several key hydroperoxycarbonyls (e.g. HPACET and HPAC) also decreases the recycling of OH compared with the previous assumption of O−OH bond scission. The overall impact of isoprene peroxy radical isomerisation reactions on boundary-layer averaged OH concentrations reaches up to about 40% over Western Amazonia and 10-15% over Southeastern U.S. and Siberia in July (Fig. 7), whereas their impact on $HO_2$ is comparatively lower, as it does not exceed 20% over Amazonia. The isomerisation reactions lead also to

reduced isoprene nitrate formation, by up to ~40% over Amazonia, as the $RO_2 + NO$ reactions compete with unimolecular reactions. The decreased NOx loss through organic nitrate formation and partial removal implies longer NOx effective lifetime and higher concentrations (by a few % over Amazonia), in spite of the higher OH levels and increased NOx loss through $NO_2 + OH$. These changes lead to slightly enhanced $O_3$ concentrations over Amazonia (a few percent). The impact on HCHO concentrations and vertically-integrated columns is very small, also of the order of a few percent at most.

The dry or wet deposition of organic (peroxy-)nitrates and the irreversible sink of organic nitrates through hydrolysis or other processes on aerosols are significant net sinks of NOx over vegetated areas (Browne et al., 2014; Romer et al., 2016; Fisher et al., 2016). As shown on Fig. 8, the combined deposition and aerosol sink of organic (peroxy-)nitrates is found to be the dominant sink of NOx over rainforests in South America and Africa, as well as over boreal forests in Siberia and Canada during the summer. This fraction even exceeds 70% over the most remote areas (e.g. Western Amazonia) where high isoprene and low NOx levels both contribute to low OH concentrations (of the order of $10^6$ molec. cm$^{-3}$ during daytime in the boundary layer). These estimates should be considered with caution given the large uncertainties in the assumed aerosol uptake coefficient and poor understanding of aerosol chemical processes. Over the Southeastern U.S. (80-94.5°W, 29.5-40°N) during August-September 2013, the MAGRITTE model calculations (regional version over the U.S., 0.5° resolution) suggest that the NOx sink through aerosol hydrolysis amounts to 14% of NOx emissions in the region, whereas the deposition of organic nitrates and peroxynitrates account for additional 7 and 5% of NOx emissions. The estimated total net loss of NOx through $RONO_2$ formation amounts therefore to 21% of NOx emissions, in good agreement with previous calculations using the GEOS-Chem model (Fisher et al., 2016) (21%). This agreement might be partly fortuitous, given the important differences between the two studies regarding the nitrate yield in the ISOPO2 + NO reactions (9% in Fisher et al. and 13% in our study) and regarding the treatment of $RONO_2$ aerosol sink: a unique uptake coefficient (0.005) was used by Fisher et al. for all isoprene nitrates except nitroxyacetone and ethanal nitrate, whereas only tertiary nitrates are assumed to undergo aerosol hydrolysis in our study (with γ=0.03). Non-tertiary nitrates might partition to the aerosol phase and possibly undergo processes preventing their eventual release to the gas-phase, in which case the overall NOx sink calculated here is underestimated.

Although SOA is not a focus of this study, SOA formation processes are included in the model. The largest source of SOA is the uptake of IEPOX, with a global flux (49 Tg or 25 TgC yr$^{-1}$) of magnitude similar to previous model estimates, of the order of 40 Tg yr$^{-1}$ (Lin et al., 2012; Stadtler et al., 2018). These estimates are very uncertain, since the reactive uptake parameterization used in models ignores the complexity of SOA formation which involves the partitioning of semi-volatile compounds and chemical transformations in the gaseous and particulate phases (D'Ambro et al., 2018). Glyoxal is another well-identified source of SOA, amounting to 10 Tg yr$^{-1}$ globally (4.3 TgC yr$^{-1}$), also well in the range of previous estimations (6-14 Tg yr$^{-1}$) (Fu et al., 2008; Stavrakou et al., 2009b; Lin et al., 2012). The dihydroxy dihydroperoxides (ISOP(OOH)$_2$) formed from the oxidation of ISOPOOH by OH were recently estimated to be a dominant source of SOA (Stadtler et al., 2018); in our mechanism, these compounds are ignored since their yields are believed to be negligible in atmospheric conditions (D'Ambro et al., 2017). The major non-IEPOX products of OH-addition to ISOPOOH are dihydroxy hydroperoxy epoxides (DHHEPOX), also believed to form SOA as discussed above (Note N6). Their global production in the model amounts to 30 Tg yr$^{-1}$ (12 TgC yr$^{-1}$). Assuming that their reactive uptake is as effective as for IEPOX, and neglecting gas-phase oxidation by OH (which generates other low-volatility compounds also expected to form SOA), we estimate with the model that SOA formation accounts for two-thirds of the sink of DHHEPOX (i.e. 20 Tg yr$^{-1}$), whereas dry/wet deposition makes up the rest. If confirmed, this would make DHHEPOX the second-largest contribution to isoprene SOA.

Other SOA formation pathways are implied, but not explicitly represented by the MAGRITTE mechanism, such as the hydrolysis of dihydroxy dinitrates (Note N12) and dihydroxy hydroperoxy nitrates (Note N13). The hydrolysis products, nitroxy- and hydroperoxy-triols

are expected to be of very low volatility and remain mostly in the aerosol phase, as their vapour pressures (Compernolle et al., 2011) are estimated to be very low. Those triols represent only a minor contribution to the global SOA budget, however, as their estimated global production is $\sim$3 Tg yr$^{-1}$ (1.2 TgC yr$^{-1}$).

## 4.3 Model evaluation against SEAC[4]RS campaign measurements

5 The regional model simulation over the U.S. is evaluated against aircraft measurements of the NASA SEAC[4]RS (Studies of Emissions and Atmospheric Composition, Clouds and Climate Coupling by Regional Surveys) campaign in August-September 2013 (Toon et al., 2016). For the most part, the SEAC[4]RS took place over the Southeastern U.S. in areas characterized by high emissions of isoprene and other BVOCs. The observations discussed below are those obtained on the NASA DC-8 (www-air.larc.nasa.gov/missions/merges/) between 9h and 17h local time. Biomass burning plumes, urban plumes and stratospheric air are excluded from the analysis (diagnosed with [CH$_3$CN] > 225

10 ppt, [NO$_2$] > 4 ppbv, and [O$_3$]/[CO] > 1.25, respectively) (Travis et al., 2016).

Figure 9 presents the observed and calculated average profiles of ozone, NO$_x$ and VOC oxidation products. The model profiles are averages based on values interpolated at each measurement location and time. As noted above, the NOx anthropogenic emissions used in the model were strongly reduced, relative to NEI official estimations, in order to match the SEAC[4]RS observations for NO$_2$ (also NO) and improve the agreement for ozone, consistent with the results of Travis et al. (2016). The model is in excellent agreement with the HCHO profile measured

by the Compact Atmospheric Multispecies Spectrometer (CAMS) (Richter et al., 2015), with only about 3% average overestimation below 4 km altitude, whereas a model underestimation of 8% is found relative to HCHO measurements by laser-induced-fluorescence (NASA GSFC ISAF instrument, Cazorla et al. (2015), not shown on Fig. 9). The model performance is also fairly good for the major products of isoprene + OH, with moderate overestimations of 14%, 1% and 24% for MVK+MACR, ISOPN (the family of primary hydroxynitrates from isoprene) and ISOPOOH, respectively. Even for ISOPOOH, the model falls well within the measurement uncertainty range (40%)

(Nguyen et al., 2015b). Note that the modelled MVKMAC accounts for the presumed interference of ISOPOOH in the measurement, as described in Müller et al. (2018). This correction increases MVKMAC by $\sim$10% on average for this campaign.

The model-calculated HPALD concentrations (dotted line on the C$_5$H$_8$O$_3$ panel of Fig. 9) are on average about a factor of two lower than the observed Caltech CIMS (Chemical Ionisation Mass Spectrometry) signal at the corresponding mass; when adding the contribution of the carbonyl hydroxyepoxides (ICHE), which have the same formula (C$_5$H$_8$O$_3$) as HPALD and can be expected to interfere with HPALD

measurements, the model falls within the measurement uncertainty range (50%) with an underestimation decreased to -34% (solid line on Fig. 9). The ICHE compounds are formed from the oxidation of IEPOX (as well as HPALDs) by OH. It is likely than other, unknown compounds contribute to the CIMS signal at the same mass, as also observed in the PROPHET campaign in Michigan, where the HPALD contribution to the CIMS measurement at the given mass was estimated at 38% based on the relative contribution of the HPALD peaks to the total GC area (Vasquez et al., 2018). This is consistent with our modelled HPALD accounting for 50% of the CIMS measurement, when

considering also that all isoprene oxidation products appear slightly overestimated by the model as suggested by the $\sim$20% overprediction of modelled ISOPOOH and MVK+MACR relative to the measurements. In spite of the important uncertainties and remaining unknowns (e.g. the identity of additional compounds contributing to the CIMS signal), this good consistency provides strong support to the high HPALD yield (75%) adopted in this work in the isomerisation of $Z$-$\delta$-OH-peroxys from isoprene (Sect. 2.1.2). Lower yield values as proposed in recent previous work, i.e. 50% (Peeters et al., 2014; Jenkin et al., 2015) or 25% (Teng et al., 2017; Wennberg et al., 2018) would lead to

much stronger HPALD underestimations against SEAC[4]RS data.

The good consistency between the model results for the major high-NOx and low-NOx isoprene oxidation products lends confidence in the major steps of the mechanism. The excellent agreement for IEPOX (+2% bias below 4 km) might be partly fortuitous given the

highly uncertain aerosol sink ($\sim$35% of the total IEPOX sink in the model simulation), without which the model would largely overestimate IEPOX observations. The slightly too low ISOPN/MVKMAC ratio in the model (0.036 vs. 0.041) could indicate an overestimation of ISOPN aerosol sink, although the measurement uncertainties ($\sim$30% for ISOPN, Fisher et al. (2016)) preclude a firm assessment. Aerosol hydrolysis represents $\sim$50% of the total sink of the tertiary hydroxynitrate ISOPBNO3 in the model (average over the model domain) or about 31% of the total ISOPN sink. The model overestimation of the secondary isoprene nitrates (MVKNO3+MACRNO3) (Fig. 9) is small (14%) and suggests an essentially correct representation of their sources and sinks, although error compensations remain a possibility. The model overestimates nitroxyacetone (NOA) by $\sim$170%, in contrast with the GEOS-Chem underestimation found by Fisher et al.. This compound is mainly produced from multiple reaction sequences in the $NO_3$-initiated oxidation mechanism of isoprene and in the OH-oxidation mechanism of the $\delta$-hydroxynitrate $HOCH_2CH{=}C(CH_3)CH_2ONO_2$ (ISOPCNO3). Although isoprene oxidation by $NO_3$ is primarily a nighttime process, NOA is formed after several oxidation steps favored by daylight. Our mechanism is more detailed and in line with the recent mechanistic conclusions from laboratory studies, but it still bears large uncertainties due to the high complexity of the mechanism. For example, the H-shift in the nitroxyperoxy radical INCO2 ($HOCH_2CH(OH)C(O_2)(CH_3)CH_2ONO_2$ and isomer) leads to NOA formation according to our mechanism; although this process is written as one reaction in the mechanism, it actually involves several steps, each of which is uncertain. The model might also overestimate nitrate radical concentrations and therefore also the importance of $NO_3$ as oxidant of isoprene. Although the reactions of $NO_3$ with major peroxy radicals and carbonyls are taken into account in the model, many reactions with unsaturated oxidation products (e.g. ISOPOOH) are neglected in current mechanisms. A careful assessment of the role of these reactions might be in order.

Despite the model overestimation for NOA, the model underestimates the SEAC$^4$RS measurement for $RONO_2$ (the sum of all organic nitrates) by $\sim$40%. A slightly larger model underestimation (factor of 2) was found by Fisher et al. (2016), in line with their lower $RONO_2$ yield in the ISOPO2 + NO reactions (see above). There are several possible explanations for the discrepancy, including the neglected reactions of $NO_3$ with unsaturated oxidation products from isoprene and other BVOCs, the neglected formation of unsaturated dinitrates from the reaction of dinitroxyperoxy radicals (NISOPO2) with NO (Li et al., 2018), a possible overestimate of the tertiary nitrate hydrolysis sink, in particular for dinitrates, and a misrepresentation of alkyl and hydroxyalkyl nitrates from other precursors than isoprene. The monoterpene nitrates are very crudely represented in the model. In particular, the assumption of 100% NOx recycling in their reaction with OH could lead to a significant overestimation of $RONO_2$ loss. Nitrates from ethane, propane, ethene and propene oxidation are included in MAGRITTE, but their concentrations are largely underestimated with respect to SEAC$^4$RS observations (not shown on Fig. 9), in part due to underestimations of precursors emissions, in particular for ethane, propane and propene. However, these nitrates account for only a small part of the $RONO_2$ bias ($\sim$16 pptv altogether out of 120 pptv below 4 km) based on SEAC$^4$RS observations and model results. Nitrates from higher alkanes are crudely included in the model, and their contribution could be underestimated. Methylnitrate ($CH_3ONO_2$) is well reproduced by the model (Fig. 9), but it makes only a very small contribution ($\sim$5 ppt). The good agreement validates the low nitrate yield used in the mechanism ($2 \cdot 10^{-4}$ at room conditions, see Note N71) for the $CH_3O_2$ + NO reaction, well below the experimental determination (1% $\pm$ 0.7% in tropospheric conditions) of Butkovskaya et al. (2012). Although a higher yield ($\sim 3 \cdot 10^{-4}$) would still remain compatible with the SEAC$^4$RS measurement (by assuming lower oceanic emissions), much higher values as reported by Butkovskaya et al. would lead to huge overestimations of $CH_3ONO_2$ mixing ratios in the troposphere.

## 4.4 Global budget of formic and acetic acid

The calculated global photochemical source of formic acid amounts to 5.6 TgC or 21 Tg(HC(O)OH) per year (Table 5). Although the model simulation incorporates newly proposed formation mechanisms, as detailed below, this total is lower than several previous model

**Table 5.** Global sources of HC(O)OH in the model simulation.

| | Tg(C)/yr | Tg(HC(O)OH)/yr |
|---|---|---|
| *Direct emissions* | | |
| Biomass burning | 0.78 | 3.0 |
| Biogenic | 1.46 | 5.6 |
| Anthropogenic | 0.58 | 2.2 |
| *Photochemical production* | | |
| $ISOP + O_3$ | 0.95 | 3.6 |
| Other Alkenes ozonolysis | 0.52 | 2.0 |
| $C_2H_2 + OH$ | 0.69 | 2.6 |
| $APIN + OH$ | 0.41 | 1.6 |
| $VA + OH$ | 1.66 | 6.4 |
| from $CH_3CHO + h\nu$ | *0.76* | *2.9* |
| from $OCHCH_2OOH + h\nu$ | *0.90* | *3.4* |
| $ISOP + OH$ (various pathways) | 1.36 | 5.2 |
| $HMAC/HMVK + OH$ | *0.91* | *3.5* |
| $ISOPOOH + OH$ | *0.44* | *1.7* |
| *Total source* | | |
| Global | 8.4 | 32 |

estimations (Paulot et al., 2011; Stavrakou et al., 2012; Millet et al., 2015), for several reasons. Firstly, the global isoprene source in our simulation (366 TgC/yr) is near the low end of the range of previous estimates (Arneth et al., 2011; Sindelarova et al., 2014). Furthermore, the formation of HC(O)OH in the oxidation of glycolaldehyde and hydroxyacetone implemented in several studies is omitted here, since the original experimental findings by Butkovskaya et al. (2006a, b) could not be confirmed (Orlando et al., 2012) and might not be effective in atmospheric conditions. HC(O)OH production from isoprene ozonolysis (1 TgC/yr) is lower than previous estimates (e.g. 1.8 and 2.3 TgC/yr in Paulot et al. (2011) and Stavrakou et al. (2012), respectively) despite our high assumed yield (0.58) of stabilized Criegee ($CH_2OO$). This is due to the combination of (1) low direct formation yield of HC(O)OH in the $CH_2OO$ reaction with the water dimer (Sheps et al., 2017), (2) high deposition sink of HMHP (over $\sim$50% of its global production) resulting from its high solubility and high deposition velocities over forests (Nguyen et al., 2015b; Müller et al., 2018), and (3) the HC(O)OH yield of only 0.45 in the reaction of HMHP with OH recently estimated from experiment (Allen et al., 2018). The very good model agreement against the SEAC[4]RS measurements of HMHP over the Southeastern U.S. suggests an essentially correct model representation of its production and sink rate, and therefore of the contribution of alkene ozonolysis to the budget of formic acid.

Vinyl alcohol (VA), originally proposed as possible source of formic acid by Archibald et al. (2007), received full attention when acetaldehyde phototautomerization to VA was shown in the laboratory to be efficient (Andrews et al., 2012) and represent a sizable source of formic acid of the order of 3 TgC/yr (Cady-Perreira et al., 2014; Millet et al., 2015). However, a recent, more detailed experimental evaluation of the phototautomerization yield led to a downward revision of the global source to about 0.8 TgC/yr (Shaw et al., 2018), in good agreement with our model calculations (Table 5). This source could be even lower if VA tautomerizes back to acetaldehyde (da Silva et al., 2010),

but acid-catalysed VA tautomerization was shown to be negligible, and aerosol-mediated tautomerization remains speculative (Peeters et al., 2015).

Another source of VA and of other enols has been identified: the photolysis of hydroperoxycarbonyls (Liu et al., 2018). Our results (Table 5) indicate that the photolysis of hydroperoxyacetaldehyde (HPAC) is a larger source of VA (and therefore of $HC(O)OH$) than $CH_3CHO$ tautomerization. The sources of HPAC (4.7 Tg/yr globally) include the oxidation of acetaldehyde by OH (35% of total), the photolysis of MVKOOH (35%) and several other pathways in isoprene oxidation, in particular through the isoprene hydroxyperoxy radical 1,6 H-shift pathway. In addition, the photolysis of the HPALDs, of $C_4$ hydroperoxydicarbonyls (HPDIAL and HPKETAL) also generated from the isomerisation pathway, and of nitroxyenals (NC4CHO) formed from isoprene + $NO_3$ all lead partly to keto-enols (HMAC and HMVK) which are oxidized for a large part into $HC(O)OH$ following their reaction with OH, adopting a similar mechanism as for VA (So et al., 2014). The photolysis and deposition of HMVK and HMAC are found to be minor sinks ($\sim$5% and 10% of their global sink, respectively). Finally, hydroperoxycarbonyls formed from minor pathways in the ISOPOOH degradation mechanism are photolyzed in part into other enol compounds, which are partly oxidized to $HC(O)OH$ (along with MVK or MACR). The estimated combined $HC(O)OH$ source due to hydroperoxycarbonyl photolysis amounts to 2.25 TgC/yr, exceeding in magnitude the source due to alkene ozonolysis (1.5 TgC/yr). As seen on Fig. 10(a), the contribution of this source to near-surface $HC(O)OH$ concentrations is highest over remote oceanic areas (up to 50%) and is comparatively much lower over biomass burning and biogenic emission areas. This is partly due to HPAC formation due to oceanic acetaldehyde emissions, and to the significant share of direct biogenic and pyrogenic emissions to the global $HC(O)OH$ budget (Table 5). Nevertheless, hydroperoxycarbonyl photolysis enhances $HC(O)OH$ levels by $\sim$20% (up to 150 pptv) near the surface over vegetated areas such as Amazonia (Fig. 10(a)), and by > 30% at higher tropospheric levels (not shown).

The largest known photochemical source of $CH_3C(O)OH$ is the reaction of acetylperoxy radical $CH_3C(O)O_2$ with peroxy radicals ($HO_2$ and $RO_2$), amounting to $\sim$16 TgC/yr globally (Table 6). This is very consistent with a previous model estimate (18 TgC/yr) by Paulot et al. (2011) but significantly lower than the estimate of Khan et al. (2018) (close to 30 TgC/yr). Our calculated contribution of $CH_3C(O)O_2 + RO_2$ reactions ($\sim$2.3 TgC/yr) is smaller than in Paulot et al. (2011) ($\sim$5.6 TgC/yr). It could be underestimated if the $CH_3C(O)OH$-forming channel ratio for the reactions of $CH_3C(O)O_2$ with major non-tertiary peroxy radicals would be significantly higher than the value assumed here for most reactions (0.1), which is based on the case of $CH_3C(O)O_2 + CH_3O_2$ (Atkinson et al., 2006). The high reported $CH_3C(O)OH$ yield (0.5) (Atkinson et al., 2006) in the case of $CH_3C(O)O_2 + CH_3C(O)CH_2O_2$ is implemented in our mechanism but assumed to be atypical.

The additional source of acetic acid due to the photolysis of hydroperoxyacetone (HPACET) and involving the oxidation of methylvinyl alcohol (MVA) by OH enhances the estimated global photochemical production of $CH_3C(O)OH$ by 4.3 TgC/yr or 26% (Table 6). The global source of HPACET (23 TgC/yr) is dominated by the acetonyl peroxy radical reaction with $HO_2$ (15 TgC/yr) and by the isoprene peroxy isomerisation pathway (2.4 TgC/yr through the 1,4 H-shift of DIHPCARP2 and 2.7 TgC/yr from the photooxidation of carbonyl hydroperoxyepoxides ICPE). The precise mechanisms for the formation of HPACET (also HPAC) in the isomerisation pathway remain uncertain. Photolysis accounts for 69% of the global HPACET sink, whereas reaction with OH and deposition account for 26 and 5%, respectively. The only significant sink of MVA, the main product of HPACET photolysis, is reaction with OH, assumed to form $CH_3C(O)OH$ (along with OH and HCHO) with a 50% yield, following a mechanism similar as for VA+OH (So et al., 2014). The calculated contribution of HPACET photolysis to the $CH_3C(O)OH$ concentration (Fig. 10(b)) is highest over forests (except in areas impacted by biomass burning), up to 23% (120 pptv) over Southeastern U.S., and 30% (120 pptv) over Amazonia.

Despite the newly-proposed large production of formic and acetic through hydroperoxycarbonyl photolysis, our derived total sources of those acids remains similar as (or even lower than) in previous modelling studies (Paulot et al., 2011; Stavrakou et al., 2012; Millet et al.,

**Table 6.** Global sources of $CH_3C(O)OH$ in the model simulation.

| | Tg(C)/yr | Tg($CH_3C(O)OH$)/yr |
|---|---|---|
| *Direct emissions* | | |
| Biomass burning | 5.7 | 14.3 |
| Anthropogenic | 2.6 | 6.6 |
| *Photochemical production* | | |
| $CH_3C(O)O_2 + HO_2$ | 14.0 | 35.0 |
| $CH_3C(O)O_2 + RO_2$ | 2.3 | 5.7 |
| HPACET + h$\nu$ (+OH) | 4.3 | 10.9 |
| *from isoprene oxidation* | *2.1* | *5.2* |
| *from acetone oxidation* | *1.5* | *3.8* |
| *other* | *0.7* | *1.8* |
| Other | 0.2 | 0.5 |
| *Total source* | | |
| Global | 29.1 | 73 |

2015; Khan et al., 2018), and is therefore insufficient to explain their high observed concentrations. Additional sources are likely at play, such as enol formation through other pathways than those considered here (e.g. in monoterpene and anthropogenic VOC oxidation, e.g. through the photolysis of aldehydes (Tadic et al., 2001a, b)) and the photodegradation of organic aerosols (Paulot et al., 2011; Malecha and Nizkodorov, 2016).

## 5  4.5  Global budget of glyoxal

The global sources of glyoxal as calculated by the model are summarized in Table 7. The model includes an important contribution from (mostly anthropogenic) acetylene and aromatic compounds to the glyoxal budget. The glyoxal yields in their reactions with OH (0.74, 0.7, 0.36 and 0.636 for benzene, toluene, xylenes and acetylene, respectively) are obtained from the MCM (Saunders et al., 2003; Bloss et al., 2005). Regarding aromatics, this yield includes not only primary formation but also later-generation production (Chan Miller et al., 2016).
Contrary to previous model evaluations (Fu et al., 2008; Stavrakou et al., 2009b; Li et al., 2016; Chan Miller et al., 2017; Silva et al., 2018), isoprene oxidation is not found to be a very large source of glyoxal, except for the significant contribution of glycolaldehyde oxidation by OH which amounts to ~4.7 TgC/yr of glyoxal. This has several causes. The oxidation of isoprene by $NO_3$ is now an almost negligible glyoxal source in our mechanism (as in the Caltech mechanism), whereas an overall yield of 35% glyoxal was inferred from the MCMv3.2 mechanism (Stavrakou et al., 2009b). First-generation glyoxal formation from ISOP + OH with a yield of ~2% at high-NOx through the
$\delta$-ISOPO2 + NO $\rightarrow$ $\delta$-ISOPO + $NO_2$ pathway (Galloway et al., 2011; Peeters and Nguyen, 2012; Nguyen and Peeters, 2015) becomes negligible under ambient atmospheric conditions due to the unimolecular reactions of the $\delta$-ISOPO2 reactions ($O_2$-elimination leading to $\beta$-ISOPO2 radicals, and 1,6 H-shift isomerisation) resulting in very small $\delta$-ISOPO2 fractions and vanishing $\delta$-ISOPO formation in the atmosphere (Peeters et al., 2014; Teng et al., 2017).

Furthermore, the oxidation of isoprene hydroxyepoxides (IEPOX), which was believed to be a potentially significant glyoxal source
(Bates et al., 2014; Li et al., 2016), is found to produce very little glyoxal in atmospheric conditions due to the proposed fast 1,4 H-shift in

**Table 7.** Global sources of glyoxal in the model simulation.

| | Tg(C)/yr | Tg(GLY)/yr |
|---|---|---|
| *Direct emissions* | | |
| Biomass burning | 1.58 | 3.8 |
| *Photochemical production* | | |
| $C_2H_2 + OH$ | 2.39 | 5.8 |
| Aromatics + OH | 3.78 | 9.1 |
| Monoterpenes oxidation | 3.67 | 8.9 |
| GLYALD + OH | 4.69 | 11.3 |
| IEPOX + OH | 0.08 | 0.2 |
| $OCHCH_2OOH + OH$ | 0.38 | 0.9 |
| HPALDs | 0.92 | 0.6 |
| ISOPOOH + OH | 0.89 | 2.2 |
| ISOP + $NO_3$ | 0.09 | 0.2 |
| Other pathways in isoprene oxidation | 1.13 | 2.7 |
| *Total source* | | |
| Global | 19.6 | 47 |

the peroxy radicals IEPOXBO2 ($HOCH_2CH(OH)C(CH_3)(O2)CHO$) formed from IEPOX + OH (Wennberg et al., 2018), outcompeting its reactions with NO and $HO_2$ (see Note N19). The 1,4 H-shift rate is very uncertain and could be overestimated, but even a factor of 10 reduction of the rate would imply a fairly small glyoxal production due to IEPOX + OH (0.6 TgC/year).

Chan Miller et al. (2017) suggested that the DIHPCARPs from the 1,6 H-shift of $\delta$-ISOPO2 partly undergoes a 1,5 H-shift to a dihydroper-
5 oxy dicarbonyl (DHDC, e.g. $OCHCH(OOH)C(CH_3)(OOH)CHO$) which would quickly photolyze to OH + an oxy radical decomposing to glyoxal and other products. However, the yield of DIHPCARPs from $\delta$-ISOPO2 isomerisation is now estimated to be much lower than previously assumed; furthermore, even under the assumption that the 1,5 H-shift would be competitive, and although DHDC photolysis should indeed be very rapid, direct OH release (followed by decomposition of the resulting oxy radical) should be negligible (Liu et al., 2018), whereas the expected preferred dissociation pathway involves formyl radical release and subsequent formation of OH and a hydroperoxy
dicarbonyl. The latter might form glyoxal upon further photolysis, but at much lower yields than in the mechanism of Chan Miller et al..

Finally, due to the fast photolysis of hydroperoxyacetaldehyde (HPAC), the fraction of the formed HPAC reacting with OH is small (23%), and only a fraction of it gives glyoxal (along with OH).

There are still large uncertainties in the mechanism, however, and direct experimental constraints on the glyoxal yields in real atmospheric conditions are lacking. Further work is needed to refine the above estimates and identify additional sources, since model evaluations against
spaceborne and in situ glyoxal measurements suggest a large photochemical source (Stavrakou et al., 2009b; Li et al., 2016; Silva et al., 2018).

# 5   Conclusions

We have presented a new BVOC oxidation mechanism for use in large-scale tropospheric chemistry-transport models. Its main focus is on isoprene, owing to its high chemical complexity and very large share of global BVOC emissions: of the 105 organic chemical species included in the mechanism, 97 compounds (74 stable compounds and 23 radicals) are involved in the chemical degradation of isoprene alone. This mechanism incorporates all major mechanistic advances from recent studies, in particular those affecting the budget of $HO_x$ and $NO_x$ radicals. Mainly thanks to $HO_x$ formation in isomerisation reactions of isoprene-derived peroxy radicals, and further OH recycling through secondary reactions, the mechanism goes a long way in explaining the large underestimations of modelled OH concentrations in isoprene-rich, $NO_x$-poor areas which prompted the community to search for OH-recycling mechanisms about a decade ago (Lelieveld et al., 2008; Hofzumahaus et al., 2009). The representation of monoterpene chemistry is much cruder, due to the still very poor understanding of its formidably complex mechanism. The simple monoterpene mechanism included here is only meant to provide an approximate reproduction of the yield of key OVOCs produced in their oxidation, based on box model simulations with the Master Chemical Mechanism (MCM).

Although smaller than e.g. the Caltech mechanism or the MCMv3.3.1, this isoprene mechanism is larger than most mechanisms implemented in large-scale models, and probably more detailed than strictly needed for many modelling purposes, such as the prediction of isoprene impacts on HOx, NOx, and ozone. Reduction techniques could be implemented to lighten the mechanism while retaining its most essential predictions, but since its current size and degree of detail can be handled by MAGRITTE, we find it useful to keep it as is in order to facilitate further analysis of model results and future mechanism updates. As pointed out by Wennberg et al. (2018), the distinction between isoprene peroxys resulting from OH addition to C1 and C4 is essential in view of the order-of-magnitude difference in bulk isomerisation rates (Fig. 6) and in the difference in the nature of the resulting products. For example, the distinction impacts also the fate of the first-generation hydroxynitrates, given the efficient hydrolysis of the tertiary 1,2-isoprene hydroxynitrate. Note that the hydrolysis rates remain very uncertain. Due to our assumption of very fast tertiary nitrate hydrolysis ($\gamma = 0.03$), about 50% of the global sink of the 1,2-isoprene hydroxynitrate is due to this process. The rate might be possibly too high, but it accounts for the fast overall hydroxynitrate loss observed in campaign measurements. This aspect of the mechanism will be revised when quantitative experimental determinations of heterogeneous processes and rates will become available.

Although many parts of our isoprene mechanism rely on the Caltech mechanism, there are notable differences. Most importantly, the 1,6 H-shift of the Z-$\delta$-hydroxyperoxy radicals generate HPALD at high yield (75% vs. 25% in the Caltech mechanism), whereas the DIHPCARPs turn out to be minor compounds, undergoing H-shift reactions along lines differing from previous work. This product distribution is fully consistent the recent experimental results of Berndt et al. (2019), supported and complemented by earlier theoretical results (Peeters and Nguyen, 2012; Peeters et al., 2014).

Another major difference between the present and previous isoprene mechanisms lies in the very fast photolysis of $\alpha$-hydroperoxycarbonyls (Liu et al., 2018), leading in several important cases to the formation of an enol which is for a large part oxidized by OH into formic or acetic acid. Also new to this mechanism, $HC(O)OH$ is formed from the OH-oxidation of keto-enols (HMVK and HMAC) produced from the photolysis of several multifunctional carbonyls. This pathway of HMVK/HMAC is all the more relevant as their photolysis is likely much slower than previously thought. More generally, the oxidation of enols formed from the oxidation of isoprene, acetaldehyde and acetone by OH is a potentially large, previously unsuspected source of carboxylic acids here estimated at 9 $Tg(HC(O)OH)$ $yr^{-1}$ (slightly larger than the contribution of alkene ozonolysis) and 11 $Tg(CH_3C(O)OH)$ $yr^{-1}$. This source amounts to a significant share (~28% for $HC(O)OH$ and 15% for $CH_3C(O)OH$) of the total identified global source, which remains however largely insufficient to account for the atmospheric observations for both compounds (e.g., Paulot et al. (2011)). Further experimental and theoretical studies of multifunctional carbonyl pho-

tolysis and enol oxidation are required to confirm and refine those estimates. The source could be larger due to the neglected contribution of hydroperoxycarbonyls formed from higher anthropogenic NMVOCs (e.g. higher ketones and their precursors) and possibly monoterpenes. Moreover, the contribution of acetaldehyde photooxidation could be much higher than estimated here, considering the large underestimation of its calculated concentrations at remote locations (Read et al., 2012).

Evaluation of MAGRITTE and of its new chemical mechanism against the SEAC[4]RS campaign measurements indicates a good overall model performance for the main isoprene oxidation products. Heterogeneous reactions of IEPOX and organic nitrates on aerosols are a large area of uncertainty, with suggestions of heterogeneous sink overestimation for tertiary organic nitrates and sink underestimations for other isoprene nitrates. The total $RONO_2$ concentrations are underestimated by about 40%, possibly due to misrepresentations of nitrates from e.g. monoterpenes and anthropogenic precursors. The low observed $CH_3ONO_2$ levels are well reproduced by the model, providing a strong
indication for a very low nitrate yield ($< 3 \cdot 10^{-4}$) in the $CH_3O_2$+NO reaction.

*Code and data availability.*  The chemical mechanism is available at http://doi.org/10.18758/71021042 in KPP (Kinetic Pre-Processor) format (last access: 15 April 2019), including equation and species files, fortran code for calculating the reaction rates, and absorption cross-sections data files for polyfunctional carbonyls. Other relevant subroutines of the MAGRITTE model can be made available upon request (email: Jean-Francois.Muller@aeronomie.be). The SEAC[4]RS airborne trace gas measurements are available from the NASA LaRC Airborne
Science Data for Atmospheric Composition (https://www-air.larc.nasa.gov/missions/merges/, last access: 15 April 2019).

*Author contributions.*  JFM and JP elaborated the mechanism and drafted the manuscript, JFM and TS conducted the model calculations, all authors analysed the model results.

*Competing interests.*  The authors declare that they have no conflict of interest.

*Acknowledgements.*  This research was supported by the Belgian Science Policy Office through the projects TROVA (2016–2018) within
the ESA/PRODEX programme, OCTAVE (2017–2021) within the BRAIN-be research programme, and BIOSOA within the SSD program (2006-2010). We gratefully acknowledge the PIs and data managers of the NASA SEAC[4]RS campaign for the measurements used in this work.

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

**Table 1.** Chemical species of the oxidation mechanism of isoprene, monoterpenes and methylbutenol (MBO).

**Table 2.** Chemical mechanism and rates. Units for $1^{st}$-, $2^{nd}$-, and $3^{d}$-order reactions are $s^{-1}$, $cm^3 molec.^{-1} s^{-1}$ and $cm^6 molec.^{-2} s^{-1}$. Read $2.7(-11)$ as $2.7 \cdot 10^{-11}$; $T$=temperature (K); $[M]$ is air density (molec.$cm^{-3}$); $K_{RO2NO} = 2.7(-12) \exp(350/T)$; the PAN-like compounds formation and decomposition rates are calculated with $k = \frac{k_0[M]}{1+k_0[M]/k_\infty} 0.3^{\{1+[\log_{10}(k_0[M]/k_\infty)/1.414]^2\}^{-1}}$. References: 1, MCM (Saunders et al., 2003; Jenkin et al., 2015); 2, Nguyen et al. (2016); 3, Wennberg et al. (2018); 4, Liu et al. (2013); 5, Peeters and Müller (2010); 6, Capouet et al. (2004); 7, Atkinson et al. (2006); 8, Peeters et al. (2014); 9, St. Clair et al. (2016); 10, D'Ambro et al. (2017); 11, Lee et al. (2014); 12, Jacobs et al. (2014); 13, Paulot et al. (2009b); 14, Bates et al. (2016); 15, Schwantes et al. (2015); 16, Xiong et al. (2016); 17, Crounse et al. (2012); 18, Gross et al. (2014); 19, Burkholder et al. (2015); 20, Nguyen et al. (2015a); 21, Galloway et al. (2011); 22, Praske et al. (2015); 23, Vu et al. (2013); 24, Baeza-Romero et al. (2007); 25, Magneron et al. (2005); 26, Taraborrelli et al. (2012); 28, So et al. (2014); 29, Assaf et al. (2016); 30, Assaf et al. (2018); 31, Müller et al. (2016); 32, Allen et al. (2018); 34, Chan et al. (2009).

**Table 3.** Photodissocation reactions. The last column gives the photorate ($J$) calculated using the TUV model (Madronich, 1993) for a zenith angle of $30°$ and 300 DU ozone. References: 1, Burkholder et al. (2015); 2, Röth and Ehhalt (2015); 3, Shaw et al. (2018); 4, Pinho et al. (2005); 5, Jenkin et al. (2015); 6, Atkinson et al. (2006); 7, Liu et al. (2018); 8, Müller et al. (2014); 9, Barnes et al. (1993); 10, Xiong et al. (2016); 11, Liu et al. (2017); 12, Nakanishi et al. (1977); 13, Back and Yamamoto (1985).

**Table 4.** Heterogeneous reactions on aqueous aerosols. $\gamma$ denotes the reactive uptake coefficient. References: 1, Liggio et al. (2005); 2, Marais et al. (2016); 3, Fisher et al. (2016); 4, Müller et al. (2016). Notes:$a$) The dependence on aerosol pH (Marais et al., 2016; Stadtler et al., 2018) is ignored.

**Table 5.** Global sources of HC(O)OH in the model simulation.

**Table 6.** Global sources of $CH_3C(O)OH$ in the model simulation.

**Table 7.** Global sources of glyoxal in the model simulation.

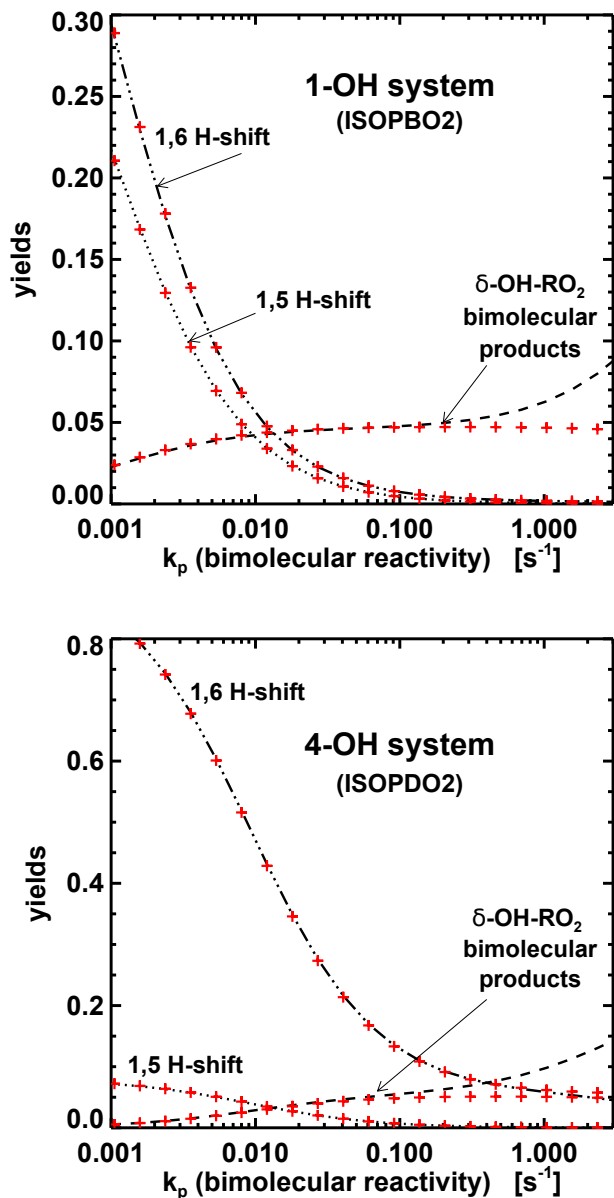

**Figure 1.** Contributions of H-shift isomerisations and $\delta$-OH-peroxy bimolecular reactions to the total reactivity of isoprene peroxy radicals resulting from addition to carbon 1 (top panel) and 4 (lower panel), as function of their bimolecular reactivity, at 295 K (Wennberg et al., 2018). The red crosses denote the yields of the parameterization used in the MAGRITTE mechanism.

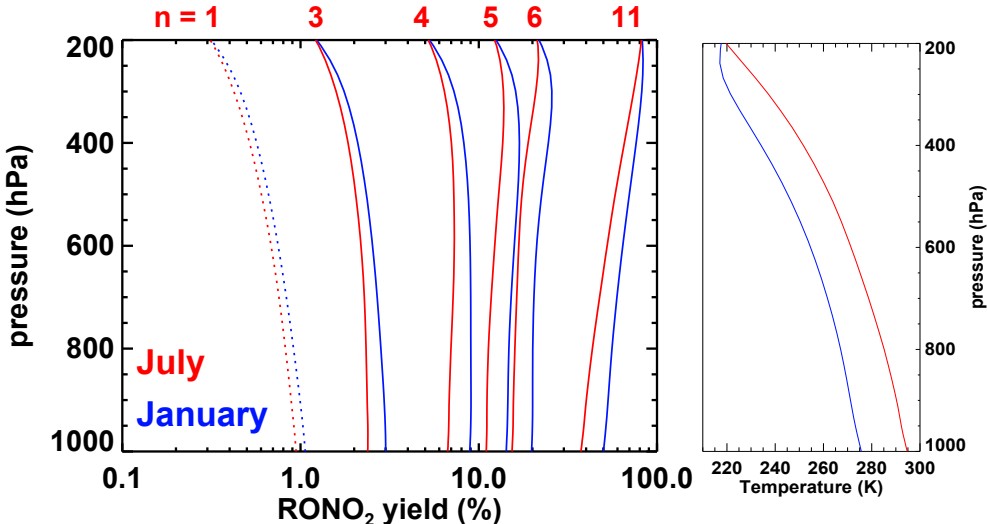

**Figure 2.** Left panel: Organic nitrate yield in the reaction of peroxy radicals with NO calculated following Wennberg et al. (2018) as function of atmospheric pressure, using temperature profiles typical of January (in blue) and July (in red) at $40°$ N (zonal average of ECMWF analyses). The temperature profiles are shown on the right panel. $n$ is the number of heavy atoms in the peroxy radical. For $n = 1$, the yield is calculated with $Z=1$ in Eq. 3.

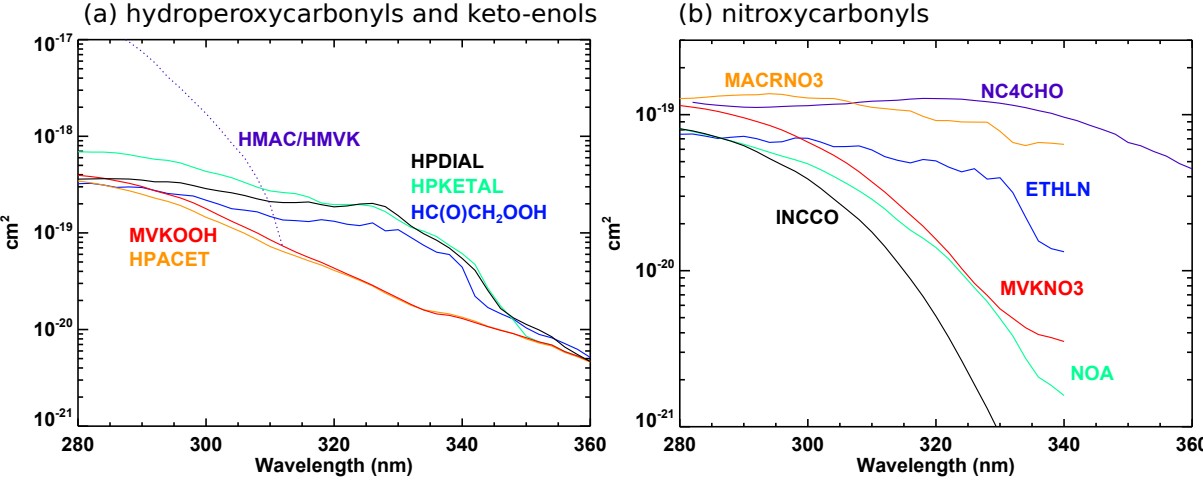

**Figure 3.** Absorption cross sections (in $cm^2$ molec.$^{-1}$) of (a) hydroperoxycarbonyls and keto-enols (HMAC and HMVK), and (b) nitrooxy-carbonyls. Species notation as in Table 1.

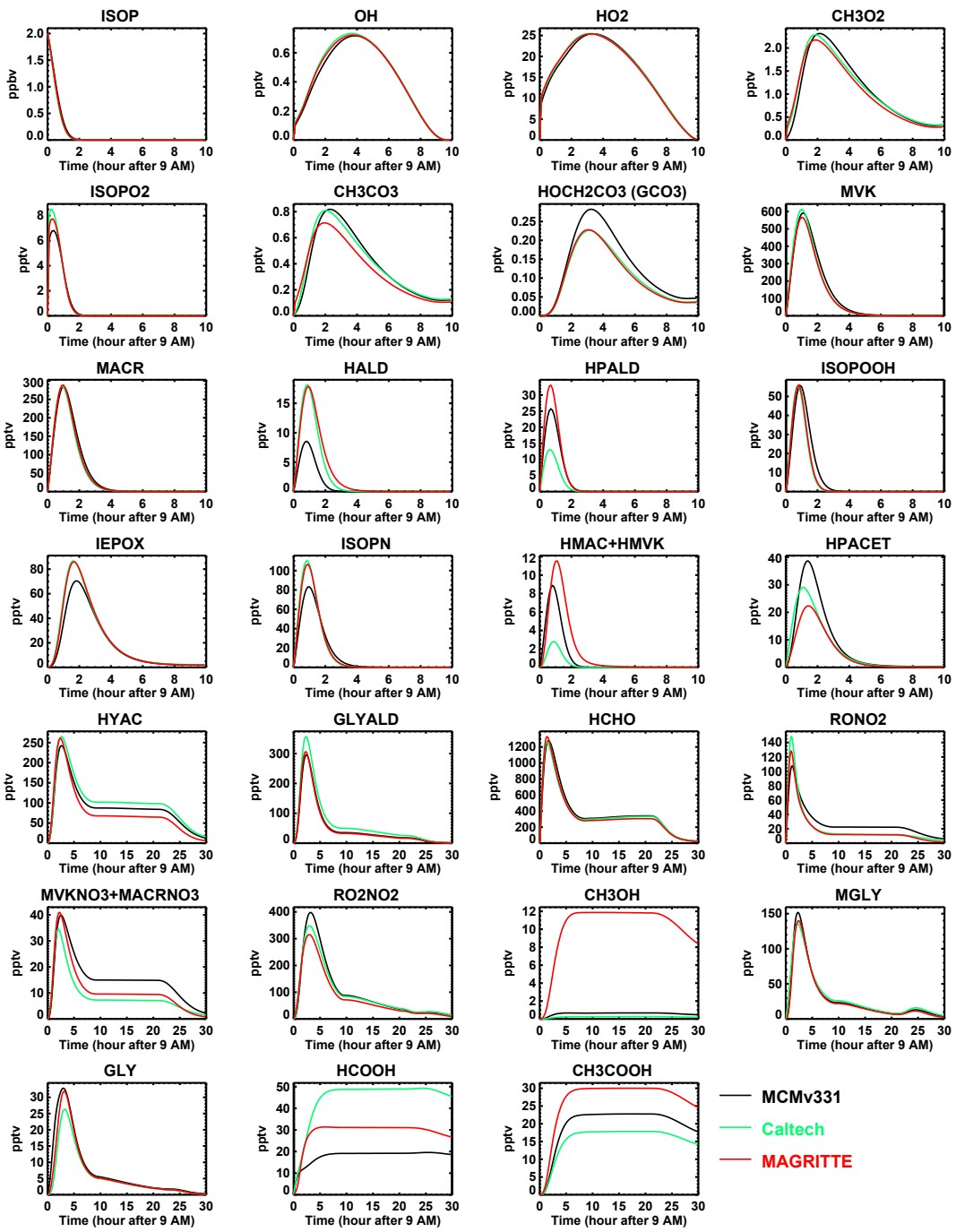

**Figure 4.** Box-model calculated mixing ratios of key compounds at 1 ppbv NOx. MCM results in black, Caltech mechanism in green, this work in red. ISOPN is the sum of isoprene hydroxynitrates), RONO2 the sum of organic nitrates), RO2NO2 the sum of PANs.

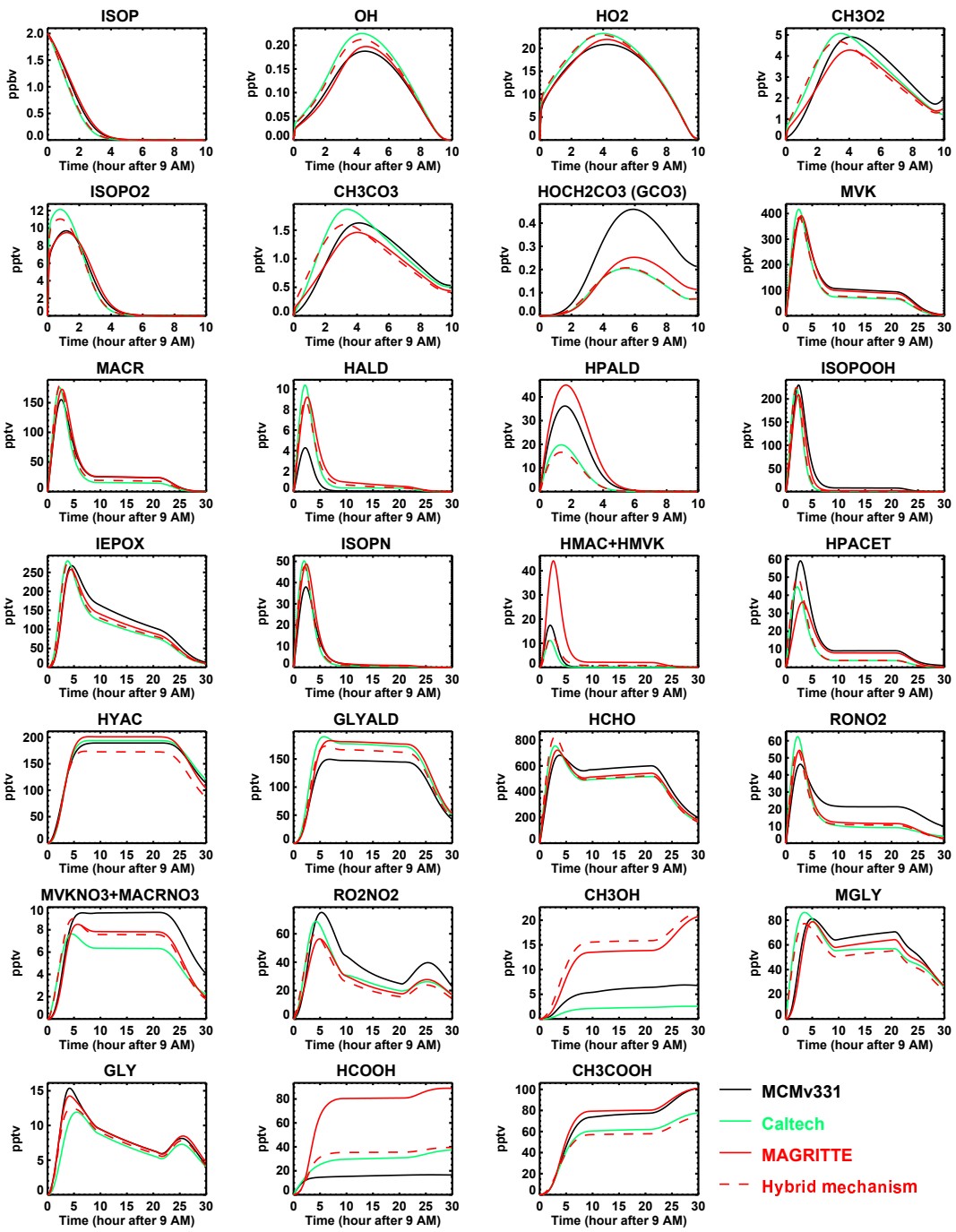

**Figure 5.** As Fig. 4, for 100 ppt NOx. The dashed red line corresponds to a simulation using the MAGRITTE mechanism with the Caltech representation of the isoprene peroxy 1,6 H-shift and of the hydroperoxycarbonyl photolysis reactions.

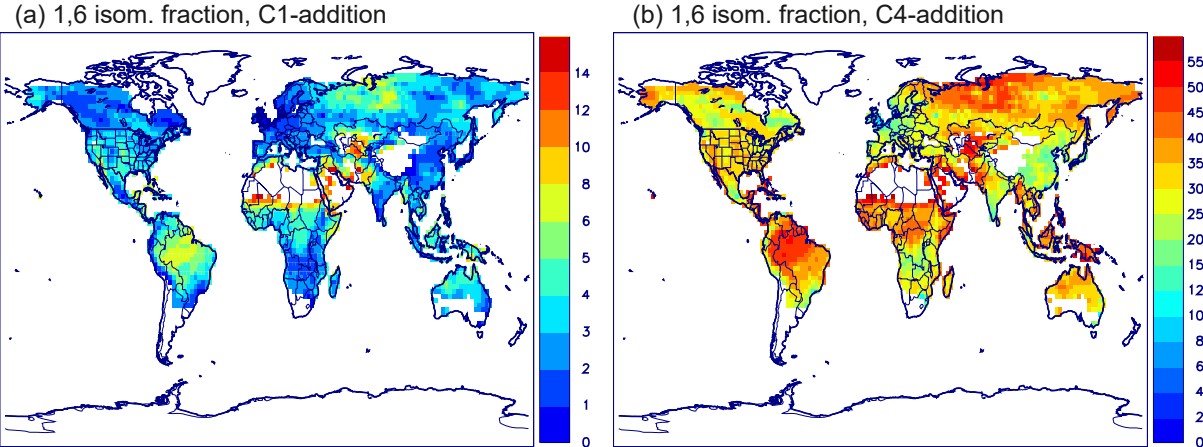

**Figure 6.** Calculated percentage contribution of Z-$\delta$-hydroxyperoxy 1,6 H-shift to the overall sink of the pool of peroxys resulting from addition of OH (a) to carbon C1, and (b) to carbon C4 of isoprene (column average, July 2013). Note the different color scales in (a) and (b).

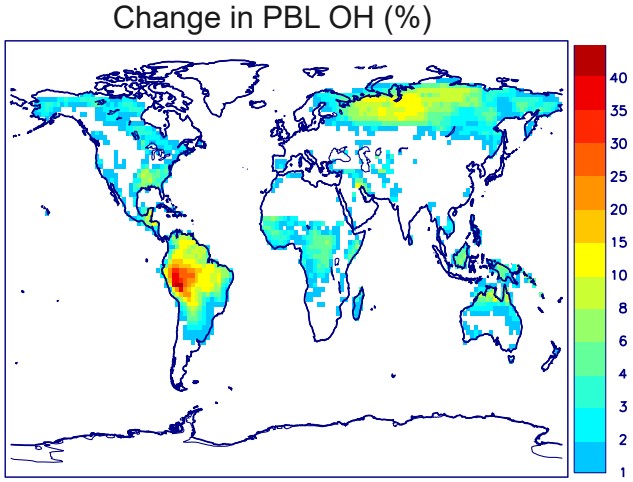

**Figure 7.** Calculated change (in %) in boundary layer OH concentration upon inclusion of isomerisation reactions of isoprene peroxy radicals (column average, July 2013).

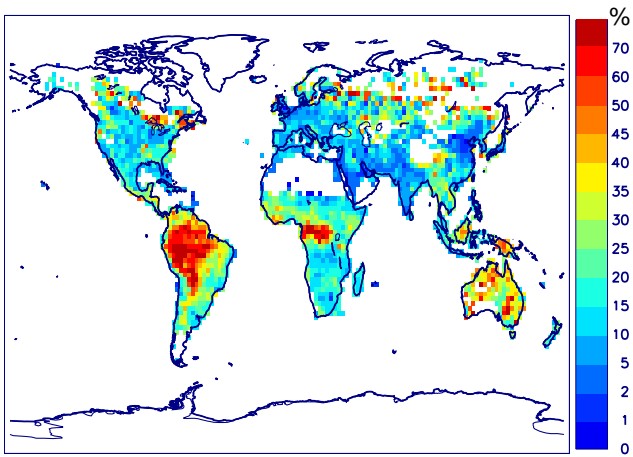

**Figure 8.** Percentage ratio of annual NOx net loss due to organic nitrate formation (i.e., their combined aerosol sink and deposition sink) to the total annual NOx emission. Blank areas are those with annually-averaged NOx emissions lower than $5 \cdot 10^9$ molec. cm$^{-2}$ s$^{-1}$.

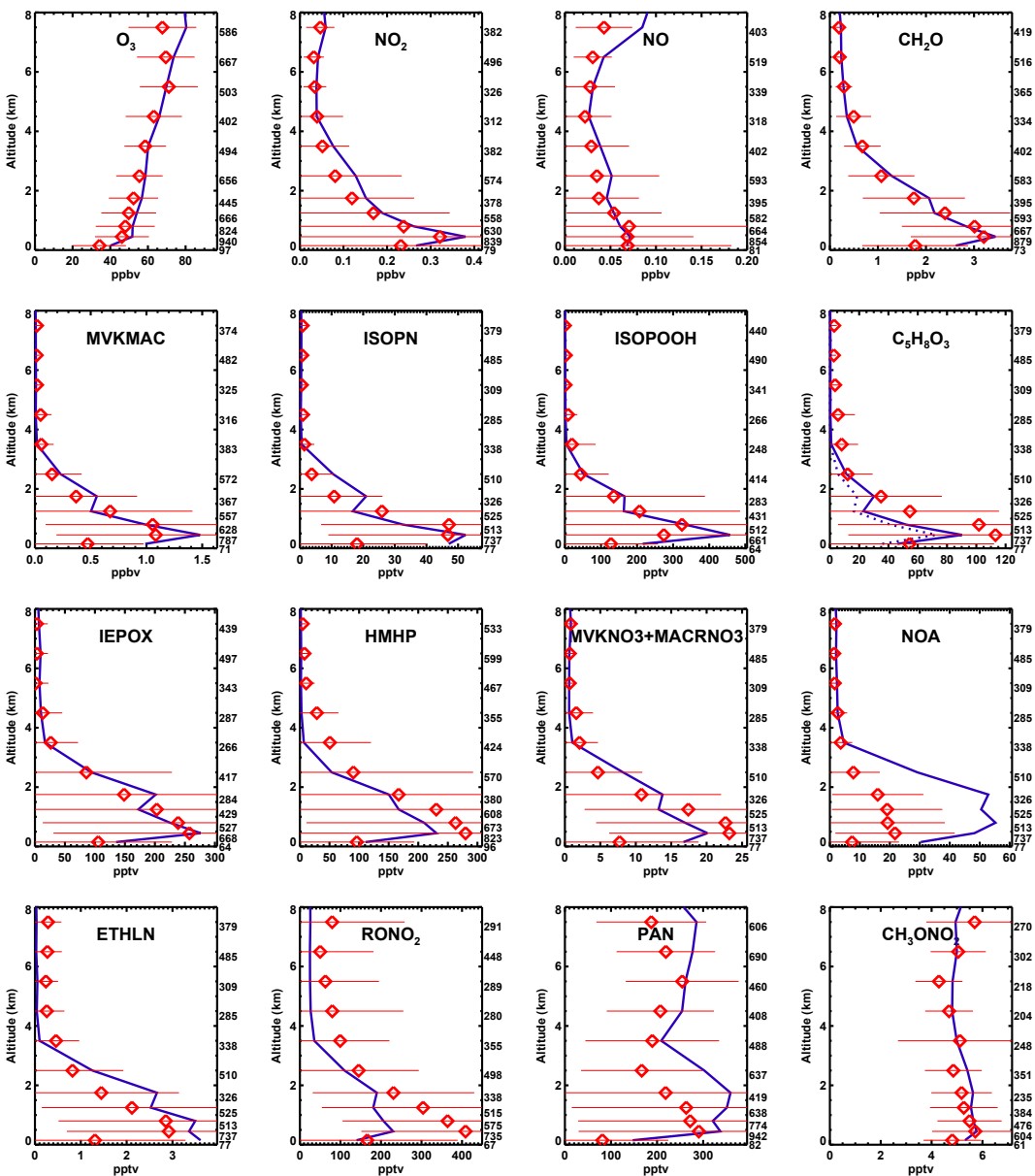

**Figure 9.** Observed (red symbols) and modelled (black lines) mean profiles of ozone, $NO_2$, NO, and major VOC oxidation products over North America during the SEAC$^4$RS campaign. The number of measurements per altitude bin is indicated on the right for each plot. The vertical bin interfaces are 0, 0.3, 0.6, 1, 1.5 km, and from 2 to 8 km by 1 km. The horizontal lines indicate the standard deviation of the measurements within each vertical bin. MVKMAC stands for the sum MVK+MACR+0.44 ISOPOOH. Both the modelled HPALD (dotted line) and HPALD+ICHE (solid line) are shown on the $C_5H_8O_3$ panel.

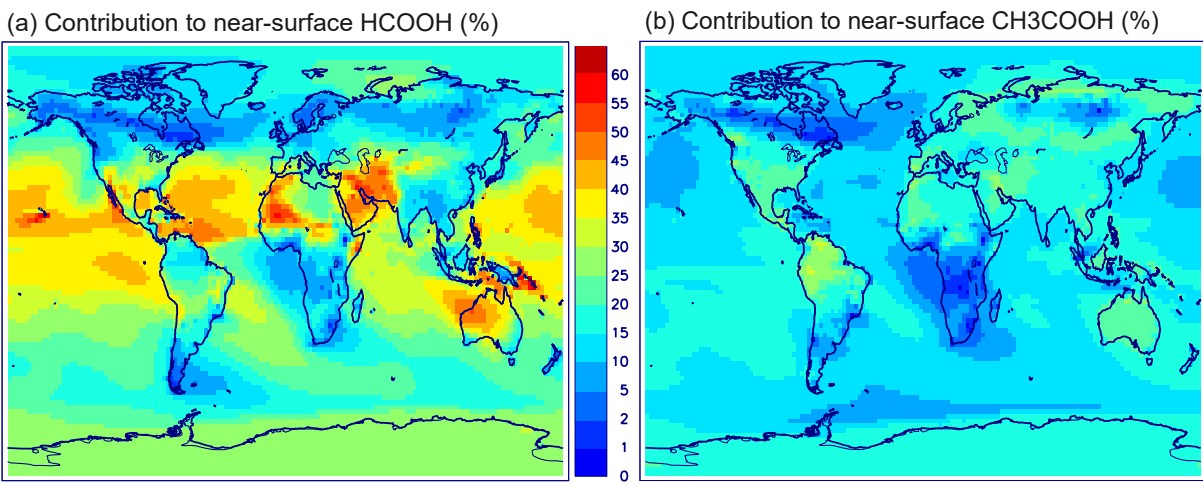

**Figure 10.** Calculated percentage contribution of hydroperoxycarbonyl photolysis to near-surface concentrations of (a) formic and (b) acetic acid for the month of July.