# Peer review of "Chemistry and deposition in the Model of Atmospheric composition at Global and Regional scales using Inversion Techniques for Trace gas Emissions (MAGRITTEv1.1). Part A. Chemical mechanism"

_Geoscientific Model Development, 2018_

## Referee Comment (RC1) · Anonymous Referee #1 · 18 Jan 2019

This manuscript describes simplified gas-phase oxidative chemical mechanisms of isoprene and other biogenically emitted hydrocarbons, and their incorporation into a new chemical transport model for use at global and regional scales. The introduction provides a detailed and comprehensive summary of the importance of recent advances in our understanding of the isoprene mechanism and the need for up-to-date mechanisms for use in simulations. This is followed by a lengthy description of the isoprene mechanism employed here, which is largely derived from a recently published review

(Wennberg et al. 2018) but with a couple notable alterations derived from recent work not incorporated into the review. The description of the mechanism is divided between text sections, which contain broader overviews of mechanistic pathways and the major recent changes due to new advancements, and the extensive footnotes of the table containing the complete mechanism. A subsequent section is dedicated to the results of regional and global simulations using this newly developed mechanism, including the general effects of isoprene oxidation on tropospheric composition, comparison to measurements of isoprene oxidation products in the southeast US, and discussions of the global budgets of formic acid, acetic acid, and glyoxal.

The well-recognized importance of isoprene to tropospheric chemistry means that it's always beneficial to have an up-to-date assessment of the isoprene mechanism, which this study provides. While much of this manuscript simply pulls together results from previous work, it does provide an important benchmark of both isoprene chemistry in the global atmosphere and the sources of gas-phase organic acids, which continue to elude explanation. However, it leaves the reader with a number of concerns regarding some elements included in the model but only given a cursory treatment, along with a need for more concrete discussions of the uncertainties associated with the implementation of this mechanism in global and regional models. These concerns are detailed below in reference to the locations in the manuscript at which they appear, but briefly, they generally include the effects of specific (particularly the more poorly-understood) isoprene oxidation pathways on model outcomes along with the treatment of SOA and monoterpenes. The authors acknowledge that all of these aspects come with substantial uncertainty, but without any quantification of that uncertainty in the model, it is difficult for the reader to know how much proverbial stock to put into simulation results. It is perfectly reasonable that the model is not intended to provide detailed accounting of global biogenic SOA formation or the effects of terpene oxidation, but more discussion is needed regarding the limitations of these aspects of the model. For example, while some SOA formation pathways are included in the model (e.g. IEPOX reactive uptake), others are only mentioned as the likely sinks of isoprene oxidation

products, without any physical reaction parameterized in the model (e.g. dinitrates and the hydroperoxy-epoxides from D'Ambro et al. (2017)). A modeler could assume that these products form SOA immediately with a 100% yield, but because some of the low-volatility species that lead to SOA formation would likely also have high deposition velocities, this could overestimate the SOA yield. How do these (and other) sources of uncertainty in the important results from this mechanism (e.g. HOx budgets, organic acid budgets, SOA) manifest themselves in a global model, and what should the reader take away regarding the potential for bias and error in simulating the overall effects of isoprene's oxidation on the chemistry of the global troposphere?

Specific questions and comments:

p4 L17: 5% can still be a lot of carbon for isoprene! Other pathways that also account for less than 5% of the total carbon from isoprene are included in this mechanism. Are there other reasons the bimolecular reactions of the delta-hydroxyperoxy radicals are excluded? Do we have any knowledge of how much this simplification might bias the results of simulations using this model?

p4 L25 & the rest of this section: the discussion of remaining uncertainties in this mechanism pathway is welcome, but given the important effects of this pathway on simulation results (e.g. as a source of HOx radicals in the otherwise HOx-consuming mechanism), it would be useful to provide the reader with some discussion (perhaps in the model results section) of how these uncertainties manifest themselves in the model. What range of possible HOx recycling rates would be compatible with what is currently known about this part of the mechanism? Given the uncertainties, can the boundary layer OH change due to this mechanistic pathway (Figure 3) be considered a bound or a best guess, and is the uncertainty on that at all quantifiable?

p9 L15: Fig 1 doesn't show latitude dependence as claimed here. It also convolutes the pressure and temperature dependences in a way that might not be useful for readers who would like to extrapolate for conditions other than 40 degrees N in January and

[Figure]

July. I would suggest either clarifying some of the details of these conditions (e.g. add side plots of temp & pressure vs. altitude in January and July, or separate this into plots of yield vs. temp and yield vs. pressure).

p9 L19: "such" should be "this". Also, Wennberg et al. (2018) does not show that this procedure inherently overestimates most measured nitrate yields, though it does suggest that this equation provides yields that seem excessive for dinitrates. Instead it goes off the recommendation of Teng et al. (2017), which explicitly says that this provides a better estimate than just n=#C, and improves this with a structure-activity-relationship-style modification.

p11: why is PAN included in C1 compounds? Also, this table would benefit from more clarification on what some of these groups mean; is OOH hydroperoxide or acid? Is OOOH trioxide or peracid? I realize this can usually be inferred from the hydrogen balance but it'd still be useful.

p26 L3: what is "isoprene-OH segregation"? What was the logic behind the 7% minor addition channel, and behind including those but not the E/Z-delta bimolecular products?

p26 L10: if you include the description of Y(Arey) in the table heading, you don't need it in the footnote; I think the table heading should be shorter and this could be a footnote. Also, don't all the scaling factors >1 suggest that N=#heavy atoms would've been better than N=#C?

p26 L11: define "room conditions" (this also comes up on p27 L4 & 13, p31 L15, 18 & 25, p32 L20, and p41 L34.

p26 L14: Does this inherently assume that all DHHEPOX is lost to aerosols? Are there any estimates of the OH reaction coefficient or uptake coefficient of DHHEPOX that might put this assumption in context, or provide the reader with some idea as to the uncertainty on this assumption? What fraction of carbon is lost to this pathway under

atmospheric conditions (and to other dead-end pathways assumed to either deposit or partition to aerosols, e.g. the dinitrates discussed in N9), and what is the resulting contribution to aerosol compared to other pathways (e.g. IEPOX)?

p26 L22: despite this being "well known" I think it deserves a citation, and more support than that the "majority" of exothermicity is alternately directed, making it "appear unlikely". It is difficult from reading this footnote to tell what is conjecture and what has experimental evidence to support the pathways used. The language generally implies certainty, but the lack of citations suggests that it is conjecture.

p32 L1: Here you state that H-abstraction from the carbon dominates, but with a higher yield of HCHO than HCOOH, the former of which is derived from H-abstraction from the hydroperoxide, isn't that backwards? Also, within their reported uncertainty, Allen et al. (2018) did not conclusively state that one path dominates over the other.

p32 L3: "he" should be "the"

p32 L5: the discussion here seems more suited for a subsection of section 2 than for a note at the bottom of a table; the generic monoterpene oxidation scheme provided here needs more discussion of its uncertainties and how the specific numbers were arrived at. While the complexity of terpene oxidation and the relative lack of quantitative knowledge about its oxidation mechanism make drastic simplification a necessity, it is not clear to the reader why this particular set of simplifications is ideal, or what the reasonable uncertainty bounds are on any of these rates and product yields. If pinene (or generic terpene) oxidation is to be included in this model, it should be given more than a paragraph in a footnote. The same might be said for MBO, but the relative simplicity of its oxidation mechanism, the overlap with isoprene oxidation products, and the smaller magnitude of its emissions make it less prone to substantial uncertainty and bias from mechanistic simplifications and shortcuts. Maybe some sensitivity studies showing the range of results you could get in a global model given the uncertainties in this mechanism would be most useful? (e.g. assuming more of less of the products

are lost to SOA/deposition, or assuming a range of nitrate and/or HOx yields). Some specific questions include: on line 10, where does the 45% number come from? How do these simulated acetone and formaldehyde yields compare to previous work? What does this mechanism inherently assume for the SOA formation from pinene, and how does that compare to both measured yields and the magnitude of SOA formed from isoprene globally? What might be the implications in a global simulation of skipping oxidative steps (and therefore likely sinks of OH, HO2, NO) in the oxidation mechanism, as is presumably the case when only one generation of products are used?

p37 L14: How is the oxidative degradation of anthropogenic NMVOCs treated in the model? I am particularly concerned about the possible contribution of degradation of non-isoprene compounds to the gas-phase budgets of glyoxal and the organic acids. Along those same lines, could some discussion of the potential for additional sources of the gas-phase organic acids not included in this model (e.g. degradation of other compounds, revolatilization from SOA) be added to those sections?

p37 L18: Are there primary emissions of MBO in the model? What is the effect of the MBO oxidation mechanism (and the terpene mechanism) in the model?

p40 L5: In the comparisons to SEAC4RS data, it would be helpful to list the measurement uncertainties and spreads alongside the over/underestimations of the model. Also, what exactly is the model output being compared with the measurements in this section? Are the simulated average profiles just the average over the entire SE USA between 0900 and 1700 hours, or are they points subsampled from the model concurrent with the flight paths? If it's the former, which assumes that the SEAC4RS observations (masked for plumes and stratospheric intrusions) are representative of the averaged regions, how might this skew the comparison between the model and measurements?

p41 L5: Are these non-HPALD compounds also isoprene products? Do we have any indication as to what they are? If they have the same mass as the HPALDs, are there other species in your mechanism that also have this mass that may account for this

mass?

Citations:

Allen, H. M., Crounse, J. D., Bates, K. H., Teng, A. P., Krawiec-Thayer, M. P., Rivera-Rios, J. C., Keutsch, F. N., St. Clair, J. M., Hanisco, T. F., Moller, K. H., Kjaergaard, H. G., and Wennberg, P. O.: Kinetics and product yields of the OH initiated oxidation of hydroxymethyl hydroperoxide, J. Phys. Chem. A, 122, 6292–6302, 2018.

D'Ambro, E. L., Moller, K. H., Lopez-Hilfiker, F. D., Schobesberger, S., Liu, J., Shilling, J. E., Kjaergaard, H. G., and Thornton, J. A.: Isomerization of second-generation isoprene peroxy radicals: Epoxide formation and implications for Secondary Organic Aerosol yields, Environ. Sci. Technol., 51, 4978–7987, 2017.

Teng, A. P., Crounse, J. D., and Wennberg, P. O.: Isoprene peroxy radical dynamics, J. Am. Chem. Soc., 139, 5367-5377, 2017.

Wennberg, P. O., Bates, K. H., Crounse, J. D., Dodson, L. G., McVay, R., Mertens, L. A., Nguyen, T. B., Praske, E., Schwantes, R. H., Smarte, M. D., St Clair, J. M., Teng, A. P., Zhang, X., and Seinfeld, J. H.: Gas-phase reactions of isoprene and its major oxidation products, Chem. Rev., 118, 3337–3390, DOI: 10.1021/acs.chemrev.7b00439, 2018.

––––––––––––––––––––

---

## Referee Comment (RC2) · Anonymous Referee #2 · 24 Jan 2019

The present manuscript details the condensation of the isoprene oxidation mechanism which has been subject of revived and intense research, both experimental and theoretical, in the last decade. The authors integrated all the major advancements and originally contributed to large portion of them. Their critical understanding of the relevant chemical processes, far from being all achieved, adds to the value of manuscript. Impact of recent experimental and theoretical advancements of the global budgets of organic acids is very interesting. The model is fairly well described and the evaluation

seems appropriate for use in global models. However, a box model comparison between MAGRITTEv1.0 and MCMv3.3.1, the mechanism presented by Wennberg et al. 2018 or even their detailed mechanism would add useful information about the model performance. I wish the authors could provide such data and information.

I have two major concerns.

1) Bulk isomerization rates

Please explain more the counter-intuitive concept by which the bulk isomerization rate of the lumped (beta- and delta-) species ISOPBO2 and ISOPDO2 should linearly increase with the traditional RO2 sink rate (kp). Why is it not or it has to be different than what Crounse et al. (2011) reported? Even if correct, neglecting the RO2 sink due to permutation reactions should yield non-negligible errors/deviations from the analytical solution. Please explain why the neglect and in case provide an estimate of the deviation caused by it.

2) Reproducibility of results

The chemical mechanism of MAGRITTEv1.0 is not exactly what can be downloaded at the link given. A few sample differences are listed below.

The reaction of CH3OH with OH is standard in the manuscript but in MAGRITTE.eqn file one finds two reactions with one including the water vapor catalysis by Jara-Toro et al. 2017.

The rate constant for the reaction

CH3O2 + HO2 = 0.9 CH3OOH + 0.1 HCHO

is 4.1E-13*exp(750/TEMP) and 3.8E-13*exp(780/TEMP), respectively.

Concerning the 1,6-H-shift of ISOPDO2 in the .eqn file one finds

ISOPDO2 = 0.25 HO2 + 0.25 HPALD2 + 0.75 OH + 0.75 CO + 0.75 DIHPCHO :

4.253E8*exp(-7254/TEMP) ;

ISOPDO2 + NO = NO + 0.25 HO2 + 0.25 HPALD2 + 0.75 OH + 0.75 CO + 0.75 DIHPCHO : 6.29E-19*exp(4012/TEMP) ;

ISOPDO2 + HO2 = HO2 + 0.25 HO2 + 0.25 HPALD2 + 0.75 OH + 0.75 CO + 0.75 DIHPCHO : 4.90E-20*exp(4962/TEMP) ;

The last two reactions constants are not the ones reported in Table 2.

PYRA (pyruvic acid) is listed in Table 1. However, it is neither in Table 2 nor in the .eqn file.

Overall, it might be that the authors uploaded another version of MAGRITTE. Please upload a v1.0 that is faithful to the Tables in the manuscripts. The files should bear the information about the exact model version.

Moreover, no file with the actual functions used for many rate constants is given. This is also the case for the cross-sections and quantum yields used for computing the photolysis frequencies. Please also provide this information.

---

## Referee Comment (RC3) · Anonymous Referee #3 · 3 Feb 2019

This paper compiles a new set of isoprene oxidation mechanism for a global model, based on recent laboratory and theoretical developments. The authors did a thorough job on explaining chemical reactions on major pathways. I find Section 2 is particularly useful for future model development. However, I feel that Section 3 could be improved by including more discussion. I have a few comments:

1. The authors use SEAC4RS dataset for their model evaluation, and compare their

results to Fisher et al. (2016) extensively for RONO2 budgets and speciation. It should be pointed out that this paper uses a RONO2 yield of 13%-14% from isoprene RO2+NO reaction, in contrast to 9% assumed in Fisher et al. (2016). Such difference would presumably lead to significant differences between these two models. I believe some caveats should be provided in the text to make reader aware of these differences.

2. Similar to Fisher et al. (2016), the authors find a model underestimate of RONO2, as shown in their Figure 5. A recent study by Li et al. (2018), suggests that a large part of discrepancy could be due to terpene nitrates and nighttime isoprene nitrates. In particular, the authors assume a 100% recycling of NOx from APINONO2 + OH. This choice may have a large impact on total RONO2. For nighttime chemistry, the authors have ignored the formation of dinitrate (N31 for Table 2), which could also contribute to RONO2, according to Li et al. (2018). Some discussion on the uncertainties of terpene nitrates and nighttime isoprene nitrates, should be included in the text.

3. The reader is also wondering how this model performs on HNO3 and PAN, which are major NOy reservoirs. Examining these species may help to justify the 60% reduction of U.S. NOx emission inventories in their model.

4. It seems that Section 3.4, Global budget of formic and acetic acid, is disconnected from the rest of the paper. It appears that the authors want to recalculate the global budget of these two acids, without any comparison to field observations. It is unclear how this new mechanism has improved current knowledge on formic and acetic acid. Some model sensitivity tests and comparison to observations would be useful.

5. While reading Section 3.4, the authors suggest CH3CO3+HO2 is the major source of CH3COOH. This seems like another good reason to examine PAN in their model.

6. Given the extensive research on isoprene oxidation over Southeast US, the authors should include two review papers on this topic in their introduction, Carlton et al. (2018) and Mao et al. (2018).

Reference

Carlton, A. G., Gouw, J. d., Jimenez, J. L., Ambrose, J. L., Attwood, A. R., Brown, S., Baker, K. R., Brock, C., Cohen, R. C., Edgerton, S., Farkas, C. M., Farmer, D., Goldstein, A. H., Gratz, L., Guenther, A., Hunt, S., Jaeglé, L., Jaffe, D. A., Mak, J., McClure, C., Nenes, A., Nguyen, T. K., Pierce, J. R., Sa, S. d., Selin, N. E., Shah, V., Shaw, S., Shepson, P. B., Song, S., Stutz, J., Surratt, J. D., Turpin, B. J., Warneke, C., Washenfelder, R. A., Wennberg, P. O., and Zhou, X.: Synthesis of the Southeast Atmosphere Studies: Investigating Fundamental Atmospheric Chemistry Questions, Bull. Amer. Meteorol. Soc., 99, 547-567, 10.1175/bams-d-16-0048.1, 2018.

Fisher, J. A., Jacob, D. J., Travis, K. R., Kim, P. S., Marais, E. A., Chan Miller, C., Yu, K., Zhu, L., Yantosca, R. M., Sulprizio, M. P., Mao, J., Wennberg, P. O., Crounse, J. D., Teng, A. P., Nguyen, T. B., St. Clair, J. M., Cohen, R. C., Romer, P., Nault, B. A., Wooldridge, P. J., Jimenez, J. L., Campuzano-Jost, P., Day, D. A., Hu, W., Shepson, P. B., Xiong, F., Blake, D. R., Goldstein, A. H., Misztal, P. K., Hanisco, T. F., Wolfe, G. M., Ryerson, T. B., Wisthaler, A., and Mikoviny, T.: Organic nitrate chemistry and its implications for nitrogen budgets in an isoprene- and monoterpene-rich atmosphere: constraints from aircraft (SEAC4RS) and ground-based (SOAS) observations in the Southeast US, Atmos. Chem. Phys., 16, 5969-5991, 10.5194/acp-16-5969-2016, 2016.

Li, J., Mao, J., Fiore, A. M., Cohen, R. C., Crounse, J. D., Teng, A. P., Wennberg, P. O., Lee, B. H., Lopez-Hilfiker, F. D., Thornton, J. A., Peischl, J., Pollack, I. B., Ryerson, T. B., Veres, P., Roberts, J. M., Neuman, J. A., Nowak, J. B., Wolfe, G. M., Hanisco, T. F., Fried, A., Singh, H. B., Dibb, J., Paulot, F., and Horowitz, L. W.: Decadal changes in summertime reactive oxidized nitrogen and surface ozone over the Southeast United States, Atmos. Chem. Phys., 18, 2341-2361, 10.5194/acp-18-2341-2018, 2018.

Mao, J., Carlton, A., Cohen, R. C., Brune, W. H., Brown, S. S., Wolfe, G. M., Jimenez, J. L., Pye, H. O. T., Lee Ng, N., Xu, L., McNeill, V. F., Tsigaridis, K., McDonald, B. C.,

Warneke, C., Guenther, A., Alvarado, M. J., de Gouw, J., Mickley, L. J., Leibensperger, E. M., Mathur, R., Nolte, C. G., Portmann, R. W., Unger, N., Tosca, M., and Horowitz, L. W.: Southeast Atmosphere Studies: learning from model-observation syntheses, Atmos. Chem. Phys., 18, 2615-2651, 10.5194/acp-18-2615-2018, 2018.

———————————————————

---

## Author Comment (AC2) · 17 Apr 2019

**Reply to Anonymous Referee #2**

We thank the referee for their comments and respond to the points raised below.

*The authors integrated all the major advancements and originally contributed to large portion of them. Their critical understanding of the relevant chemical processes, far from being all achieved, adds to the value of manuscript. Impact of recent experimental and theoretical advancements of the global budgets of organic acids is very interesting. The model is fairly well described and the evaluation seems appropriate for use in global models. However, a box model comparison between MAGRITTEv1.0 and MCMv3.3.1, the mechanism presented by Wennberg et al. 2018 or even their detailed mechanism would add useful information about the model performance. I wish the authors could provide such data and information.*

We thank the Reviewer for the suggestion. We added a new section "Box model comparison with other isoprene mechanisms". We intercompare the MAGRITTE mechanism version 1.1, MCMv3.3.1 and the reduced Caltech mechanism. We don't believe useful to include the "full" version of the Caltech mechanism in this comparison, as it does not treat the further degradation of numerous oxidation products. We perform 30-hour simulations using KPP, starting at 9 AM with 2 ppbv isoprene. NOx is fixed at either 1 or 0.1 ppbv. The photolysis rates are calculated for mid-July clear-sky conditions at 30°N, using the TUV model of Madronich (1993). For computational efficiency, the photorates are parameterized as a function of solar zenith angle using MCM-type expressions (Saunders et al., 2003). All rate coefficient expressions are available at the MAGRITTE repository (http://doi.org/10.18758/71021042). Since Wennberg et al. does not provide detailed recommendations for the calculation of photolysis rates, we use our own expressions in their mechanism.

Note that the new version (v1.1) of the MAGRITTE mechanism differs from the initial version (v1.0) described in the GMDD paper. The most important updates include

(1) updated product distribution of the 1,6-H-shift isomerisation of the $Z$-$\delta$-OH-peroxys from ISOP+OH, including a higher HPALD yield (0.75 instead of 0.25), in agreement

with recent laboratory data (Berndt et al., 2019) and with theoretical calculations, as described in detail in the revised version of the manuscript (see also our Reply to Reviewer #1),

(2) inclusion of the bimolecular reactions of the $Z$-$\delta$-OH-peroxys from ISOP+OH, following a comment of Reviewer #1,

(3) calculation of RONO2 yields in RO2+NO reactions following Wennberg et al. (2018).

The comparisons show that MAGRITTEv1.1 leads to lower HOx recycling than the Caltech mechanism. Sensitivity calculations show that the difference is primarily due to (i) differences in the $Z$-$\delta$-OH-peroxy isomerisation rates and products, and (ii) differences in the product distribution of hydroperoxycarbonyl (especially HPACET and HPAC) photolysis. Important differences between the mechanisms are also found for e.g. carboxylic acids, PANS, nitrates and methanol, as discussed in the revised version of the manuscript.

*1) Bulk isomerization rates*
*Please explain more the counter-intuitive concept by which the bulk isomerization rate of the lumped (beta- and delta-) species ISOPBO2 and ISOPDO2 should linearly increase with the traditional RO2 sink rate (kp). Why is it not or it has to be different than what Crounse et al. (2011) reported? Even if correct, neglecting the RO2 sink due to permutation reactions should yield non-negligible errors/deviations from the analytical solution. Please explain why the neglect and in case provide an estimate of the deviation caused by it.*

We provide now a better justification of the bulk isomerisation rate expressions:

"Based on a detailed steady-state analysis, the bulk isomerisation rate of ISOPBO2 and ISOPDO2 was shown to increase linearly with the sink rate ($k_p$) of the traditional peroxy reaction (Peeters et al., 2014). The reason for this behaviour is that at low $k_p$, the ratio of the $Z$-$\delta$-OH-peroxys over the lower-energy $\beta$-OH-peroxys is close to their equilibrium ratio, of order of only ~0.01, whereas at the high $k_p$ limit, where all peroxys have a similar lifetime, their ratio is governed by their initial formation branching ratio, which is an order magnitude higher (Peeters et al., 2014; Teng et al., 2017)." Note that the linear dependence of bulk isomerisation rates on $k_p$ was verified experimentally by Teng et al. (2017).

Neglecting the $RO_2$ sink due to permutation reactions in those bulk isomerisation rate expressions has a negligible impact, estimated at ~0.6% of the bulk isomerisation rate for ISOPDO2, and even less for ISOPBO2.

*2) Reproducibility of results*
*The chemical mechanism of MAGRITTEv1.0 is not exactly what can be downloaded at the link given. A few sample differences are listed below. The reaction of CH3OH with OH is standard in the manuscript but in MAGRITTE.eqn file one finds two reactions with one including the water vapor catalysis by Jara-Toro et al. 2017.*

We thank the Reviewer for pointing this out. Updated equation and species files are now available at the MAGRITTE repository. The water vapor catalysis proposed by Jara-Toro et al. is not included, as it was recently disproved by a recent laboratory study (Chao et al., 2019).

*The rate constant for the reaction*

*CH3O2 + HO2 = 0.9 CH3OOH + 0.1 HCHO*

*is 4.1E-13\*exp(750/TEMP) and 3.8E-13\*exp(780/TEMP), respectively.*

Corrected.

*Concerning the 1,6-H-shift of ISOPDO2 in the .eqn file one finds*
*ISOPDO2 = 0.25 HO2 + 0.25 HPALD2 + 0.75 OH + 0.75 CO + 0.75 DIHPCHO :*
*4.253E8\*exp(-7254/TEMP) ;*
*ISOPDO2 + NO = NO + 0.25 HO2 + 0.25 HPALD2 + 0.75 OH + 0.75 CO + 0.75 DIHP-*
*CHO : 6.29E-19\*exp(4012/TEMP) ;*
*ISOPDO2 + HO2 = HO2 + 0.25 HO2 + 0.25 HPALD2 + 0.75 OH + 0.75 CO + 0.75*
*DIHPCHO : 4.90E-20\*exp(4962/TEMP) ;*

*The last two reactions constants are not the ones reported in Table 2.*

Corrected.

*PYRA (pyruvic acid) is listed in Table 1. However, it is neither in Table 2 nor in the .eqn file.*

Corrected (PYRA is not a model species).

*Overall, it might be that the authors uploaded another version of MAGRITTE. Please upload a v1.0 that is faithful to the Tables in the manuscripts. The files should bear the information about the exact model version.*

The new version of the mechanism (v1.1) supersedes version v1.0. The files now bears the information about the model version.

*Moreover, no file with the actual functions used for many rate constants is given. This is also the case for the cross-sections and quantum yields used for computing the photolysis frequencies. Please also provide this information.*

We thank the Reviewer for the excellent suggestion. We now provide the functions

used for calculations of rate constants (including photolysis rates as discussed above) in the MAGRITTE repository, as well as data files with the absorption cross-sections of polyfunctional carbonyls not found in current recommendations (IUPAC, JPL). The photolysis parameters of other compounds are readily available from e.g. those recommendations.

**References**

Berndt, T., Jokinen, T., Sipilä, M., Mauldin III, R. L., Herrmann, H., Stratmann, F., Junninen, H. and Kulmala, M.: $H_2SO_4$ formation from the gas-phase reaction of stabilized Criegee Intermediate with $SO_2$: Influence of water vapour content and temperature, Atmos. Environ., 89, 603–612, 2014.

Chao, W., Lin, J.-M., Takahashi, D., Thomas, A., Yu, L., Kajii, Y., Batut, S., Schoemaecker, C., and Fittschen, C.: Water vapor does not catalyze the reaction between methanol and OH radicals, Angew. Chem. Int. Ed., 131, 5067–5071, 2019.

Jara-Toro, R. A., Hernandez, F. J., Taccone, R. A., Lane, S. I., and Pino, G. A.: Water catalysis of the reaction between methanol and OH at 294 K and the atmospheric implications, Angew. Chem. Int. Ed., 56, 2166–2170, 2017.

Peeters, J., Müller, J.-F., Stavrakou, T., and Nguyen, S. V.: Hydroxyl radical recycling in isoprene oxidation driven by hydrogen bonding and hydrogen tunneling: the upgraded LIM1 mechanism, J. Phys. Chem. A, 118, 8625–8643, 2014.

Saunders, S. M., M. E. Jenkin, R. G. Derwent, and M. J. Pilling: Protocol for the

development of the Master Chemical Mechanism, MCM v3 (Part A): tropospheric degradation of non-aromatic volatile organic compounds, Atmos. Chem. Phys., 3, 161–180, 2003.

Teng, A. P., Crounse, J. D., Lee, L., St. Clair, J. M., Cohen, R. C., and Wennberg, P. O.: Hydroxy nitrate production in the OH-initiated oxidation of alkenes, Atmos. Chem. Phys., 15, 4297–4316, 2015.

Wennberg, P. O., Bates, K. H., Crounse, J. D., Dodson, L. G., McVay, R., Mertens, L. A., Nguyen, T. B., Praske, E., Schwantes, R. H., Smarte, M. D., St Clair, J. M., Teng, A. P., Zhang, X., and Seinfeld, J. H.: Gas-phase reactions of isoprene and its major oxidation products, Chem. Rev., 118, 3337–3390, 2018.

---

## Author Comment (AC3) · 17 Apr 2019

**Reply to Anonymous Referee #3**

We thank the referee for their comments and respond to the points raised below.
*1. The authors use SEAC4RS dataset for their model evaluation, and compare their results to Fisher et al. (2016) extensively for RONO2 budgets and speciation. It should be pointed out that this paper uses a RONO2 yield of 13%-14% from isoprene RO2+NO reaction, in contrast to 9% assumed in Fisher et al. (2016). Such difference would presumably lead to significant differences between these two models. I believe some caveats should be provided in the text to make reader aware of these differences.*

This is correct. This difference in RONO2 yield between the two studies is now mentioned in the discussion of the NOx loss through RONO2 formation (Sect. 4.2) and again in the evaluation of total RONO2 against SEAC[4]RS measurements (Sect. 4.3).

*2. Similar to Fisher et al. (2016), the authors find a model underestimate of RONO2, as shown in their Figure 5. A recent study by Li et al. (2018), suggests that a large part of discrepancy could be due to terpene nitrates and nighttime isoprene nitrates. In particular, the authors assume a 100% recycling of NOx from APINONO2 + OH. This choice may have a large impact on total RONO2. For nighttime chemistry, the authors have ignored the formation of dinitrate (N31 for Table 2), which could also contribute to RONO2, according to Li et al. (2018). Some discussion on the uncertainties of terpene nitrates and nighttime isoprene nitrates, should be included in the text.*

Thank you for these valid points. We include now a more complete discussion of the possible causes of RONO2 underestimation in the model: "There are several possible explanations for the discrepancy, including the neglected reactions of $NO_3$ with unsaturated oxidation products from isoprene and other BVOCs, the neglected formation of unsaturated dinitrates from the reaction of dinitroxyperoxy radicals (NISOPO2) with NO (Li et al., 2018), a possible overestimate of the tertiary nitrate hydrolysis sink (for dinitrates from ISOP+OH), and a misrepresentation of alkyl and hydroxyalkyl nitrates from other precursors than isoprene. The monoterpene nitrates are very crudely represented in the model. In particular, the assumption of 100% NOx recycling in their

reaction with OH could lead to a significant overestimation of RONO2 loss. "

*3. The reader is also wondering how this model performs on HNO3 and PAN, which are major NOy reservoirs. Examining these species may help to justify the 60% reduction of U.S. NOx emission inventories in their model.*

We now include PAN in the model comparison with SEAC$^4$RS (Fig. 9). A moderate model overestimation is found, similar to previous studies (Travis et al. 2016, Li et al. 2018). The model also overestimates HNO$_3$ measurements from SEAC$^4$RS, but reproduces well the average NO$_3^-$ wet deposition measurements over the U.S. (data obtained from R. Larson, NADP Database Manager, Wisconsin State Laboratory of Hygiene). A detailed discussion and justification of the NOx emission reduction (similar to a previous model study, Travis et al. 2016) is clearly beyond the scope of our study.

*4. It seems that Section 3.4, Global budget of formic and acetic acid, is disconnected from the rest of the paper. It appears that the authors want to recalculate the global budget of these two acids, without any comparison to field observations. It is unclear how this new mechanism has improved current knowledge on formic and acetic acid. Some model sensitivity tests and comparison to observations would be useful.*

As reported in the text, there is wide consensus that models underestimate formic and acetic acid abundances by large factors. This has been shown through numerous model comparisons with aircraft, ground-based and in situ measurements. Our study does not claim to reconcile models with observations. Since our newly-derived global sources of those acids are similar as (or even lower than) in previous modelling studies, it is clear that the large model underestimations remain, and extensive comparisons with atmospheric observations are not needed to make that point. We added the following sentence to Section 4.4: "Despite the newly-proposed large production of formic and acetic through hydroperoxycarbonyl photolysis, our derived total sources of those acids remains similar as (or even lower than) in previous modelling studies

(Paulot et al., 2011; Stavrakou et al., 2012; Millet et al., 2015; Khan et al., 2018), and are therefore insufficient to explain their high observed abundances."

*5. While reading Section 3.4, the authors suggest CH3CO3+HO2 is the major source of CH3COOH. This seems like another good reason to examine PAN in their model.*

PAN is now included in the model comparison against SEAC$^4$RS observations.

*6. Given the extensive research on isoprene oxidation over Southeast US, the authors should include two review papers on this topic in their introduction, Carlton et al. (2018) and Mao et al. (2018).*

Done.

**References**

Carlton, A. G., de Gouw, J., Jimenez, J. L., Ambrose, J. L., Attwood, A. R., Brown, S., Baker, K. R., Cohen, R. C., Edgerton, S., Farkas, C. M., Farmer, D., Goldstein, A. H., Gratz, L., Guenther, A., Hunt, S., Jaeglé, L., Jaffe, D. A., Mak, J., McClure, C., Nenes, A., Nguyen, T. K., Pierce, J. R., de Sa, S., Selin, N. E., Shah, V., Shaw, S., Shepson, P. B., Song, S., Stutz, J., Surratt, J. D., Turpin, B. J., Warneke, C., Washenfelder, R. A., Wennberg, P. O., and Zhou, X.: Synthesis of the Southeast Atmosphere Studies: Investigating fundamental atmospheric chemistry questions, Bull. Am. Met. Soc., 547–567, 2018.

Fisher, J. A., Jacob, D. J., Travis, K., Kim, P. S., Marais, E. A., Chan Miller, C., Yu, K., Zhu, L., Yantosca, R. M., Sulprizio, M. P., Mao, J., Wennberg, P. O., Crounse, J.

D., Teng, A. P., Nguyen, T. B., St. Clair, J. M., Cohen, R. C., Romer, P., Nault, B. A., Wooldridge, P. J., Jimenez, J. L., Campuzano-Jost, P., Day, D. A., Hu, W., Shepson, P. B., Xiong, F., Blake, D. R., Goldstein, A. H., Misztal, P. K., Hanisco, T. F., Wolfe, G. M., Ryerson, T. B., Wisthaler, A., and Mikoviny, T.: Organic nitrate chemistry and its implications for nitrogen budgets in an isoprene- and monoterpene-rich atmosphere: constraints from aircraft (SEAC[4]RS) and ground-based (SOAS) observations in the Southeast US, Atmos. Chem. Phys., 16, 5969–5991, 2016.

Khan, M. A. H., Lyons, K., Chhantyal-Pun, R., McGillen, M. R., Caravan, R. L., Taatjes, C. A., Orr-Ewing, A. J., Percival, C. J., and Shallcross, D. E. :Investigating the tropospheric chemistry of acetic acid using the global 3-D chemistry transport model, STOCHEM-CRI, J. Geophys., 123, 6267–6281, 2018.

Li, J., Mao, J., Fiore, A. M., Cohen, R. C., Crounse, J. D., Teng, A. P., Wennberg, P. O., Lee, B. H., Lopez-Hilfiker, F. D., Thornton, J. A., Peischl, J., Pollack, I. B., Ryerson, T. B., Veres, P., Roberts, J. M., Neuman, J. A., Nowak, J. B., Wolfe, G. M., Hanisco, T. F., Fried, A., Singh, H. B., Dibb, J., Paulot, F., and Horowitz, L. W.: Decadal changes in summertime reactive oxidized nitrogen and surface ozone over the Southeast United States, Atmos. Chem. Phys., 18, 2341–2361, 2018.

Mao, J., Carlton, A., Cohen, R. C., Brune, W. H., Brown, S. S., Wolfe, G. M., Jimenez, J. L., Pye, H. O. T., Lee Ng, N., Xu, L., McNeill, V. F., Tsigaridis, K., McDonald, B. C., Warneke, C., Guenther, A., Alvarado, M. J., de Gouw, J., Mickley, L. J., Leibensperger, E. M., Mathur, R., Nolte, C. G., Portmann, R. W., Unger, N., Tosca, M., and Horowitz, L. W.: Southeast Atmosphere Studies: Learning from model-observation syntheses, Atmos. Chem. Phys., 18, 2615–2651, 2018.

[Figure]

Millet, D. B., Baasandorj, M., Farmer, D. K., Thornton, J. A., Baumann, K., Brophy, P., Chaliyakunnel, S., de Gouw, J. A., Graus, M., Hu, L., Koss, A., Lee, B. H., Lopez-Hilfiker, F. D., Neuman, J. A., Paulot, F., Peischl, J., Pollack, I. B., Ryerson, T. B., Warnecke, C., Williams, B. J., and Xu, J.: A large and ubiquitous source of atmospheric formic acid, Atmos. Chem. Phys., 15, 6283–6304, 2015.

Paulot, F., Wunch, D., Crounse, J. D., Toon, G. C., Millet, D. B., DeCarlo, P. F., Vigouroux, C., Deutscher, N. M., González Abad, G., Notholt, J., Warneke, T., Hannigan, J. W., Warneke, C., de Gouw, J. A., Dunlea, E. J., De Mazière, M., Griffith, D. W. T., Bernath, P., Jimenez, J. L., and Wennberg, P. O.: Importance of secondary sources in the atmospheric budgets of formic and acetic acids, Atmos. Chem. Phys., 11, 1989–2013, 2011.

Stavrakou, T., Müller, J.-F., Peeters, J., Razavi, A., Clarisse, L., Clerbaux, C., Coheur, P.-F., Hurtmans, D., De Mazière, M., Vigouroux, C., Deutscher, N. M., Griffith, D. W. T., Jones, N., and Paton-Walsh, C.: Satellite evidence for a large source of formic acid from boreal and tropical forests, Nat. Geosci., 5, 26–30, 2012.
* * *

---

## Author Response (AR1)

Dear Editor,

Please find hereafter the replies to all reviewers' comments, the list of changes to the manuscript, and a marked-up manuscript version with all the changes. The reviewers comments were generally productive and stimulating, and the resulting revisions contributed to improve the manuscript. We trust that our replies will be considered satisfactory. Note that besides the changes suggested by the Reviewers, the chemical mechanism has undergone significant changes prompted by the results of a recent laboratory study, as described in the new subsection 2.1.2.

Note that the marked-up manuscript version was generated by latexdiff with the option –math-markup=whole. This option was indispensable in order to produce an output (there is a bug with latexdiff in the case of complicated equations). For this reason, the differences in the equations do not show up as they should. We apologize for the inconvenience.

I hope that you will find the present version of the paper suitable for publication in GMD.

Yours sincerely,

Jean-Francois Muller
jfm@aeronomie.be

**Reply to Anonymous Referee #1**

We thank the referee for their comments and respond to the points raised below.

*These concerns are detailed below in reference to the locations in the manuscript at which they appear, but briefly, they generally include the effects of specific (particularly the more poorly- understood) isoprene oxidation pathways on model outcomes along with the treatment of SOA and monoterpenes. The authors acknowledge that all of these aspects come with substantial uncertainty, but without any quantification of that uncertainty in the model, it is difficult for the reader to know how much proverbial stock to put into simula- tion results. It is perfectly reasonable that the model is not intended to provide detailed accounting of global biogenic SOA formation or the effects of terpene oxidation, but more discussion is needed regarding the limitations of these aspects of the model. For example, while some SOA formation pathways are included in the model (e.g. IEPOX reactive uptake), others are only mentioned as the likely sinks of isoprene oxidation products, without any physical reaction parameterized in the model (e.g. dinitrates and the hydroperoxy-epoxides from D'Ambro et al. (2017)). A modeler could assume that these products form SOA immediately with a 100% yield, but because some of the low-volatility species that lead to SOA formation would likely also have high deposition velocities, this could overestimate the SOA yield. How do these (and other) sources of uncertainty in the important results from this mechanism (e.g. HOx budgets, organic acid budgets, SOA) manifest themselves in a global model, and what should the reader take away regarding the potential for bias and error in simulating the overall effects of isoprene's oxidation on the chemistry of the global troposphere?*

The Reviewer is of course correct that the hydroperoxy-epoxides from D'Ambro et al. (2017) are not entirely converted to SOA. We now provide more discussion on SOA formation in the global modeling results Section (Sect. 4.2). We provide quantitative estimates of the main different pathways and their impact on the model results, as discussed further below (see response to Referee comment on DHHEPOX and SOA, p26 L14). In any case, we want to remind the Reviewer that the current focus of our mechanism is not on SOA formation.

On the issue of uncertainties: As discussed further below, uncertainties are plenty, and not confined to a single specific part of the mechanism. It is very difficult, and, to our view, out of scope of the present study, to go through every significant uncertainty in the mechanism and quantify their potential consequences. As required be the Reviewer, we provide some estimation of the potential impact of uncertainties related to the effect of SOA formation and wet/dry deposition on the formaldehyde production from monoterpene oxidation. Following the suggestion of Reviewer #2, we also present a box model comparison of the MAGRITTE, Caltech and MCM mechanisms, which will help readers to better evaluate in which conditions and for which species the mechanisms present important discrepancies.

*p4 L17: 5% can still be a lot of carbon for isoprene! Other pathways that also account for less than 5% of the total carbon from isoprene are included in this mech- anism. Are there other reasons the bimolecular reactions of the delta-hydroxyperoxy radicals are excluded? Do we have any knowledge of how much this simplification might bias the results of simulations using this model?*

We now include these pathways in the mechanism. We have added a new sub-section detailing this chemistry (Section 2.1.3 Traditional chemistry of the initial $\delta$-OH peroxy radicals). New model compounds are added: the $C_5$ hydroxycarbonyls, HALD1 and HALD2, and the $\delta$-hydroxynitrate ISOPANO3 (HOCH$_2$-C(CH$_3$)=CH-CH$_2$ONO$_2$), not lumped anymore with ISOPCNO3 (HOCH$_2$-CH=C(CH$_3$)-CH$_2$ONO$_2$).

*L25 & the rest of this section: the discussion of remaining uncertainties in this mechanism pathway is welcome, but given the important effects of this pathway on*

*simulation results (e.g. as a source of HOx radicals in the otherwise HOx-consuming mechanism), it would be useful to provide the reader with some discussion (perhaps in the model results section) of how these uncertainties manifest themselves in the model. What range of possible HOx recycling rates would be compatible with what is currently known about this part of the mechanism? Given the uncertainties, can the boundary layer OH change due to this mechanistic pathway (Figure 3) be considered a bound or a best guess, and is the uncertainty on that at all quantifiable?*

This part of the chemical mechanism has undergone major changes. As now discussed in great detail in a new subsection (Section 2.1.2), the quantitative product distribution from the 1,6 H-shift of the $Z$-$\delta$-OH-peroxys is adopted from the recent experimental study of Berndt et al. (2019), supported and complemented by computational results of the LIM1 paper (Peeters et al., 2014). A crucial point is that, contrary to speculative suggestion in the LIM1 paper, the $Z$–$E$ isomerism of the transition states is conserved in the allylic-radical products and in the resulting peroxys. The implications are detailed in the new Section 2.1.2. Both theoretical expectation and experimental results imply a high HPALD yield (ca. 75%), whereas hydroperoxy carbonyl epoxides (HPCE, 15%) and the dihydroperoxycarbonyl peroxys (DIHP-CARPs, 10%) make up the rest. The further chemistry of HPCE and DIHPCARPs is also discussed in this Section. The MAGRITTE mechanism (v1.1) has been revised to accomodate these important changes, including new model compounds (HPCE as well as several compounds resulting from the further chemistry of HPCE and DIHP-CARPs). Note that the high HPALD yield is also comforted by the model evaluation against SEAC[4]RS measurements at the CIMS mass corresponding to HPALD (see our response to the last Reviewer comment).

In consequence, the product distribution of the 1,6 H-shift of the $Z$-$\delta$-OH-peroxys is probably not the most important source of uncertainty in the overall mechanism. Uncertainties remain important, but are not confined to this part of the mechanism. It is therefore very difficult to go through every significant source of uncertainty in the mechanism and quantify its potential consequences. Following the suggestion of Reviewer #2, we now present a box model comparison of the MAGRITTE, Caltech and MCM mechanisms, which will help readers to better evaluate in which conditions and for which species the mechanisms present important discrepancies. For example, significant differences are found for OH at low-NOx, with Caltech predicting higher concentrations by about 30% higher than the other mechanisms. For the most part, the differences can be traced back to assumptions regarding the isomerization of $Z$-$\delta$-OH-peroxys and the photolysis of hydroperoxycarbonyls. Although the MAGRITTE results should be viewed as state-of-the-art, more work is needed to confirm and refine the assumptions made in our study. An extensive review of the uncertainties and their potential consequences is out of scope of the present study.

*p9 L15: Fig 1 doesn't show latitude dependence as claimed here. It also convolutes the pressure and temperature dependences in a way that might not be useful for readers who would like to extrapolate for conditions other than 40 degrees N in January and July. I would suggest either clarifying some of the details of these conditions (e.g. add side plots of temp and pressure vs. altitude in January and July, or separate this into plots of yield vs. temp and yield vs. pressure).*

The yields are now shown as functions of pressure instead of altitude, and the plot now includes a side plot of temperature vs. pressure.

*p9 L19: "such" should be "this". Also, Wennberg et al. (2018) does not show that this procedure inherently overestimates most measured nitrate yields, though it does suggest that this equation provides yields that seem excessive for dinitrates. Instead it goes off the recommendation of Teng et al. (2017), which explicitly says that this provides a better estimate than just n=#C, and improves this with a structure-activity-relationship-style modification.*

We now use the parameterization of Wennberg et al. (2018) for the calculation of

RONO$_2$ yields. We have updated Section 2.6 (Peroxy radical reactions with NO and HO$_2$) accordingly.

*p26 L3: what is "isoprene-OH segregation"? What was the logic behind the 7% minor addition channel, and behind including those but not the E/Z-delta bimolecular products?*

As explained above, we have now included the $E/Z$-$\delta$-OH-peroxy bimolecular reactions and products. The "isoprene-OH segregation" effect results from incomplete mixing within a model grid cell and unresolved anti-correlation between isoprene and OH (due to their mututal reaction). The 10% reduction estimate is consistent with the results of Pugh et al. (2011) based on measurements in a tropical forest in Borneo.

*p26 L10: if you include the description of Y(Arey) in the table heading, you don't need it in the footnote; I think the table heading should be shorter and this could be a footnote. Also, don't all the scaling factors >1 suggest that N=#heavy atoms wouldve been better than N=#C?*

As suggested by the Reviewer, we shortened the table heading. As noted above, we now use the organic nitrate parameterization by Wennberg et al. (2018).

*p26 L11: define "room conditions" (this also comes up on p27 L4 & 13, p31 L15, 18 & 25, p32 L20, and p41 L34.*

Done as requested.

*p26 L14: Does this inherently assume that all DHHEPOX is lost to aerosols? Are there any estimates of the OH reaction coefficient or uptake coefficient of DHHEPOX that might put this assumption in context, or provide the reader with some idea as to the uncertainty on this assumption? What fraction of carbon is lost to this pathway under atmospheric conditions (and to other dead-end pathways assumed to either deposit or partition to aerosols, e.g. the dinitrates discussed in N9), and what is the resulting contribution to aerosol compared to other pathways (e.g. IEPOX)?*

The Reviewer is correct that DHHEPOX is not entirely lost to aerosols. We have updated the text (Note N6) as follows: "The further chemistry of the dihydroxy hydroperoxy epoxide resulting from this isomerisation, DHHEPOX, is not considered. Its saturation vapour pressure is estimated to be of the order of $3 \cdot 10^{-9}$ atm at 298 K using a group contribution method (Compernolle et al., 2011), i.e. three orders of magnitude lower than the estimated vapour pressure of $\beta$-IEPOX ($3 \cdot 10^{-6}$ atm). The Henry's law constant (HLC) of DHHEPOX estimated as described in Muller et al. (2018) is equal to $\sim 3 \cdot 10^9$ M atm$^{-1}$ at 298 K, almost three orders above the estimated value for IEPOX. DHHEPOX is therefore very probably more soluble and prone to loss by deposition or SOA formation than IEPOX, which has been shown to deposit very rapidly on vegetation (Nguyen et al., 2015b) and to be a prominent SOA precursor (Surratt et al., 2010). Furthermore, the products of the oxidation of DHHEPOX by OH (at a rate estimated at $\sim 2.1 \cdot 10^{-11}$ molec.$^{-1}$ cm$^3$ $s^{-1}$) are also expected to consist, for the most part, of highly oxygenated products prone to deposition and heterogeneous uptake. "

We thank the Reviewer for the interesting question on SOA formation. We inserted a new paragraph at the end of Section 4.2: "Although SOA is not a focus of this study, SOA formation processes are included in the model. The largest source of SOA is the uptake of IEPOX, with a global flux (49 Tg or 25 TgC yr$^{-1}$) of magnitude similar to previous model estimates, of the order of 40 Tg yr$^{-1}$ (Lin et al., 2012; Stadtler et al., 2018). These estimates are very uncertain, since the reactive uptake parameterization used in models ignores the complexity of SOA formation which involves the partitioning of semi-volatile compounds and chemical transformations in the gaseous and particulate phases (D'Ambro et al., 2019). Glyoxal is another well-identified source of SOA, amounting to 10 Tg yr$^{-1}$ globally (4.3 TgC yr$^{-1}$), also well in the range of previous estimations (6-14 Tg yr$^{-1}$) (Fu et al., 2008; Stavrakou et al., 2009b; Lin et al., 2012). The dihydroxy dihydroperoxides (ISOP(OOH)$_2$) formed from

the oxidation of ISOPOOH by OH were recently estimated to be a dominant source of SOA (Stadtler et al., 2018); in our mechanism, these compounds are ignored since their yields are believed to be negligible in atmospheric conditions (D'Ambro et al., 2017). The major non-IEPOX products of OH-addition to ISOPOOH are dihydroxy hydroperoxy epoxides (DHHEPOX), also believed to form SOA as discussed above (Note N6). Their global production in the model amounts to 30 Tg yr$^{-1}$ (12 TgC yr$^{-1}$). Assuming that their reactive uptake is as effective as for IEPOX, and neglecting gas-phase oxidation by OH (which generates other low-volatility compounds also expected to form SOA), we estimate with the model that SOA formation accounts for two-thirds of the sink of DHHEPOX (i.e. 20 Tg yr$^{-1}$), whereas dry/wet deposition makes up the rest. If confirmed, this would make DHHEPOX the second-largest contribution to isoprene SOA.

Other SOA formation pathways are implied, but not explicitly represented by the MAGRITTE mechanism, such as the hydrolysis of dihydroxy dinitrates (Note N12) and dihydroxy hydroperoxy nitrates (Note N13). The hydrolysis products, nitroxy- and hydroperoxy-triols are expected to be of very low volatility and remain mostly in the aerosol phase, as their vapour pressures (Compernolle et al., 2011) are estimated to be very low. Those triols represent only a minor contribution to the global SOA budget, however, as their estimated global production is ~3 Tg yr$^{-1}$ (1.2 TgC yr$^{-1}$). "

*p26 L22: despite this being "well known" I think it deserves a citation, and more support than that the "majority" of exothermicity is alternately directed, making it "appear unlikely". It is difficult from reading this footnote to tell what is conjecture and what has experimental evidence to support the pathways used. The language generally implies certainty, but the lack of citations suggests that it is conjecture.*

We have modified this part of the Note as follows: "Abstraction of hydroperoxide-H (75%) and of hydroxy-$\alpha$-H (25%) (Wennberg et al., 2018). The latter leads to a radical proposed to undergo epoxide formation (Wennberg et al., 2018); we neglect this very minor and uncertain pathway as the product was suggested to be due to an impurity (St. Clair et al., 2016). "

*p32 L1: Here you state that H-abstraction from the carbon dominates, but with a higher yield of HCHO than HCOOH, the former of which is derived from H-abstraction from the hydroperoxide, isn't that backwards? Also, within their reported uncertainty, Allen et al. (2018) did not conclusively state that one path dominates over the other.*

We thank the Reviewer for spotting this mistake. The text has been changed as follows: "H-abstraction from hydroperoxide group, followed by decomposition of the hydroxymethylperoxy radical, is slightly dominant (Allen et al., 2018). H-abstraction from the carbon is followed by OH expulsion."

*p32 L3: "he" should be "the"*

Corrected.

*p32 L5: the discussion here seems more suited for a subsection of section 2 than for a note at the bottom of a table; the generic monoterpene oxidation scheme provided here needs more discussion of its uncertainties and how the specific numbers were arrived at. While the complexity of terpene oxidation and the relative lack of quantitative knowledge about its oxidation mechanism make drastic simplification a necessity, it is not clear to the reader why this particular set of simplifications is ideal, or what the reasonable uncertainty bounds are on any of these rates and product yields. If pinene (or generic terpene) oxidation is to be included in this model, it should be given more than a paragraph in a footnote. The same might be said for MBO, but the relative simplicity of its oxidation mechanism, the overlap with isoprene oxidation products, and the smaller magnitude of its emissions make it less prone to substantial uncertainty and bias from mechanistic simplifications and shortcuts. Maybe some sensitivity studies showing the range of results you could get in a global model given*

*the uncertainties in this mechanism would be most useful? (e.g. assuming more of less of the products are lost to SOA/deposition, or assuming a range of nitrate and/or HOx yields). Some specific questions include: on line 10, where does the 45% number come from? How do these simulated acetone and formaldehyde yields compare to previous work? What does this mechanism inherently assume for the SOA formation from pinene, and how does that compare to both measured yields and the magnitude of SOA formed from isoprene globally? What might be the implications in a global simulation of skipping oxidative steps (and therefore likely sinks of OH, HO2, NO) in the oxidation mechanism, as is presumably the case when only one generation of products are used?*

We moved this discussion to a new subsection (Section 2.4). As noted by the Reviewer, drastic simplification of the monoterpene mechanism is a necessity. We now better emphasize the limited scope of our simple mechanism, which is the reproduction of the final yields of a few key products. The subsection text is as follows:

"Due to the complexity and poor understanding of monoterpene oxidation, we adopt a simple parameterization based on box model simulations of $\alpha$- and $\beta$-pinene oxidation using the MCMv3.2 (Saunders et al., 2003). The scope of the parameterization is limited to the reproduction of total yields of several key products; those yields reflect not only primary production but also secondary formation. The influence of monoterpenes on radicals (e.g. $HO_x$, $RO_2$) and on ozone production is therefore likely not well represented by this simple mechanism. It should be stressed that even the monoterpene mechanism in MCM is greatly oversimplified, as it neglects many possibly important pathways (in particular H-shift isomerisations in peroxy radicals), with potentially very large effects on radicals and other products. A thorough evaluation of mechanisms against laboratory data will be needed in order to assess their uncertainties, but is out of scope of the present study.

The parameterization relies on sixty-day simulations performed using the Kinetic PreProcessor (KPP) package (Damian et al., 2002). The photolysis rates are calculated for clear-sky conditions at 30°N on July 15th. Although both high-NOx (1 ppbv $NO_x$, 40 ppbv $O_3$ and 250 ppbv CO maintained throughout the simulation) and low-NOx simulations (100 pptv NOx, 20 ppbv $O_3$ and 150 ppbv CO) are conducted, only the low-NOx results are used for the parameterization. Temperature and $H_2O$ are kept at 298 K and 1% v/v. To determine the product yields, counter compounds are introduced in the equation file (e.g. HCHOa, MGLYOXa, etc.) having the same production terms as the species they represent, but without any chemical loss.

The yield of acetone from both $\alpha$- and $\beta-$pinene is very close to 100% after several days of reaction, independent of the NOx level. The yield of methylglyoxal is low (4% and 5% for $\alpha$- and $\beta$-pinene, not counting the contribution of acetone oxidation by OH). The overall yield of formaldehyde obtained in these simulations is $\sim$4.2 HCHO per monoterpene oxidized, almost independent of $NO_x$, for both precursors. The HCHO yield comes down to 2.3 after subtracting the contributions of acetone and methylglyoxal oxidation. This yield is further reduced by 45% to account for wet/dry deposition of intermediates and secondary organic aerosol formation. That fraction is higher, but of the same order, as the estimated overall impact of deposition on the average formaldehyde yield from isoprene oxidation ($\sim$30%), based on global model (MAGRITTE) calculations. The higher fraction is justified by the larger number of oxidation steps and the generally lower volatility of intermediates involved in formaldehyde formation from monoterpene oxidation. Nevertheless, this adjustment introduces a significant uncertainty in the model results. A sensitivity calculation shows that adopting a lower yield reduction (20% instead of 45%) in the global model (Sect. 4.1) has negligible impact on the calculated HCHO abundances ($<\sim$1%) in most regions, but leads to higher HCHO vertical columns in monoterpene emission regions, by $\sim$5% over Amazonia and by up to 8% over Siberia. The associated impact on OH reaches +2% in those regions, due to the additional $HO_x$ formation through HCHO photolysis.

The overall carbon balance of monoterpene oxidation in the mechanism is $\sim$50%

due to the combined effects of deposition, SOA formation and CO and $CO_2$ formation besides their production through the degradation of the explicit products. "

To our understanding, the assessment of uncertainties requested by the Reviewer is currently out of reach, in absence of any reliable reference mechanism validated by laboratory data. Note that the mechanism does not make specific assumption regarding SOA formation, besides the fact that it is expected to remove HCHO precursors from the gas-phase, and therefore reduce the overall HCHO yield.

*p37 L14: How is the oxidative degradation of anthropogenic NMVOCs treated in the model? I am particularly concerned about the possible contribution of degradation of non-isoprene compounds to the gas-phase budgets of glyoxal and the organic acids. Along those same lines, could some discussion of the potential for additional sources of the gas-phase organic acids not included in this model (e.g. degradation of other compounds, revolatilization from SOA) be added to those sections?*

The chemical oxidation mechanism of anthropogenic and biomass burning VOCs has been described in previous publications with the IMAGES model, as now mentioned in the model description (Section 4.1). The yields of glyoxal in the oxidation of aromatic compounds and acetylene are now provided in Section 4.5: "The glyoxal yields in their reactions with OH (0.74, 0.7, 0.36 and 0.636 for benzene, toluene, xylenes and acetylene, respectively) are obtained from the MCM (Saunders et al., 2003; Bloss et al., 2005). Regarding aromatics, this yield includes not only primary formation but also later-generation production (Chan Miller et al., 2016)." Since the topic of our study is the oxidation mechanism of biogenic VOCs, we don't believe necessary to lengthen the paper with more details on the chemistry of other VOCs.

We now include a short discussion of potential additional sources of formic and acetic acids at the end of Section 4.4: "Additional sources are likely at play, such as enol formation through other pathways than those considered here (e.g. in monoterpene and anthropogenic VOC oxidation, e.g. through the photolysis of aldehydes (Tadic et al., 2001a; Tadic et al., 2001b)) and the photodegradation of organic aerosols (Paulot et al., 2011; Malecha and Nizkodorov, 2016)."

*p37 L18: Are there primary emissions of MBO in the model? What is the effect of the MBO oxidation mechanism (and the terpene mechanism) in the model?*

The global biogenic emissions of MBO amount to 0.93 TgC $yr^{-1}$. This is now mentioned in the model description section. The effect of MBO is small, due to its low emissions. Its oxidation is a source of acetone ($\sim$0.5 TgC $yr^{-1}$).

Monoterpenes have multiple, but very uncertain effects. The comparison of model simulations performed with and without monoterpene emissions indicates significant increases of several compounds due to monoterpenes, e.g. glyoxal (global burden +22%), acetone (+40%), acetic acid (+9%), formic acid (+12%) and formaldehyde (+3%). Much larger impacts are calculated over emission regions, e.g. HCHO vertical columns are increased by up to 15-20% over boreal forests and Amazonia. As monoterpene chemistry is by far the most uncertain part of the BVOC mechanism, we prefer not to discuss these impacts in the article, which is already very long.

*p40 L5: In the comparisons to SEAC4RS data, it would be helpful to list the measurement uncertainties and spreads alongside the over/underestimations of the model. Also, what exactly is the model output being compared with the measurements in this section? Are the simulated average profiles just the average over the entire SE USA between 0900 and 1700 hours, or are they points subsampled from the model concurrent with the flight paths? If it's the former, which assumes that the SEAC4RS observations (masked for plumes and stratospheric intrusions) are representative of the averaged regions, how might this skew the comparison between the model and measurements?*

The model profiles are averages based on values interpolated at each measurement location and time. This is now mentioned in the manuscript. Wherever relevant, measurement uncertainties and model over/underestimations are reported in the text. We

don't believe especially helpful to make the paper longer with a new Table providing comparison statistics and measurement uncertainties.

*p41 L5: Are these non-HPALD compounds also isoprene products? Do we have any indication as to what they are? If they have the same mass as the HPALDs, are there other species in your mechanism that also have this mass that may account for this mass?*

Yes, the isoprene carbonyl hydroxy epoxides (ICHE) formed mainly from the oxidation of IEPOX by OH have the same formula ($C_5H_8O_3$) as HPALD. We now present a model comparison with the SEAC4RS measurement for that mass (Fig. 9). The following text accompanies this comparison: "The model-calculated HPALD concentrations (dotted line on the $C_5H_8O_3$ panel of Fig. 9) are on average about a factor of two lower than the observed Caltech CIMS (Chemical Ionisation Mass Spectrometry) signal at the corresponding mass; when adding the contribution of the carbonyl hydroxyepoxides (ICHE), which have the same formula ($C_5H_8O_3$) as HPALD and can be expected to interfere with HPALD measurements, the model falls within the measurement uncertainty (50%) with an underestimation decreased to -34% (solid line on Fig. 9). The ICHE compounds are formed from the oxidation of IEPOX (as well as HPALDs) by OH. It is likely than other, unknown compounds contribute to the CIMS signal at the same mass, as also observed in the PROPHET campaign in Michigan, where the HPALD contribution to the CIMS measurement at the given mass was estimated at 38% based on the relative contribution of the HPALD peaks to the total GC area (Vasquez et al., 2018). This is consistent with our modelled HPALD accounting for 50% of the CIMS measurement, when considering also that all isoprene oxidation products appear slightly overestimated by the model as suggested by the ~20% overprediction of modelled ISOPOOH and MVK+MACR relative to the measurements. In spite of the important uncertainties and remaining unknowns (e.g. the identity of additional compounds contributing to the CIMS signal), this good consistency provides strong support to the high HPALD yield (75%) adopted in this work in the isomerisation of $Z$-$\delta$-OH-peroxys from isoprene (Sect. 2.1.2). Lower yield values as proposed in recent previous work, i.e. 50% (Peeters et al., 2014; Jenkin et al., 2015) or 25% (Teng et al., 2017; Wennberg et al., 2018) would lead to much stronger HPALD underestimations against SEAC$^4$RS data. "
* * *
**Reply to Anonymous Referee #2**

We thank the referee for their comments and respond to the points raised below.

*The authors integrated all the major advancements and originally contributed to large portion of them. Their critical understanding of the relevant chemical processes, far from being all achieved, adds to the value of manuscript. Impact of recent experimental and theoretical advancements of the global budgets of organic acids is very interesting. The model is fairly well described and the evaluation seems appropriate for use in global models. However, a box model comparison between MAGRITTEv1.0 and MCMv3.3.1, the mechanism presented by Wennberg et al. 2018 or even their detailed mechanism would add useful information about the model performance. I wish the authors could provide such data and information.*

We thank the Reviewer for the suggestion. We added a new section "Box model comparison with other isoprene mechanisms". We intercompare the MAGRITTE mechanism version 1.1, MCMv3.3.1 and the reduced Caltech mechanism. We don't believe useful to include the "full" version of the Caltech mechanism in this comparison, as it does not treat the further degradation of numerous oxidation products. We perform 30-hour simulations using KPP, starting at 9 AM with 2 ppbv isoprene. NOx is fixed at either 1 or 0.1 ppbv. The photolysis rates are calculated for mid-July clearsky conditions at 30°N, using the TUV model of Madronich (1993). For computational efficiency, the photorates are parameterized as a function of solar zenith angle using MCM-type expressions (Saunders et al., 2003). All rate coefficient expressions are available at the MAGRITTE repository (http://doi.org/10.18758/71021042). Since Wennberg et al. does not provide detailed recommendations for the calculation of photolysis rates, we use our own expressions in their mechanism.

Note that the new version (v1.1) of the MAGRITTE mechanism differs from the initial version (v1.0) described in the GMDD paper. The most important updates include

(1) updated product distribution of the 1,6-H-shift isomerisation of the $Z$-$\delta$-OH-peroxys from ISOP+OH, including a higher HPALD yield (0.75 instead of 0.25), in agreement with recent laboratory data (Berndt et al., 2019) and with theoretical calculations, as described in detail in the revised version of the manuscript (see also our Reply to Reviewer #1),

(2) inclusion of the bimolecular reactions of the $Z$-$\delta$-OH-peroxys from ISOP+OH, following a comment of Reviewer #1,

(3) calculation of RONO2 yields in RO2+NO reactions following Wennberg et al. (2018).

The comparisons show that MAGRITTEv1.1 leads to lower HOx recycling than the Caltech mechanism. Sensitivity calculations show that the difference is primarily due to (i) differences in the $Z$-$\delta$-OH-peroxy isomerisation rates and products, and (ii) differences in the product distribution of hydroperoxycarbonyl (especially HPACET and HPAC) photolysis. Important differences between the mechanisms are also found for e.g. carboxylic acids, PANS, nitrates and methanol, as discussed in the revised version of the manuscript.

*1) Bulk isomerization rates*
*Please explain more the counter-intuitive concept by which the bulk isomerization rate of the lumped (beta- and delta-) species ISOPBO2 and ISOPDO2 should linearly increase with the traditional RO2 sink rate (kp). Why is it not or it has to be different than what Crounse et al. (2011) reported? Even if correct, neglecting the RO2 sink due to permutation reactions should yield non-negligible errors/deviations from the analytical solution. Please explain why the neglect and in case provide an estimate of the deviation caused by it.*

We provide now a better justification of the bulk isomerisation rate expressions:

"Based on a detailed steady-state analysis, the bulk isomerisation rate of ISOPBO2 and ISOPDO2 was shown to increase linearly with the sink rate ($k_p$) of the traditional peroxy reaction (Peeters et al., 2014). The reason for this behaviour is that at low $k_p$, the ratio of the $Z$-$\delta$-OH-peroxys over the lower-energy $\beta$-OH-peroxys is close to their equilibrium ratio, of order of only ∼0.01, whereas at the high $k_p$ limit, where all peroxys have a similar lifetime, their ratio is governed by their initial formation branching ratio, which is an order magnitude higher (Peeters et al., 2014; Teng et al., 2017)." Note that the linear dependence of bulk isomerisation rates on $k_p$ was verified experimentally by Teng et al. (2017).

Neglecting the $RO_2$ sink due to permutation reactions in those bulk isomerisation rate expressions has a negligible impact, estimated at ∼0.6% of the bulk isomerisation rate for ISOPDO2, and even less for ISOPBO2.

*2) Reproducibility of results*
*The chemical mechanism of MAGRITTEv1.0 is not exactly what can be downloaded at the link given. A few sample differences are listed below. The reaction of CH3OH with OH is standard in the manuscript but in MAGRITTE.eqn file one finds two*

*reactions with one including the water vapor catalysis by Jara-Toro et al. 2017.*

We thank the Reviewer for pointing this out. Updated equation and species files are now available at the MAGRITTE repository. The water vapor catalysis proposed by Jara-Toro et al. is not included, as it was recently disproved by a recent laboratory study (Chao et al., 2019).

*The rate constant for the reaction*
*CH3O2 + HO2 = 0.9 CH3OOH + 0.1 HCHO*
*is 4.1E-13\*exp(750/TEMP) and 3.8E-13\*exp(780/TEMP), respectively.*

Corrected.

*Concerning the 1,6-H-shift of ISOPDO2 in the .eqn file one finds*
*ISOPDO2 = 0.25 HO2 + 0.25 HPALD2 + 0.75 OH + 0.75 CO + 0.75 DIHPCHO*
*: 4.253E8\*exp(-7254/TEMP) ;*
*ISOPDO2 + NO = NO + 0.25 HO2 + 0.25 HPALD2 + 0.75 OH + 0.75 CO + 0.75*
*DIHPCHO : 6.29E-19\*exp(4012/TEMP) ;*
*ISOPDO2 + HO2 = HO2 + 0.25 HO2 + 0.25 HPALD2 + 0.75 OH + 0.75 CO +*
*0.75 DIHPCHO : 4.90E-20\*exp(4962/TEMP) ;*
*The last two reactions constants are not the ones reported in Table 2.*

Corrected.

*PYRA (pyruvic acid) is listed in Table 1. However, it is neither in Table 2 nor in the .eqn file.*

Corrected (PYRA is not a model species).

*Overall, it might be that the authors uploaded another version of MAGRITTE. Please upload a v1.0 that is faithful to the Tables in the manuscripts. The files should bear the information about the exact model version.*

The new version of the mechanism (v1.1) supersedes version v1.0. The files now bears the information about the model version.

*Moreover, no file with the actual functions used for many rate constants is given. This is also the case for the cross-sections and quantum yields used for computing the photolysis frequencies. Please also provide this information.*

We thank the Reviewer for the excellent suggestion. We now provide the functions used for calculations of rate constants (including photolysis rates as discussed above) in the MAGRITTE repository, as well as data files with the absorption cross-sections of polyfunctional carbonyls not found in current recommendations (IUPAC, JPL). The photolysis parameters of other compounds are readily available from e.g. those recommendations.
* * *
**Reply to Anonymous Referee #3**

We thank the referee for their comments and respond to the points raised below.

*1. The authors use SEAC4RS dataset for their model evaluation, and compare their results to Fisher et al. (2016) extensively for RONO2 budgets and speciation. It should be pointed out that this paper uses a RONO2 yield of 13%-14% from isoprene RO2+NO reaction, in contrast to 9% assumed in Fisher et al. (2016). Such difference would presumably lead to significant differences between these two models. I believe some caveats should be provided in the text to make reader aware of these differences.*

This is correct. This difference in RONO2 yield between the two studies is now mentioned in the discussion of the NOx loss through RONO2 formation (Sect. 4.2) and

again in the evaluation of total RONO2 against SEAC⁴RS measurements (Sect. 4.3).

*2. Similar to Fisher et al. (2016), the authors find a model underestimate of RONO2, as shown in their Figure 5. A recent study by Li et al. (2018), suggests that a large part of discrepancy could be due to terpene nitrates and nighttime isoprene nitrates. In particular, the authors assume a 100% recycling of NOx from APINONO2 + OH. This choice may have a large impact on total RONO2. For nighttime chemistry, the authors have ignored the formation of dinitrate (N31 for Table 2), which could also contribute to RONO2, according to Li et al. (2018). Some discussion on the uncertainties of terpene nitrates and nighttime isoprene nitrates, should be included in the text.*

Thank you for these valid points. We include now a more complete discussion of the possible causes of RONO2 underestimation in the model: "There are several possible explanations for the discrepancy, including the neglected reactions of $NO_3$ with unsaturated oxidation products from isoprene and other BVOCs, the neglected formation of unsaturated dinitrates from the reaction of dinitroxyperoxy radicals (NISOPO2) with NO (Li et al., 2018), a possible overestimate of the tertiary nitrate hydrolysis sink (for dinitrates from ISOP+OH), and a misrepresentation of alkyl and hydroxyalkyl nitrates from other precursors than isoprene. The monoterpene nitrates are very crudely represented in the model. In particular, the assumption of 100% NOx recycling in their reaction with OH could lead to a significant overestimation of RONO2 loss. "

*3. The reader is also wondering how this model performs on HNO3 and PAN, which are major NOy reservoirs. Examining these species may help to justify the 60% reduction of U.S. NOx emission inventories in their model.*

We now include PAN in the model comparison with SEAC⁴RS (Fig. 9). A moderate model overestimation is found, similar to previous studies (Travis et al. 2016, Li et al. 2018). The model also overestimates $HNO_3$ measurements from SEAC⁴RS, but reproduces well the average $NO_3^-$ wet deposition measurements over the U.S. (data obtained from R. Larson, NADP Database Manager, Wisconsin State Laboratory of Hygiene). A detailed discussion and justification of the NOx emission reduction (similar to a previous model study, Travis et al. 2016) is clearly beyond the scope of our study.

*4. It seems that Section 3.4, Global budget of formic and acetic acid, is disconnected from the rest of the paper. It appears that the authors want to recalculate the global budget of these two acids, without any comparison to field observations. It is unclear how this new mechanism has improved current knowledge on formic and acetic acid. Some model sensitivity tests and comparison to observations would be useful.*

As reported in the text, there is wide consensus that models underestimate formic and acetic acid abundances by large factors. This has been shown through numerous model comparisons with aircraft, ground-based and in situ measurements. Our study does not claim to reconcile models with observations. Since our newly-derived global sources of those acids are similar as (or even lower than) in previous modelling studies, it is clear that the large model underestimations remain, and extensive comparisons with atmospheric observations are not needed to make that point. We added the following sentence to Section 4.4: "Despite the newly-proposed large production of formic and acetic through hydroperoxycarbonyl photolysis, our derived total sources of those acids remains similar as (or even lower than) in previous modelling studies (Paulot et al., 2011; Stavrakou et al., 2012; Millet et al., 2015; Khan et al., 2018), and are therefore insufficient to explain their high observed abundances."

*5. While reading Section 3.4, the authors suggest CH3CO3+HO2 is the major source of CH3COOH. This seems like another good reason to examine PAN in their model.*

PAN is now included in the model comparison against SEAC⁴RS observations.

*6. Given the extensive research on isoprene oxidation over Southeast US, the authors should include two review papers on this topic in their introduction, Carlton et al. (2018) and Mao et al. (2018).*

Done.
* * *
**LIST OF CHANGES TO MANUSCRIPT**

The changes to the manuscript are listed below, along with their justifications.

- The version of the mechanism becomes 1.1 instead of 1.0

- Due to changes in the mechanism, the global model results have changed, and so have the figures and numbers in the abstract, global results section, and Conclusions

- We added a sentence in the abstract on the impact of OH recycling in isoprene oxidation

- As response to a comment by Rev.#2, the linear increase of bulk isomerisation rates is discussed and justified (P 4)

- As response to a comment of Rev.#1, we now include the chemistry following the bimolecular reactions of the Z-$\delta$-OH-peroxys. The product yields from those reactions are illustrated in a new Figure (Fig. 1) and compared with their parameterization in MAGRITTE. The traditional chemistry of those radicals is described in a new subsection (2.1.3).

- The chemistry following the isomerisation of the Z-$\delta$-hydroxyperoxy radicals from isoprene has undergone important changes, as detailed in our response to Rev.#1. This chemistry is the topic of a new subsection (2.1.2).

- As response to a comment by Rev.#1, the treatment of monoterpene chemistry is now the topic of a separate subsection (2.4) which better explains the procedure adopted for deriving the parameterization. Sensitivity calculations were performed to assess the potential impact of uncertainties associated with SOA formation and wet/dry deposition of intermediates.

- The rates of the cross reactions of peroxy radicals from ISOP+OH has been revised, based on the recommendations of Wennberg et al. (2018)

- As response to a comment by Rev.#1, the parameterization of the RONO2 yield in RO2+NO reactions has been revised, based on the recommendation of Wennberg et al. (2018). The Figure showing the yields has been adapted and modified based on a comment by Rev.#1. Note N3 was also adapted to reflect the changed parameterization

- As response to a comment by Rev.#1, the fate of DHHEPOX and the respective roles of SOA formation and deposition are discussed (Note N6)

- As response to a comment by Rev.#1, we modified the justification for neglecting epoxide formation hydroxy-alpha-H abstraction from ISOPBOOH

- The 1,5 and 1,6 H-shift in peroxy radicals from OH-addition to the isoprene hydroxynitrates are now included (Notes N12 and N14)

- The chemistry of $\delta$-OH-nitrates from isoprene has been updated to account for their formation from $\delta$-OH-peroxy reactions with NO (as response to a comment by Rev.#1) (Notes N43-44, N48-49)

- New compounds (HALD1, HALD2) were added, resulting from $\delta$-OH-peroxy reactions with NO (response to comment by Rev.#1) (Notes 51-52)

- The oxidation of HMML was revised (Note N53)

- As response to a comment by Rev.#1, a mistake in the explanatory Note on HMHP+OH was corrected (Note N74)

- As response to a comment by Rev.#1, the content of explanatory Note on monoterpene oxidation was transferred to a subsection (2.4)

- The photolysis rate calculation is better described in Section 2.10. As response to a comment by Rev.#2, absorption cross section data are provided in the MAGRITTE model repository (http://doi.org/10.18758/71021042) and a new Figure illustrating those cross sections is added (Fig. 3). In addition, a new column has been added to Table 3 with calculated photorates for all photodissociations.

- The quantum yields of glyoxal photolysis were changed (from Salter et al. to the JPL recommendation)

- New photolysis reactions were added (for ICHE, HPCE, PGA)

- The channel ratios of BIACETOH photolysis were modified as justified in Note t

- The reactive uptake coefficient of tertiary nitrates on aerosol has been reduced (0.03 for ISOPBNO3)

- As response to a comment by Rev.#1, the fate of minor tertiary nitrates assumed to undergo hydrolysis is discussed, as well as the possible role of uncertainties in the assumed hydrolysis rate (page 48)

- In response to a comment by Rev.#2, a box model comparison of several isoprene oxidation mechanisms is presented (Section 3)

- The discussion of the global model results has been updated to reflect the changed mechanism and impact (Section 4.2)

- As response to a comment by Rev.#3, the role of the RONO2 yield differences between Fisher et al. and our study is mentioned in the text (P. 54 and P. 56)

- As response to a comment by Rev.#1, quantitative estimation of different SOA formation pathways are provided (end of Sect. 4.2)

- As response to a comment by Rev.#1, the HPALD CIMS measurement issue is discussed (Sect. 4.3 page 55)

- As response to a comment by Rev.#3, additional possible causes for the RONO2 underestimation are discussed (Sect. 4.3, page 56)

- As response to a comment by Rev.#1, potential missing sources of formic and acetic acid are mentioned (Sect. 4.4, page 61)

- As response to a comment by Rev.#1, glyoxal yields from anthropogenic yields are reported (Sect. 4.5, page 61)

- The Conclusions are adapted to reflect the changes in the isoprene mechanism

The differences between the GMDD version and the revised version of the manuscript are highlighted below. Note that the standard latexdiff command failed to generate any output. We had to use the option –math-markup=whole to avoid the bug. For this reason, the differences in the equations do not show up. We apologize for the inconvenience.

[revised manuscript text omitted]

 +0.9 CH$_3$O$_2$ + 0.9 CO$_2$ + 0.1 CH$_3$COOH + 0.1 HCOC5 | $2.0(-12)\exp(500/T)$ | 6,7 |  |
| ISOPDO2 → 0.75 HPALD2 + 0.75 HO$_2$ + 0.15 HPCE
 +0.15 OH + 0.1 DHPAO2 | $4.253(+8)\exp(-7254/T)$
 $+6.29(-19)\exp(4012/T)\cdot[\text{NO}]$
 $+4.9(-20)\exp(4962/T)\cdot[\text{HO}_2]$ |  | N4 |
| ISOPDO2 → MACR + HCHO + OH | $1.77(+11)\exp(-9752/T)$ | 8 |  |
| HPCE + OH → 1.82 CO + 0.82 OH + 0.82 HPACET + 0.18 KPO2 | $2.5(-11)$ |  | N5 |
| KPO2 + NO → NO$_2$ + 0.5 CH$_3$CO$_3$ + 0.5 HPAC
 +0.5 HCHO + 0.5 OH + 0.5 MGLY | $2.7(-12)\exp(350/T)$ |  | N5 |
| KPO2 + NO$_3$ → NO$_2$ + 0.5 CH$_3$CO$_3$ + 0.5 HPAC
 +0.5 HCHO + 0.5 OH + 0.5 MGLY | $2.3(-12)$ |  | N5 |
| KPO2 + HO$_2$ → OH + 0.5 CH$_3$CO$_3$ + 0.5 HPAC
 +0.5 HCHO + 0.5 OH + 0.5 MGLY | $2.26(-13)\exp(1300/T)$ |  | N5 |
| DHPAO2 + NO → NO$_2$ + HPACET + OH + PGA | $2.7(-12)\exp(350/T)$ |  | N5 |
| DHPAO2 + NO$_3$ → NO$_2$ + HPACET + OH + PGA | $2.3(-12)$ |  | N5 |
| DHPAO2 + HO$_2$ → OH + HPACET + OH + PGA | $2.64(-13)\exp(1300/T)$ |  | N5 |

| Reaction | Rate | Ref. | Note |
|---|---|---|---|
| $ISOPDOOH + OH \rightarrow 0.85\,IEPOX + 0.15\,DHHEPOX + OH$ | $3.0(-11)\exp(390/T)$ | 9,3,10 | N6 |
| $ISOPDOOH + OH \rightarrow 0.6\,ISOPDO2 + 0.32\,HCOOH + 0.48\,HO_2$ $+0.08\,HCHO + 0.08\,OH + 0.4\,MACR$ | $4.1(-12)\exp(200/T)$ | 9,3 | N8 |
| $ISOPEO2 + NO \rightarrow MACR + HO_2 + HCHO + NO_2$ | $K_{RO2NO} \cdot Y_{oxy}(T, M, 6, 1.27)$ | 1,3 | N3 |
| $ISOPEO2 + NO \rightarrow ISOPENO3$ | $K_{RO2NO} \cdot Y_{nit}(T, M, 6, 1.27)$ | 1,3 | N3 |
| $ISOPEO2 + HO_2 \rightarrow ISOPEOOH$ | $2.1(-13)\exp(1300/T)$ | 1,3 | |
| $ISOPEO2 + ISOPBO2 \rightarrow 0.7\,MVK + 1.4\,HCHO + 1.4\,HO_2$ $+0.3\,ISOPBOH + 0.7\,MACR + 0.3\,HCOC5$ | $1.2(-12)$ | 5 | |
| $ISOPEO2 + ISOPDO2 \rightarrow MACR + HCHO + HO_2 + 0.5\,HCOC5$ $+0.5\,ISOPDOH$ | $1.1(-11)$ | 5 | |
| $ISOPEO2 + ISOPEO2 \rightarrow MACR + HCHO + HO_2$ $+0.5\,HCOC5 + 0.5\,ISOPDOH$ | $5.0(-12)$ | 5 | |
| $ISOPEOOH + OH \rightarrow 0.83\,HYAC + 0.83\,GLY + 0.17\,MACR + HO_2$ | $1.0(-10)$ | 1 | N9 |
| $ISOPENO3 + OH \rightarrow HYAC + ETHLN + HO_2$ | $6.0(-11)$ | 1,11 | N9 |
| $ISOPBNO3 + OH \rightarrow 0.85\,INBO2 + 0.15\,IEPOX + 0.15\,NO_2$ | $8.4(-12)\exp(390/T)$ | 1,3 | |
| $INBO2 \rightarrow 2\,HO_2 + CO + MVKOOH + NO_2$ | $7.5E12 * exp(-10000/T)$ | 3 | N11 |
| $INBO2 + NO \rightarrow HNO_3$ | $K_{RO2NO} \cdot Y_{nit}(T, M, 11, 6.3)$ | 1,3 | N12 |
| $INBO2 + NO \rightarrow 1.85\,NO_2 + 0.85\,GLYALD + 0.85\,HYAC$ $+0.15\,MACRNO3 + 0.15\,HO_2 + 0.15\,HCHO$ | $K_{RO2NO} \cdot Y_{oxy}(T, M, 11, 6.3)$ | 1,13,3 | |
| $INBO2 + NO_3 \rightarrow 1.85\,NO_2 + 0.85\,GLYALD + 0.85\,HYAC$ $+0.15\,MACRNO3 + 0.85\,HO_2 + 0.15\,HCHO$ | $2.3(-12)$ | 1 | |
| $INBO2 + HO_2 \rightarrow HNO_3$ | $2.5(-13)\exp(1300/T)$ | 1,3 | N13 |
| $ISOPDNO3 + OH \rightarrow 0.85\,INDO2 + 0.15\,IEPOX + 0.15\,NO_2$ | $3.9(-11)$ | 1,3 | |
| $INDO2 \rightarrow 3\,HO_2 + 2\,CO + OH + HYAC + NO_2$ | $7.5E12 * exp(-10000/T)$ | 3 | N14 |
| $INDO2 + NO \rightarrow HNO_3$ | $K_{RO2NO} \cdot Y_{nit}(T, M, 11, 7.9)$ | 1,3 | N12 |
| $INDO2 + NO \rightarrow HCHO + HO_2 + MVKNO3 + NO_2$ | $K_{RO2NO} \cdot Y_{oxy}(T, M, 11, 7.9)$ | 1,3,11,12 | |
| $INDO2 + NO3 \rightarrow HCHO + HO_2 + MVKNO3 + NO_2$ | $2.3(-12)$ | 1 | |
| $INDO2 + HO_2 \rightarrow 0.39\,INDOOH + 0.65\,HCHO + 0.65\,HO_2$ $+0.65\,MVKNO3$ | $2.5(-13)\exp(1300/T)$ | 1,3 | |
| $INDOOH + OH \rightarrow 0.39\,INDO2 + 1.22\,HO_2 + 0.61\,CO$ $+0.61\,MVKNO3 + 0.61\,OH$ | $9.2(-12)$ | 1 | N15 |
| $IEPOX + OH \rightarrow 0.19\,ICHE + 0.58\,IEPOXAO2 + 0.23\,IEPOXBO2$ | $4.4(-11)\exp(-400/T)$ | 3 | N16 |
| $ICHE + OH \rightarrow 0.28\,OH + 1.28\,CO + 0.28\,HYAC + 0.72\,MVKO2$ | $1.5(-11)$ | | N17 |
| $ICHE + OH \rightarrow CO + HO_2 + 0.28\,HPDIAL + 0.72\,HPKETAL$ | $2.2(-11)\exp(-400/T)$ | | N18 |

| Reaction | Rate | Ref. | Note |
|---|---|---|---|
| IEPOXAO2 → DHBO + OH + CO | $1.0(7)\exp(-5000/T)$ | 3 | N19 |
| IEPOXAO2 → CO + 2.5 HO$_2$ + 1.5 OH + 0.5 HOBA
+ 0.5 HPDIAL | $1.875(13)\exp(-10000/T)$ | 3 | N20 |
| IEPOXAO2 + NO → NO$_2$ + HO$_2$ + 0.8 MGLY + 0.8 GLYALD
+0.2 DHBO + 0.2 CO | $K_{\mathrm{RO2NO}}$ | 1,3 | |
| IEPOXAO2 + HO$_2$ → OH + HO$_2$ + 0.8 MGLY + 0.8 GLYALD
+0.2 DHBO + 0.2 CO | $1.6(-13)\exp(1300/T)$ | 3 | N21 |
| IEPOXAO2 + HO$_2$ → CO + HO$_2$ + OH + DHBO | $0.8(-13)\exp(1300/T)$ | 3 | N22 |
| IEPOXBO2 → MACROH + OH + CO | $1.0(7)\exp(-5000/T)$ | 3 | N19 |
| IEPOXBO2 → 1.5 CO + 3 HO$_2$ + 0.5 MGLY + 0.5 HPKETAL | $1.875(13)\exp(-10000/T)$ | 3 | N23 |
| IEPOXBO2 + NO → NO$_2$ + HO$_2$ + 0.8 GLY + 0.8 HYAC
+0.2 MACROH + 0.2 CO | $K_{\mathrm{RO2NO}}$ | 1,3 | |
| IEPOXBO2 + HO$_2$ rightarrowOH + HO$_2$ + 0.8 GLY + 0.8 HYAC
+0.2 MACROH + 0.2 CO | $1.6(-13)\exp(1300/T)$ | 3 | N21 |
| IEPOXBO2 + HO$_2$ → CO + HO$_2$ + OH + MACROH | $0.8(-13)\exp(1300/T)$ | 3 | N24 |
| HCOC5 + OH → C59O2 | $3.81(-11)$ | 1 | |
| C59O2 + NO → HYAC + GCO3 + NO$_2$ | $K_{\mathrm{RO2NO}}$ | 1 | |
| C59O2 + NO$_3$ → HYAC + GCO3 + NO$_2$ | $2.3(-12)$ | 1 | |
| C59O2 + HO$_2$ → HYAC + GCO3 + OH | $2.4(-13)\exp(1300/T)$ | 1,3 | N25 |
| C59O2 + CH$_3$O$_2$ → HYAC + GCO3 + HCHO + HO$_2$ | $9.2(-14)$ | 1 | |
| C59O2 + CH$_3$CO$_3$ → HYAC + GCO3 + CO$_2$ + CH$_3$O$_2$ | $1.8(-12)\exp(500/T)$ | 6,7 | |
| ISOPBOH + OH → DHBO + CO | $3.85(-11)$ | 10 | N26 |
| ISOPDOH + OH → 0.9 DHBO + 0.9 CO + 0.1 HCOC5 + 0.1 HO$_2$ | $7.38(-11)$ | 10 | N26 |
| HPALD1 + OH → 0.45 OH + 1.35 CO$_2$ + 0.55 HCHO + 0.65 CH$_3$CO$_3$
+0.2 MMAL + 0.15 MGLY + 0.15 CO + 0.1 GLY | $1.0(-11)$ | 5,3 | N27 |
| HPALD1 + OH → MVK + OH + 0.5 CO + 0.5 CO$_2$ | $0.5(-11)$ | 5,3 | N27 |
| HPALD1 + OH → MVK + OH + CO$_2$ | $1.5(-11)$ | 5,3 | N27 |
| HPALD1 + OH → MVKOOH + OH + CO | $1.4(-11)$ | 5,3 | N27 |
| HPALD1 + OH → ICHE | $0.8(-11)$ | 5,3 | N27 |
| HPALD1 + O$_3$ → 0.35 MGLY + 0.27 GLY + 1.19 OH + 0.65 CO
+0.65 CH$_3$CO$_3$ + 0.08 H$_2$O$_2$ + 0.73 HPAC | $2.4(-17)$ | 1 | |
| HPALD2 + OH → 0.45 OH + 1.35 CO$_2$ + 0.55 HCHO + 0.65 CH$_3$CO$_3$
+0.2 MMAL + 0.15 MGLY + 0.15 CO + 0.1 GLY | $1.0(-11)$ | 5,3 | N28 |

[revised manuscript text omitted]

---

## Author Response (AR2)

Dear Editor, David,

Thank you for accepting our revised manuscript. Find hereafter the list of changes (mostly typos) to the manuscript, and the marked-up manuscript version with all the changes.

Yours sincerely,

Jean-Francois Muller
jfm@aeronomie.be
* * *
**LIST OF CHANGES TO MANUSCRIPT**

- page 3 line 18: remove "and" and add a coma

- page 3 line 19: remove a comma

- page 4 line 11: reactions (plural)

- page 68 line 8: Kurtén (typo)

- page 68 line 10: corrected http page of ACPD paper D'Ambro et al.

- page 72 lines 72-73: added http page of GMDD paper Müller et al.

[revised manuscript text omitted]

+0.9 CH$_3$O$_2$ + 0.9 CO$_2$ + 0.1 CH$_3$COOH + 0.1 HCOC5 | $2.0(-12)\exp(500/T)$ | 6,7 | |
| ISOPDO2 → 0.75 HPALD2 + 0.75 HO$_2$ + 0.15 HPCE
+0.15 OH + 0.1 DHPAO2 | $4.253(+8)\exp(-7254/T)$
$+ 6.29(-19)\exp(4012/T)\cdot[\text{NO}]$
$+ 4.9(-20)\exp(4962/T)\cdot[\text{HO}_2]$ | | N4 |
| ISOPDO2 → MACR + HCHO + OH | $1.77(+11)\exp(-9752/T)$ | 8 | |
| HPCE + OH → 1.82 CO + 0.82 OH + 0.82 HPACET + 0.18 KPO2 | $2.5(-11)$ | | N5 |
| KPO2 + NO → NO$_2$ + 0.5 CH$_3$CO$_3$ + 0.5 HPAC
+0.5 HCHO + 0.5 OH + 0.5 MGLY | $2.7(-12)\exp(350/T)$ | | N5 |
| KPO2 + NO$_3$ → NO$_2$ + 0.5 CH$_3$CO$_3$ + 0.5 HPAC
+0.5 HCHO + 0.5 OH + 0.5 MGLY | $2.3(-12)$ | | N5 |
| KPO2 + HO$_2$ → OH + 0.5 CH$_3$CO$_3$ + 0.5 HPAC
+0.5 HCHO + 0.5 OH + 0.5 MGLY | $2.26(-13)\exp(1300/T)$ | | N5 |
| DHPAO2 + NO → NO$_2$ + HPACET + OH + PGA | $2.7(-12)\exp(350/T)$ | | N5 |
| DHPAO2 + NO$_3$ → NO$_2$ + HPACET + OH + PGA | $2.3(-12)$ | | N5 |
| DHPAO2 + HO$_2$ → OH + HPACET + OH + PGA | $2.64(-13)\exp(1300/T)$ | | N5 |

| Reaction | Rate | Ref. | Note |
|---|---|---|---|
| ISOPDOOH + OH → 0.85 IEPOX + 0.15 DHHEPOX + OH | $3.0(-11)\exp(390/T)$ | 9,3,10 | N6 |
| ISOPDOOH + OH → 0.6 ISOPDO2 + 0.32 HCOOH + 0.48 HO$_2$ +0.08 HCHO + 0.08 OH + 0.4 MACR | $4.1(-12)\exp(200/T)$ | 9,3 | N8 |
| ISOPEO2 + NO → MACR + HO$_2$ + HCHO + NO$_2$ | $K_{\text{RO2NO}} \cdot Y_{\text{oxy}}(T, M, 6, 1.27)$ | 1,3 | N3 |
| ISOPEO2 + NO → ISOPENO3 | $K_{\text{RO2NO}} \cdot Y_{\text{nit}}(T, M, 6, 1.27)$ | 1,3 | N3 |
| ISOPEO2 + HO$_2$ → ISOPEOOH | $2.1(-13)\exp(1300/T)$ | 1,3 | |
| ISOPEO2 + ISOPBO2 → 0.7 MVK + 1.4 HCHO + 1.4 HO$_2$ +0.3 ISOPBOH + 0.7 MACR + 0.3 HCOC5 | $1.2(-12)$ | 5 | |
| ISOPEO2 + ISOPDO2 → MACR + HCHO + HO$_2$ + 0.5 HCOC5 +0.5 ISOPDOH | $1.1(-11)$ | 5 | |
| ISOPEO2 + ISOPEO2 → MACR + HCHO + HO$_2$ +0.5 HCOC5 + 0.5 ISOPDOH | $5.0(-12)$ | 5 | |
| ISOPEOOH + OH → 0.83 HYAC + 0.83 GLY + 0.17 MACR + HO$_2$ | $1.0(-10)$ | 1 | N9 |
| ISOPENO3 + OH → HYAC + ETHLN + HO$_2$ | $6.0(-11)$ | 1,11 | N9 |
| ISOPBNO3 + OH → 0.85 INBO2 + 0.15 IEPOX + 0.15 NO$_2$ | $8.4(-12)\exp(390/T)$ | 1,3 | |
| INBO2 → 2 HO$_2$ + CO + MVKOOH + NO$_2$ | $7.5E12 * exp(-10000/T)$ | 3 | N11 |
| INBO2 + NO → HNO$_3$ | $K_{\text{RO2NO}} \cdot Y_{\text{nit}}(T, M, 11, 6.3)$ | 1,3 | N12 |
| INBO2 + NO → 1.85 NO$_2$ + 0.85 GLYALD + 0.85 HYAC +0.15 MACRNO3 + 0.15 HO$_2$ + 0.15 HCHO | $K_{\text{RO2NO}} \cdot Y_{\text{oxy}}(T, M, 11, 6.3)$ | 1,13,3 | |
| INBO2 + NO$_3$ → 1.85 NO$_2$ + 0.85 GLYALD + 0.85 HYAC +0.15 MACRNO3 + 0.85 HO$_2$ + 0.15 HCHO | $2.3(-12)$ | 1 | |
| INBO2 + HO$_2$ → HNO$_3$ | $2.5(-13)\exp(1300/T)$ | 1,3 | N13 |
| ISOPDNO3 + OH → 0.85 INDO2 + 0.15 IEPOX + 0.15 NO$_2$ | $3.9(-11)$ | 1,3 | |
| INDO2 → 3 HO$_2$ + 2 CO + OH + HYAC + NO$_2$ | $7.5E12 * exp(-10000/T)$ | 3 | N14 |
| INDO2 + NO → HNO$_3$ | $K_{\text{RO2NO}} \cdot Y_{\text{nit}}(T, M, 11, 7.9)$ | 1,3 | N12 |
| INDO2 + NO → HCHO + HO$_2$ + MVKNO3 + NO$_2$ | $K_{\text{RO2NO}} \cdot Y_{\text{oxy}}(T, M, 11, 7.9)$ | 1,3,11,12 | |
| INDO2 + NO3 → HCHO + HO$_2$ + MVKNO3 + NO$_2$ | $2.3(-12)$ | 1 | |
| INDO2 + HO$_2$ → 0.39 INDOOH + 0.65 HCHO + 0.65 HO$_2$ +0.65 MVKNO3 | $2.5(-13)\exp(1300/T)$ | 1,3 | |
| INDOOH + OH → 0.39 INDO2 + 1.22 HO$_2$ + 0.61 CO +0.61 MVKNO3 + 0.61 OH | $9.2(-12)$ | 1 | N15 |
| IEPOX + OH → 0.19 ICHE + 0.58 IEPOXAO2 + 0.23 IEPOXBO2 | $4.4(-11)\exp(-400/T)$ | 3 | N16 |
| ICHE + OH → 0.28 OH + 1.28 CO + 0.28 HYAC + 0.72 MVKO2 | $1.5(-11)$ | | N17 |
| ICHE + OH → CO + HO$_2$ + 0.28 HPDIAL + 0.72 HPKETAL | $2.2(-11)\exp(-400/T)$ | | N18 |

| Reaction | Rate | Ref. | Note |
|---|---|---|---|
| IEPOXAO2 → DHBO + OH + CO | $1.0(7)\exp(-5000/T)$ | 3 | N19 |
| IEPOXAO2 → CO + 2.5 HO$_2$ + 1.5 OH + 0.5 HOBA
+ 0.5 HPDIAL | $1.875(13)\exp(-10000/T)$ | 3 | N20 |
| IEPOXAO2 + NO → NO$_2$ + HO$_2$ + 0.8 MGLY + 0.8 GLYALD
+0.2 DHBO + 0.2 CO | $K_{\text{RO2NO}}$ | 1,3 | |
| IEPOXAO2 + HO$_2$ → OH + HO$_2$ + 0.8 MGLY + 0.8 GLYALD
+0.2 DHBO + 0.2 CO | $1.6(-13)\exp(1300/T)$ | 3 | N21 |
| IEPOXAO2 + HO$_2$ → CO + HO$_2$ + OH + DHBO | $0.8(-13)\exp(1300/T)$ | 3 | N22 |
| IEPOXBO2 → MACROH + OH + CO | $1.0(7)\exp(-5000/T)$ | 3 | N19 |
| IEPOXBO2 → 1.5 CO + 3 HO$_2$ + 0.5 MGLY + 0.5 HPKETAL | $1.875(13)\exp(-10000/T)$ | 3 | N23 |
| IEPOXBO2 + NO → NO$_2$ + HO$_2$ + 0.8 GLY + 0.8 HYAC
+0.2 MACROH + 0.2 CO | $K_{\text{RO2NO}}$ | 1,3 | |
| IEPOXBO2 + HO$_2$ rightarrowOH + HO$_2$ + 0.8 GLY + 0.8 HYAC
+0.2 MACROH + 0.2 CO | $1.6(-13)\exp(1300/T)$ | 3 | N21 |
| IEPOXBO2 + HO$_2$ → CO + HO$_2$ + OH + MACROH | $0.8(-13)\exp(1300/T)$ | 3 | N24 |
| HCOC5 + OH → C59O2 | $3.81(-11)$ | 1 | |
| C59O2 + NO → HYAC + GCO3 + NO$_2$ | $K_{\text{RO2NO}}$ | 1 | |
| C59O2 + NO$_3$ → HYAC + GCO3 + NO$_2$ | $2.3(-12)$ | 1 | |
| C59O2 + HO$_2$ → HYAC + GCO3 + OH | $2.4(-13)\exp(1300/T)$ | 1,3 | N25 |
| C59O2 + CH$_3$O$_2$ → HYAC + GCO3 + HCHO + HO$_2$ | $9.2(-14)$ | 1 | |
| C59O2 + CH$_3$CO$_3$ → HYAC + GCO3 + CO$_2$ + CH$_3$O$_2$ | $1.8(-12)\exp(500/T)$ | 6,7 | |
| ISOPBOH + OH → DHBO + CO | $3.85(-11)$ | 10 | N26 |
| ISOPDOH + OH → 0.9 DHBO + 0.9 CO + 0.1 HCOC5 + 0.1 HO$_2$ | $7.38(-11)$ | 10 | N26 |
| HPALD1 + OH → 0.45 OH + 1.35 CO$_2$ + 0.55 HCHO + 0.65 CH$_3$CO$_3$
+0.2 MMAL + 0.15 MGLY + 0.15 CO + 0.1 GLY | $1.0(-11)$ | 5,3 | N27 |
| HPALD1 + OH → MVK + OH + 0.5 CO + 0.5 CO$_2$ | $0.5(-11)$ | 5,3 | N27 |
| HPALD1 + OH → MVK + OH + CO$_2$ | $1.5(-11)$ | 5,3 | N27 |
| HPALD1 + OH → MVKOOH + OH + CO | $1.4(-11)$ | 5,3 | N27 |
| HPALD1 + OH → ICHE | $0.8(-11)$ | 5,3 | N27 |
| HPALD1 + O$_3$ → 0.35 MGLY + 0.27 GLY + 1.19 OH + 0.65 CO
+0.65 CH$_3$CO$_3$ + 0.08 H$_2$O$_2$ + 0.73 HPAC | $2.4(-17)$ | 1 | |
| HPALD2 + OH → 0.45 OH + 1.35 CO$_2$ + 0.55 HCHO + 0.65 CH$_3$CO$_3$
+0.2 MMAL + 0.15 MGLY + 0.15 CO + 0.1 GLY | $1.0(-11)$ | 5,3 | N28 |

[revised manuscript text omitted]